# Machine Learning with Physics Knowledge for Prediction: A Survey

**Joe Watson**[1,2,†], **Chen Song**[6], **Oliver Weeger**[3,7], **Theo Gruner**[1,3], **An T. Le**[1], **Kay Pompetzki**[1], **Ahmed Hendawy**[1,3], **Oleg Arenz**[1], **Will Trojak**[8], **Miles Cranmer**[9], **Carlo D'Eramo**[1,3,4], **Fabian Bülow**[6], **Tanmay Goyal**[6], **Jan Peters**[1,2,3,5], **Martin W. Hoffman**[6]

*{joe, theo, an, kay, ahmed, oleg, carlo, jan}@robot-learning.de*
*{chen.song, martin.w.hoffmann, fabian.buelow,tanmay.goyal}@de.abb.com*
*weeger@cps.tu-darmstadt.de*
*w.trojak@ibm.com*

[1] *Department of Computer Science, Technical University of Darmstadt, Germany*
[2] *Systems AI for Robot Learning, German Research Center for AI (DFKI), Germany*
[3] *Hessian Center for Artificial Intelligence (hessian.AI), Germany*
[4] *Center for Artificial Intelligence and Data Science, University of Würzburg, Germany*
[5] *Centre for Cognitive Science, Technical University of Darmstadt, Germany*
[6] *ABB Corporate Research Center, Mannheim, Germany*
[7] *Department of Mechanical Engineering, Technical University of Darmstadt, Germany*
[8] *IBM Research UKI, United Kingdom*
[9] *Data Intensive Science, University of Cambridge, United Kingdon*
[†] *Now at the Oxford Robotics Institute, University of Oxford*

**Reviewed on OpenReview:** *https://openreview.net/forum?id=ZiJYahyXLU*

## Abstract

This survey examines the broad suite of methods and models for combining machine learning with physics knowledge for prediction and forecasting, with a focus on partial differential equations. These methods have attracted significant interest because of their potential impact on the advancement of scientific research and industrial practices, promising improvements to using small- or large-scale datasets and expressive predictive models with useful inductive biases. The survey has two parts. The first considers incorporating physics knowledge on an architectural level through objective functions, structured predictive models, and data augmentation. The second considers *data* as physics knowledge, which motivates looking at multi-task, meta, and contextual learning as an alternative approach to incorporating physics knowledge in a data-driven fashion. Finally, we also provide an industrial perspective on the application of these methods and a survey of the open-source ecosystem for physics-informed machine learning.

## 1 Introduction

Prediction lies at the heart of science and engineering, and many advances involve the discovery of simple patterns — often expressed as symbolic expressions — that approximate the evolution of our physical world. Still, these mathematical models are only achieved through some degree of simplification and abstraction, and thus rarely capture the complexity of the physical system in its full fidelity. As a result, there is significant interest in learning models from only measurements of the real world, spurring the development of machine learning (ML) (Bishop and Nasrabadi, 2006) for the physical sciences. This field relies on using large datasets of measurements to constrain highly flexible parametric models rather than evoking domain knowledge.

Machine learning methods differ greatly in the small- and big-data regimes. The small data regime, popularized since the early days of machine learning, covering state estimation, system identification, and kernel

methods, considers the setting where data is used to estimate a few unknown parameters given many assumptions (Chiuso and Pillonetto, 2019). The big data era, popularized by deep learning, can leverage large-scale datasets to train over-parameterized models that learn their own internal representations of underlying systems (L'heureux et al., 2017). Fueled by the ever-increasing data and compute available, machine learning can now directly learn internal representations, bypassing the need for extensive domain expertise and manual feature engineering (Goodfellow et al., 2016). This explosion in data capacity is further complemented by advancements in hardware and software. Development in graphical processor units (GPUs) has increased both the size of models that can be designed and the speed at which they can be trained. Mature libraries optimize linear algebra computations, allowing these complex tasks to be implemented easily and computed efficiently.

However, for applications that require a high degree of reliability, robustness, and trust in their predictions, purely data-driven models could be deemed insufficient, especially when there is limited data. A natural compromise is to build predictive models that leverage the prior knowledge obtained through centuries of scientific study with the flexibility and scale of modern machine learning (Karniadakis et al., 2021; Karpatne et al., 2022). Achieving this requires models that can integrate prior knowledge in the form of *inductive biases* (Baxter, 2000), e.g., additional task-informed structure incorporated into the model, objective, or learning algorithm. This added structure acts as a filter, essentially guiding the model to focus on solutions that align with scientific understanding. As a result, the model's search space becomes smaller, with less irrelevant territory to explore. This targeted exploration allows the model to learn effectively even with limited data. The key challenge lies in designing these inductive biases carefully. If the model is too restrictive, the resulting solution might be suboptimal due to underfitting. This review surveys such methods to inform machine learning models with prior physics knowledge for prediction.

There are several tasks where such physics-informed models are useful. One is complex forecasting tasks, such as weather prediction, where lots of measurement data are available, and traditional physics models struggle to model such chaotic systems that involve multiple spatial and temporal scales (Schultz et al., 2021; Kurth et al., 2023). Another is solving inverse problems (Ghattas and Willcox, 2021), where a realistic physics model can be examined and queried to calculate what input conditions a system needs for a desired output. These results can then inform the design and operation of the system. Additionally, active fault monitoring is crucial in industry for ensuring the smooth operation of production. By integrating physical models with real-time measurements, these systems assess operational accuracy and predict potential failures, ensuring system reliability (Aldrich and Auret, 2013). This survey covers the diverse ways physics knowledge can be incorporated into ML across datasets $\mathcal{D}$, objectives $\mathcal{L}$, and models $u_\theta$ (Figure 1). Physics-informed models are architectures and prediction techniques informed by domain knowledge, covered in sections 3.1,

| Section | Solve the simulation | Infer the system | Generalize across systems | Inductive bias |
|---|---|---|---|---|
| 3.1: Differential equations | ✗ | ✓ | ✗ | prediction in continuous time |
| 3.2: Parameterizing equations | ✗ | ✓ | ✗ | physics-informed model |
| 3.3: Physics-informed losses | ✓ | ✗ | ✗ | physics-informed objective |
| 3.4: Neural operators | ✓ | ✗ | ✓ | multi-instance learning |
| 3.5: Latent variable model | ✗ | ✓ | ✗ | latent features and dynamics |
| 3.6: System identification | ✓/✗ | ✓ | ✗ | physics-informed models, objectives |
| 3.7: Model invariances | ✗ | ✓ | ✗ | physics-inspired invariances |
| 4.1: Multi-task learning | ✓/✗ | ✓/✗ | ✓ | multi-instance learning |
| 4.2: Meta learning | ✓/✗ | ✓/✗ | ✓ | multi-instance learning |
| 4.3: Neural processes | ✓ | ✗ | ✓ | multi-instance, contextual learning |

Table 1: The review is structured to cover the various ways learning can be incorporated into physics-informed learning. Here, learning can refer to the solution (or simulation) of a physical system, inferring the physical system from data, and learning a model that generalizes across several instantiations of the physical system, e.g., different boundary conditions.

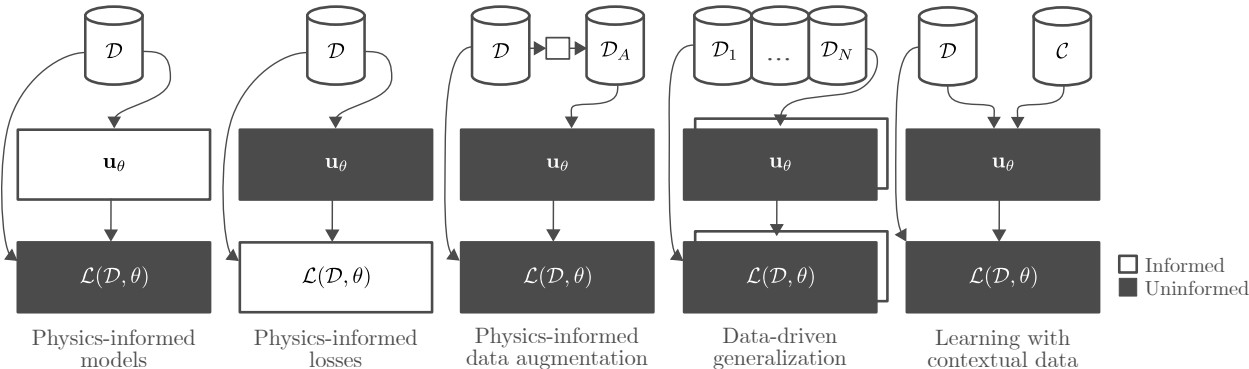

Figure 1: This survey covers the diverse ways physics knowledge can be incorporated into ML across datasets $\mathcal{D}$, objectives $\mathcal{L}$ and models $\mathbf{u_\theta}$. Physics-informed models are architectures and prediction techniques informed by domain knowledge, covered in sections 3.1, 3.2, and 3.5. Learning arbitrary models with physics-informed losses is discussed in sections 3.3 and 3.6. Data augmentation can also incorporate physics knowledge to encode invariances (section 3.7). Data-driven learning across multiple datasets and experiments includes neural operators (section 3.4), multi-task learning (section 4.1) and meta-learning. Finally, contextual data is another data-driven approach to incorporate additional domain knowledge, as seen in operator learning (section 3.4) and neural processes (section 4.3).

3.2, and 3.5. Learning arbitrary models with physics-informed losses is discussed in sections 3.3 and 3.6. Various ways to encode physics knowledge in the form of model invariances are covered in section 3.7. Data-driven learning across multiple datasets and experiments includes neural operators (section 3.4), multi-task learning (section 4.1) and meta-learning (section 4.2), and has been used for both physics-informed and uninformed models and losses. Finally, contextual data is another data-driven approach to incorporate additional domain knowledge, as seen in operator learning (section 3.4) and neural processes (section 4.3). This review is structured to cover many of these cases, as shown in Table 1.

**Contribution.** Machine learning that leverages prior knowledge from sciences and engineering is a rapidly developing field. Several surveys have already been conducted, focusing on specific techniques such as physics-informed neural networks (PINNs) (Karniadakis et al., 2021; Pateras et al., 2023), neural operators (Huang et al., 2022a), Koopman operators (Brunton et al., 2022), or exploring how best to combine scientific knowledge with machine learning methods (Von Rueden et al., 2021; Seyyedi et al., 2023). Wang and Yu (2021) recently provided a survey that focuses primarily on deep learning models for dynamical systems with a focus on architectures and objectives, and therefore overlaps with our survey in areas such as physics-based equivariances. However, our survey takes a broader machine learning perspective, investigating how prior physics knowledge can be incorporated through not only architectures, but also datasets, and the combination of the two. The survey is designed to fulfill several key objectives. Firstly, our aim is to identify and communicate the outstanding challenges in physics-informed learning to machine learning researchers, providing a pathway for future research directions. Secondly, the survey offers critical guidance, positioning physics-informed learning concepts within the broader landscape of machine learning and connections between various engineering domains. Thirdly, we establish a refined method taxonomy, providing a structured framework for categorizing and distinguishing the different approaches. Additionally, the survey assesses frameworks for evaluating the industrial relevance of these methods and use cases. Lastly, we identify future trends that could shape industrial applications. Through these contributions, the survey seeks to bridge the gap between theoretical research and practical implementation, making physics-informed machine learning more powerful and useful to tackle tough real-world problems.

## 2 Background

This section delves into the technical concepts foundational to this review, with a particular focus on the physical and mathematical modeling techniques employed in sciences and engineering.

### 2.1 Ordinary Differential Equations

An ordinary differential equation (ODE) involves functions of one independent variable and their derivatives (Hale, 2009). ODEs emerge across a broad spectrum of scientific and engineering disciplines, modeling phenomena that exhibit temporal changes in quantities. Examples include the motion of celestial bodies, population dynamics, chemical reactions, and mechanical systems, among others. In many cases, the relationship between a (scalar or vector-valued) function $\boldsymbol{u}$ and its rate of change $\boldsymbol{f}$ can be expressed (or re-formulated) in terms of the initial value problem of a first-order ODE as:

$$\frac{\mathrm{d}\boldsymbol{u}}{\mathrm{d}t} = \boldsymbol{f}(t, \boldsymbol{u}(t), \boldsymbol{\theta}), \quad t \in \mathcal{T} = [0, T], \tag{1}$$

$$\boldsymbol{u}(0) = \boldsymbol{u}_0. \tag{2}$$

Solving ODEs involves finding a function $\boldsymbol{u}(t)$ that satisfies the given equation (1) under specific initial conditions (equation (2)). Those methods are often categorized into two families: analytical and numerical. Analytical approaches, such as separation of variables, characteristics equations, integrating factor, or eigenfunction methods, aim to find exact solutions under deep insights into the equation structure. When analytical solutions are not possible or intractable, numerical methods become crucial to approximate the solution by breaking down the desired interval into smaller steps. In the numerical solutions of ordinary differential equations, they are commonly categorized into explicit and implicit schemes, each possessing distinct features and applicability. Explicit methods compute the state of the system at a future time solely based on the current state, it makes them straightforward and computationally efficient. For example, the classic Euler and Runge-Kutta methods. These methods are generally preferred for problems that require computational speed and do not exhibit stiff behavior. In contrast, implicit methods involve solving one or more equations to find the future state, since the future state itself is also part of the equation. This design makes these methods, such as backward Euler or Crank-Nicolson, more computationally expensive but significantly more stable compared to explicit methods. They are particularly useful for stiff equations where explicit schemes might fail or require very small time steps to maintain accuracy and stability.

Furthermore, solving systems of ODEs is crucial for analyzing complex systems, such as when addressing partial differential equations (PDEs), see also section 2.2, where the method-of-lines emerges as a key technique. This method converts PDEs into ODE systems by discretizing the spatial domain and considering time as a continuous variable, simplifying their solution. For further details, see Ascher and Petzold (1998).

### 2.2 Partial Differential Equations

While ODEs provide a robust framework for modeling time dependent processes, many real-world phenomena involve interdependencies across multiple dimensions. PDEs extend the principles of ODEs by incorporating spatial variability and are used to model systems where the state depends on multiple variables. This section covers the general form of PDEs, including boundary and initial conditions, and discusses various numerical methods for solving them, highlighting their critical role in modeling.

The general form of a partial differential equation for a dynamical system such as heat conduction, fluid dynamics, structure mechanics, or electromagnetics, can be defined as (Evans, 2022),

$$\mathsf{L}_{\boldsymbol{a}}[\boldsymbol{u}(\boldsymbol{x})] = \boldsymbol{f}(\boldsymbol{x}), \qquad\qquad \boldsymbol{x} \in \Omega, \tag{3}$$

$$\boldsymbol{c}_1(\boldsymbol{x})\boldsymbol{u}(\boldsymbol{x}) + \boldsymbol{c}_2(\boldsymbol{x})\frac{\partial \boldsymbol{u}}{\partial \boldsymbol{n}} = \boldsymbol{b}(\boldsymbol{x}), \qquad\qquad \boldsymbol{x} \in \partial\Omega. \tag{4}$$

$\mathsf{L}_{\boldsymbol{a}}$ is a differential operator that contains partial derivatives of the function $\boldsymbol{u} : \Omega \to \mathbb{R}^{d_u}$ with respect to spatial coordinates and (potentially also) time, and $\Omega$ is the space-domain domain containing the set of all

points in both a bounded space and a time interval over which equation (3) is defined. $\boldsymbol{f}(\boldsymbol{x})$ represents a source term that may influence the system's behavior, and $\boldsymbol{a} : \Omega \to \mathbb{R}^{d_a}$ are the parameters of the differential operator, which could vary across $\boldsymbol{x}$. Moreover, $\boldsymbol{b}$ is the boundary condition and $\boldsymbol{n}$ is normal to the spatial boundary. Notably, equation (4) addresses the initial conditions at the starting time and accommodates three principal types of boundary conditions, i.e., Dirichlet, Neumann, and Robin boundary conditions depending on the value of $\boldsymbol{c}_1$ and $\boldsymbol{c}_2$.

While the analytical solution of PDEs is restricted to very specific problems, there have been enormous developments in numerical methods for approximating $\boldsymbol{u}$ and these methods rely on discretization, which is a process that transforms the continuous nature of space and time into a series of discrete points. Spatially, the common approaches are finite difference (LeVeque, 2007), finite volume (LeVeque, 2002), and finite element methods (Hughes, 2003). Temporally, one breaks the time interval into multiple steps. Here, methods like the method of lines convert the PDE to a system of ordinary differential equations solvable by techniques such as Runge-Kutta. Even a finite difference scheme can be used for temporal discretization. Beyond these separate approaches, advanced methods like finite element space-time methods tackle both spatial and temporal aspects together, offering increased efficiency and accuracy for complex problems. Needless to say, there are many numerical methods developed in approximating $\boldsymbol{u}$ in a PDE, we only focus on the few common approaches in this review. The finite difference method (FDM) approximates derivatives by differences, essentially replacing continuous changes with discrete steps. It then solves the resulting equation on a grid of points (Thomas, 2013). The finite volume method (FVM) , similar to FDM, focuses on the conservation laws over discrete volumes, such as fluid dynamics and heat transfer problems (Eymard et al., 2000). The finite element method (FEM) breaks down the domain into smaller, simpler parts called elements. Within each element, the solution is approximated using specific basis functions that can accurately represent the behavior of $u$ (Szabó and Babuška, 2021). Spectral methods, which employ globally defined high-order basis functions, offer exceptional accuracy for problems with regular geometries. These methods represent the solution as a sum of mathematical functions like Fourier series or polynomials on the domain (Canuto et al., 2007). In contrast, mesh-free methods do not require a pre-defined mesh. They employ scattered points throughout the domain to represent the solution, making them well-suited for problems with complex geometries or discontinuities (Griebel et al., 2005).

## 2.3 System Identification

Solving ODEs and PDEs is sometimes referred to as the *forward* problem, simulating a physical system to a satisfactory fidelity. The inverse problem, inferring the parameters of a physical system from data, is also of interest (Colton et al., 1990). This setting has many names. *System identification* is the classical name from the control theory literature (Åström and Eykhoff, 1971), where typically some domain knowledge is assumed and only control input and measurement data is available. In more modern machine learning terminology, *model learning* or *regression* can also be used (Bishop and Nasrabadi, 2006), but these terms apply to a wider set of problems that do not necessarily involve timeseries data, domain knowledge or dynamical systems.

System identification involves defining the dynamics of a system through a sequence of inputs $\boldsymbol{w}_t$, measurements $\boldsymbol{y}_t$, and an initial state $\boldsymbol{x}_1$ over a finite horizon $\mathcal{T}$ as a trajectory, i.e. $\boldsymbol{Y} = [\boldsymbol{y}_1, \ldots, \boldsymbol{y}_T]$ (Tangirala, 2018). Typically, a Markovian dynamical system is assumed, i.e. the state $\boldsymbol{x}_{t+1} = \boldsymbol{f}_{\boldsymbol{\theta}}(\boldsymbol{x}_t, \boldsymbol{w}_t)$ and the observation $\boldsymbol{y}_t = \boldsymbol{g}_{\boldsymbol{\theta}}(\boldsymbol{x}_t)$ are both parameterized by $\boldsymbol{\theta}$, and the mapping between states and observations $\boldsymbol{g}$ is often known by design to simplify the learning problem and ensure observability. Moreover, additive noise terms may be added to capture uncertainty in the dynamics and sensors, resulting in likelihoods $p(\boldsymbol{x}_{t+1} \mid \boldsymbol{x}_t, \boldsymbol{w}_t, \boldsymbol{\theta})$ and $p(\boldsymbol{y}_t \mid \boldsymbol{x}_t, \boldsymbol{\theta})$. As the trajectory data is not independently and identically distributed (IID), a typical regression approach on the dataset may not produce satisfactory results, as the forecast may drift due to accumulating prediction errors over time. One possible solution is backpropagation through time,

$$\mathcal{L}_{\text{BPTT}}(\boldsymbol{\theta}) = \sum_{k=1}^{K} \sum_{t=1}^{T-h} \sum_{j=1}^{h} \mathcal{L}\left(\boldsymbol{y}_{t+j+1}^{(k)}, \boldsymbol{g}_{\boldsymbol{\theta}}\left(\boldsymbol{f}_{\boldsymbol{\theta}} \circ \cdots \circ \boldsymbol{f}_{\boldsymbol{\theta}}\left(\boldsymbol{x}_t^{(k)}, \boldsymbol{w}_t^{(k)}\right)\right)\right), \tag{5}$$

this loss is considered as the prediction error $\mathcal{L}$ for $K$ trajectories over a time horizon $h$. This captures the prediction problem's sequential nature but requires a more expensive objective that may be harder to optimize. A probabilistic approach considers the joint distribution $p(\boldsymbol{Y}, \boldsymbol{X} \mid \boldsymbol{W}, \boldsymbol{\theta})$ which, continuing the Markov

assumption, factorizes into $p(\boldsymbol{x}_1) \prod_{t=1}^{T} p(\boldsymbol{y}_t \mid \boldsymbol{x}_t, \boldsymbol{\theta}) p(\boldsymbol{x}_{t+1} \mid \boldsymbol{x}_t, \boldsymbol{w}_t, \boldsymbol{\theta})$. The latent states $\boldsymbol{X}$ are inferred using Bayesian filtering and smoothing algorithms (Särkkä and Svensson, 2023), and the model parameters $\boldsymbol{\theta}$ are optimized to maximize the log marginal likelihood of the measurements $\mathcal{L}_{\text{LML}}(\boldsymbol{\theta}) = \log \int p(\boldsymbol{Y}, \boldsymbol{X} \mid \boldsymbol{W}, \boldsymbol{\theta}) \, \mathrm{d}\boldsymbol{X}$ using algorithms such as expectation maximization (EM). Approximate inference techniques are required if the posterior $p(\boldsymbol{X} \mid \boldsymbol{Y}, \boldsymbol{W}, \boldsymbol{\theta})$ can not be computed in closed form.

### 2.4 Function Approximation with Neural Networks

We denote a parametric function $\boldsymbol{f_\theta} : \mathbb{R}^{d_x} \to \mathbb{R}^{d_y}$ parameterized by some $\boldsymbol{\theta} \in \Theta$. This review will focus on the deep learning approaches to function approximation (Goodfellow et al., 2016), but we will also touch on Gaussian processes (Rasmussen et al., 2006).

A (feed-forward) neural network (FFNN) is comprised of $L$ consecutive nonlinear operations $\boldsymbol{g}^{(l)} : \mathbb{R}^{d_l} \to \mathbb{R}^{d_{l+1}}$, where $d_1 = d_x$ and $d_L = d_y$, on internal representations $\boldsymbol{z}^{(l)}$:

$$\boldsymbol{f_\theta}(\boldsymbol{x}) = \boldsymbol{g}^{(L)}_{\sigma_L, \boldsymbol{\theta}_L} \circ \boldsymbol{g}^{(L-1)}_{\sigma_{L-1}, \boldsymbol{\theta}_{L-1}} \circ \dots \boldsymbol{g}^{(1)}_{\sigma_1, \boldsymbol{\theta}_1}(\boldsymbol{x}). \tag{6}$$

Each $\boldsymbol{g}^{(l)}$ is specified by a nonlinear function $\sigma$ and an affine transform:

$$\boldsymbol{g}^{(l)}_{\sigma_i, \boldsymbol{\theta}_i}(\boldsymbol{z}^{(l)}) = \sigma_i(\boldsymbol{W}_i \boldsymbol{z}^{(l)} + \boldsymbol{b}_i), \tag{7}$$

where $\boldsymbol{\theta}_i = \{\boldsymbol{W}_i, \boldsymbol{b}_i\}$ are the weights and biases of the layer $l$ and $\boldsymbol{\theta} = \{\boldsymbol{\theta}_L, \dots, \boldsymbol{\theta}_1\}$ is the set of parameters of the FFNN. $\sigma$ is an element-wise nonlinear 'activation' function, such as the sigmoid function, tanh, and rectified linear unit (ReLU). These models are typically referred to as multi-layer perceptions (MLP).

Given a dataset of input and output data $\{\boldsymbol{x}_j, \boldsymbol{y}_j\}$, supervised machine learning determines parameters $\boldsymbol{\theta}$ of a neural network (NN) such that $\boldsymbol{y}_j \approx \boldsymbol{f_\theta}(\boldsymbol{x}_j)$. This training can be performed efficiently using the backpropagation algorithm, which implements the chain rule efficiently for the network structure using dynamic programming. For training these models, it is common to use accelerated first-order methods such as stochastic gradient descent (SGD) with momentum and Adam (Kingma and Ba, 2014). Random mini-batch updates are also deployed to alleviate memory issues and incorporate random exploration. Many deep learning architectures exist with specific inductive biases well suited to certain tasks.

For sequential tasks that require memory, recurrent neural networks (RNNs) (Rumelhart et al., 1985) have a stateful representation state $\boldsymbol{z}^s$ so that $(\boldsymbol{y}_t, \boldsymbol{z}^s_{t+1}) = \boldsymbol{f_\theta}(\boldsymbol{x}_t, \boldsymbol{z}^s_t)$. A non-Markovian alternative is self-attention, which computes its own attention weights that dictate which values inform the internal representation. Mathematically, this is written as $\boldsymbol{Z}^{(l)} = \text{Softmax}(\boldsymbol{Q_\theta}(\boldsymbol{Z}^{(l-1)})\boldsymbol{K_\theta}(\boldsymbol{Z}^{(l-1)})^\top / \sqrt{d})\boldsymbol{V_\theta}(\boldsymbol{Z}^{(l-1)})$, where $\boldsymbol{Z}$ is a sequence of representations. The softmax operation and 'query' $\boldsymbol{Q}$ and 'key' $\boldsymbol{K}$ terms compute the self-attention weights that scale the 'value' terms $\boldsymbol{V}$. The outer product in the softmax means self-attention scales quadratically in the sequence length. Self-attention can be stacked in a scalable fashion to form the transformer architecture, which is capable of learning rich correlations between complex sequences of data, such as natural language (Vaswani et al., 2017).

For structured data that can be defined using a graph with nodes N and edges E, a graph neural network computes its representations by aggregating across the graph using messaging passing, i.e., $\boldsymbol{z}_i^{(l)} = \sigma\left(\sum_{j \in \text{N}_i} \boldsymbol{f}_{\boldsymbol{\theta}}^{(l)}(\boldsymbol{z}_i^{(l-1)}, \boldsymbol{z}_j^{(l-1)}, \text{E}_{ij})\right)$, applying the function approximator to neighboring representations (Bronstein et al., 2021).

Finally, auto-encoding architectures are used to learn a compressed representation $\boldsymbol{z}^c$ of data using an 'encoder' / 'decoder' architecture, minimizing any reconstruction error from the use of this representation $\hat{\boldsymbol{x}} = \boldsymbol{h_\theta}(\boldsymbol{z}^c), \boldsymbol{z}^c = \boldsymbol{g_\theta}(\boldsymbol{x})$. Throughout this survey, these architectures are seen as a means to design function approximators with relevant structural properties for prediction. For further technical details regarding deep learning, please refer to Goodfellow et al. (2016).

## 3 Knowledge-Driven Priors from Physics

Many machine learning methods are inspired by domain knowledge, such as the families of geometric deep neural networks (Gerken et al., 2023), e.g., convolution neural networks and graph neural networks, to

capture equivariant features from data efficiently. This section reviews techniques for efficiently learning differential equations, either through data, the model architecture, the learning objective, or the underlying computation.

### 3.1 Learning Differential Models from Data

A challenge in learning models of physical processes is that our reality exists in continuous time, while the numerical algorithms require discretization. A useful inductive bias in this setting is to define a black-box model in continuous time by incorporating integration into prediction. The result is a more flexible predictive model that can learn and forecast on irregular time intervals rather than a rigid discretization. This approach is referred to as learning the differential equations.

**Ordinary differential equations.** The connection between ODEs and deep learning, the neural ODE, enjoys a long history. Perhaps the earliest predecessor to 'deep' learning of differential equations from data is the ResNet architecture (i.e., residual flow block) proposed by He et al. (2016), although the original motivation was to mitigate gradient attenuation for parameters in early layers. A ResNet block defines a discrete-time residual transformation

$$\boldsymbol{x}_{t+1} = \boldsymbol{x}_t + \boldsymbol{f}_{\boldsymbol{\theta}_t}(\boldsymbol{x}_t), \tag{8}$$

where $\boldsymbol{f}$ is a transformation block with parameters $\boldsymbol{\theta}_t$, $\boldsymbol{x}_t$ and $\boldsymbol{x}_{t+1}$ are the input and output at time step $t$, respectively. Inspired by the ResNet architecture, Weinan (2017) and Haber et al. (2018) both proposed to parameterize the (infinitesimal) continuous-time transformation

$$\frac{\mathrm{d}\boldsymbol{x}(t)}{\mathrm{d}t} = \boldsymbol{f}_{\boldsymbol{\theta}}(\boldsymbol{x}(t), t), \tag{9}$$

which is equivalent to the discrete-time transformation $\boldsymbol{x}_{t+1} - \boldsymbol{x}_t = \epsilon \, \boldsymbol{f}_{\boldsymbol{\theta}}(\boldsymbol{x}_t)$ with an infinitesimal residual block $\epsilon \, \boldsymbol{f}_{\boldsymbol{\theta}}(\cdot)$ as $\epsilon \to 0$. From the neural network perspective, Chen et al. (2018) shows that equation (9) is equivalent to a 'continuous-depth' neural network with starting input $\boldsymbol{x}_0$ and continuous weights $\boldsymbol{\theta}$. The invertibility of such networks naturally comes from the theorem of the existence and uniqueness of the ODE solution.

Predicting and training with a neural ODE requires a choice of ODE solving technique, which in turn requires discretization of the time variable, see section 2.1. The power of neural ODE comes from its memory efficiency from training without backpropagation through the solver, thus enabling it to learn sequences of long-horizon data efficiently.

**Training neural ODEs.** The common optimization paradigm in learning neural function approximations require a sufficiently large dataset to prevent overfitting, especially for highly nonlinear dynamics. When data is scarce, alternative strategies are necessary. Classical system identification methods (Voss et al., 2004) estimate structured model parameters using statistical techniques. Meanwhile, hybrid neural system identification approaches (Chen et al., 2021; Forgione and Piga, 2021) incorporate inductive biases, such as physics-based constraints or regularized latent representations, to improve generalization. To obtain the gradient of the objective w.r.t. the parameters $\boldsymbol{\theta}$, it is required to propagate the gradients through the numerical integration. A straightforward approach is to store all of the function evaluations in the forward pass of the ODE and backpropagate through the whole numerical integration. Yet, the space complexity increases linearly with time $T$, which becomes a computational burden for long-time series and high-order numerical solvers (Norcliffe and Deisenroth, 2023). Given a scalar-valued loss function $L$, the adjoint of $\boldsymbol{x}(t)$, $\boldsymbol{a}(t) = \mathrm{d}L/\mathrm{d}\boldsymbol{x}(t)$, having the dynamic derived from the chain rule,

$$\frac{\mathrm{d}\boldsymbol{a}(t)}{\mathrm{d}t} = -\boldsymbol{a}(t)^{\mathsf{T}} \frac{\partial \boldsymbol{f}_{\boldsymbol{\theta}}(\boldsymbol{x}(t), t)}{\partial \boldsymbol{x}}, \tag{10}$$

provides a memory-efficient way to obtain the gradient of the loss function by integrating an additional time-reversed ODE (Chen et al., 2018)

$$\frac{\mathrm{d}\mathcal{L}}{\mathrm{d}\boldsymbol{\theta}} = -\int_{t_1}^{t_0} \boldsymbol{a}(t)^{\mathsf{T}} \frac{\partial \boldsymbol{f}_{\boldsymbol{\theta}}(\boldsymbol{x}(t), t)}{\partial \boldsymbol{\theta}} \mathrm{d}t. \tag{11}$$

As the gradient is obtained as the solution of the time-reversed ODE, the gradients can be evaluated without storing any additional data. Thus, the adjoint method optimizes computational resources by balancing time against space complexity. Several works (Gholaminejad et al., 2019; Zhuang et al., 2020; Onken and Ruthotto, 2020) showed that the gradient estimation based on the adjoint method is error-prone due to the numerical inaccuracies of solving the backward ODE, due to the typical high Lyapunov characteristic exponent of highly nonlinear ODEs. Many improved backward integrators have been proposed to mitigate this issue (Zhuang et al., 2020; 2021a; Gholaminejad et al., 2019). Concurrently, Ott et al. (2020) studied the validity of neural ODEs representing continuous flows, as true continuous-time analogs of ResNets. They claimed that valid neural ODE is invariant to solver configurations, achievable through critical step-size adaptations for integrators. Krishnapriyan et al. (2023) then generalized neural ODE architecture to RK4 integration scheme, showing generalization and convergence to true continuous flow.

**Augmented neural ODEs.** In system identification tasks, where a dynamical system is to be learned from measurements, see section 2.3, formulating a (neural) ODE in terms of the observable quantities is typically not sufficient to describe the system dynamics, see also section 3.5. The idea of augmented neural ODEs (Dupont et al., 2019) is thus to augment the space of (observable) state variables $\boldsymbol{x}$ with additional variables $\boldsymbol{h}$ that lift the dimensionality of the problem:

$$\frac{\mathrm{d}}{\mathrm{d}t} \begin{bmatrix} \boldsymbol{x}(t) \\ \boldsymbol{h}(t) \end{bmatrix} = \boldsymbol{f_\theta} \left( \begin{bmatrix} \boldsymbol{x}(t) \\ \boldsymbol{h}(t) \end{bmatrix}, t \right). \tag{12}$$

Augmented neural ODEs allow observed trajectories to intersect and enable the learning of more general dynamics over the observed state space. Furthermore, according to Dupont et al. (2019), augmented neural ODEs can achieve lower losses, better generalization, and lower computational cost than regular neural ODEs, also since time discretizations can be coarser as the learned rate functions $\boldsymbol{f_\theta}$ are smoother. However, challenges are the determination of the size of $\boldsymbol{h}$ and training without knowing the values of $\boldsymbol{h}$.

**Partial differential equations.** Learning and solving PDEs have recently emerged as significant areas of interest, reflecting the complexities inherent in accurately capturing spatiotemporal dynamics across extremely fine temporal and spatial scales. In contrast to the ODE operator (equation (9)), PDEs involve working with infinite-dimensional spaces. Specifically, the learning process involves creating a function that maps between two function spaces, representing the input data (like initial conditions) and the output solutions of the PDEs.

We highlight that the adjoint method has been used in the context of gradient-based shape optimization (Zahr and Persson, 2016). However, utilizing the adjoint method for learning PDE is unexplored. Earliest works propose to learn a convolutional neural network as a map between finite Euclidean spaces $f_\theta$ (Guo et al., 2016; Zhu and Zabaras, 2018; Bhatnagar et al., 2019). Building on these approaches, auto-regressive methods have been introduced to estimate the solution at discrete time steps $\boldsymbol{u}(\boldsymbol{x}, t + \Delta t) = f_\theta(\Delta t, \boldsymbol{u}(\boldsymbol{x}, t))$, and either, mix with conventional numerical methods in classical PDE solvers (Bar-Sinai et al., 2019; Greenfeld et al., 2019; Hsieh et al., 2019), or propagate the solution through time by recurrent calls of the networks (Brandstetter et al., 2022). All previous learning PDE methods have in common that they rely on a discretization of the domain, yet in many scenarios, it is desirable to evaluate the solution at any point in the domain. For these reasons, learning approaches that directly learn a mapping from a point in the domain to its solution $u_\theta$ have been actively explored in recent years (Yu et al., 2018; Raissi et al., 2019; Bar and Sochen, 2019), which introduces physics constraints as losses, is covered in detail in section 3.3. Yet, these models come at the cost that they provide solutions for a particular PDE instance and do not generalize to different inputs. Addressing these contemporary issues of learning PDEs, directly learning the solution operator of the PDE as a map between infinite-dimensional spaces has gained traction in recent years (Lu et al., 2019; Li et al., 2020a; 2021a; Gupta et al., 2021). These models extend the fundamental work of Chen and Chen (1995) that first showed a universal approximation theorem for learning operator networks. These models are explored in more detail in section 3.4.

## 3.2 Learning Models with Algebraic Structures

Rather than explicitly solving the differential equation, parameterizing the structure of the differential equation to guide the learning problem of inferring the differential equations from algebraic expressions is an emergent line of work. In particular, we first focus on learning the dynamical system equation as an ODE derived from Lagrangian or Hamiltonian constraints, which ensure energy conservation. The ODE is then only implicitly described with various neural networks encoding partial derivatives utilizing AD. This achieves sample-efficient model learning and efficient computation within a single forward pass and greatly enhances stability and generalization, benefiting many robotics control applications. Then, we shift the focus on a recent specific line of work parameterizing differential-algebraic equation, which models more general mathematical laws from observed data. We do not discuss the related but distinct task of using machine learning methods to discover symbolic models that explain observed measurements (Cranmer et al., 2020a).

**Parameterizing dynamical system equations.** Liu et al. (2005) was the first to introduce Lagrangian constraints into the optimization problem to enforce physics consistency for learning the equation of motion. Contrasting to the physic-informed neural network paradigm, they typically parameterize the general Euler-Lagrange equation from the Lagrangian $\mathcal{L}(\boldsymbol{q}, \dot{\boldsymbol{q}}) = \mathcal{T}(\boldsymbol{q}, \dot{\boldsymbol{q}}) - \mathcal{V}(\boldsymbol{q}) = \text{const}$, for each index $i$ of the coordinate $\boldsymbol{q}$,

$$\frac{\mathrm{d}}{\mathrm{d}\boldsymbol{q}_i} \left( \frac{\partial \mathcal{L}}{\partial \dot{\boldsymbol{q}}_i} \right) - \frac{\partial \mathcal{L}}{\partial \boldsymbol{q}_i} = \boldsymbol{\tau}_i, \tag{13}$$

where $\boldsymbol{\tau}$ is the generalized force. Liu et al. (2005) optimize directly the system trajectory and dynamic parameters, minimizing the violation of the constraint from equation (13). Much later on, by realizing the rigid-body kinetic energy $\mathcal{T}(\boldsymbol{q}, \dot{\boldsymbol{q}}) = \frac{1}{2}\dot{\boldsymbol{q}}^\intercal \boldsymbol{M}(\boldsymbol{q})\dot{\boldsymbol{q}}$ under gravitational potential field $\mathrm{d}\mathcal{V}/\mathrm{d}\boldsymbol{q} = \boldsymbol{g}(\boldsymbol{q})$, Lutter and Peters (2023) propose deep Lagrangian networks (DeLAN), parameterizing directly the inertia tensor $\boldsymbol{M_\theta}(\boldsymbol{q})$ with positivity constraint ensuring dynamical stability property, and the gravity force $\boldsymbol{g_\theta}(\boldsymbol{q})$ (Figure 2). Combining automatic differentiation with the state partial derivatives, the derived derivatives w.r.t. time and, thus, the violation of Lagrangian constraint are efficiently computed in a forward pass (see Figure 2), effectively facilitating robot model learning with sample efficiency. Cranmer et al. (2020b) then generalizes DeLAN to the Lagrangian Neural Network (LNN), which assumes a more general form of kinetic energy rather than rigid-body kinetics. The LNN works by directly computing the matrix inverse of the Hessian of a learned Lagrangian $\left( \frac{\partial^2 \mathcal{L}}{\partial \dot{\boldsymbol{q}}^\top \partial \dot{\boldsymbol{q}}} \right)$, and computing the implied acceleration

$$\ddot{\boldsymbol{q}} = \left( \frac{\partial^2 \mathcal{L}}{\partial \dot{\boldsymbol{q}}^\top \partial \dot{\boldsymbol{q}}} \right)^{-1} \left( \frac{\partial \mathcal{L}}{\partial \boldsymbol{q}} - \dot{\boldsymbol{q}} \frac{\partial^2 \mathcal{L}}{\partial \boldsymbol{q}^\top \partial \dot{\boldsymbol{q}}} \right), \tag{14}$$

where the inverse is typically approximated with a Moore-Penrose pseudoinverse. However, employing the Moore-Penrose pseudoinverse may cause numerical instability due to ill-conditioned Hessians and suffers from poor scalability in high dimensions (Golub and Van Loan, 2013). The training is then done via backpropagating through a loss between the true and predicted $\ddot{\boldsymbol{q}}$.

Concurrently to DeLAN, Hamiltonian Neural Networks (HNN) were developed (Greydanus et al., 2019), which enforce energy conservation via the constraint violation of the separable Hamiltonian $\mathcal{H}(\boldsymbol{q}, \boldsymbol{p}) = \mathcal{T}(\boldsymbol{q}, \boldsymbol{p}) + \mathcal{V}(\boldsymbol{q}) = \text{const}$. In particular, the violation of the Hamiltonian equation of motion on the phase space represents the training loss

$$\mathcal{L}_{\text{HNN}}(\boldsymbol{\theta}) := \left\| \frac{\partial \mathcal{H_\theta}}{\partial \boldsymbol{p}} - \frac{\partial \boldsymbol{q}}{\partial t} \right\| + \left\| \frac{\partial \mathcal{H_\theta}}{\partial \boldsymbol{q}} + \frac{\partial \boldsymbol{p}}{\partial t} \right\|, \tag{15}$$

where $\boldsymbol{q}$ is now the canonical coordinate, and $\boldsymbol{p}$ is the state momentum. HNNs typically parameterize the Hamiltonian $\mathcal{H_\theta}$ with a neural network and utilize the AD similar to DeLAN to compute the partial derivatives to minimize equation (15).

Toth et al. (2019) applies the same concept of HNN to discover conservative law in pixel observations, effectively learning the Hamiltonian flow mapping between the initial condition distribution to any time state distribution. In the presence of observation noise, the symplectic recurrent neural network (SRNN) (Chen

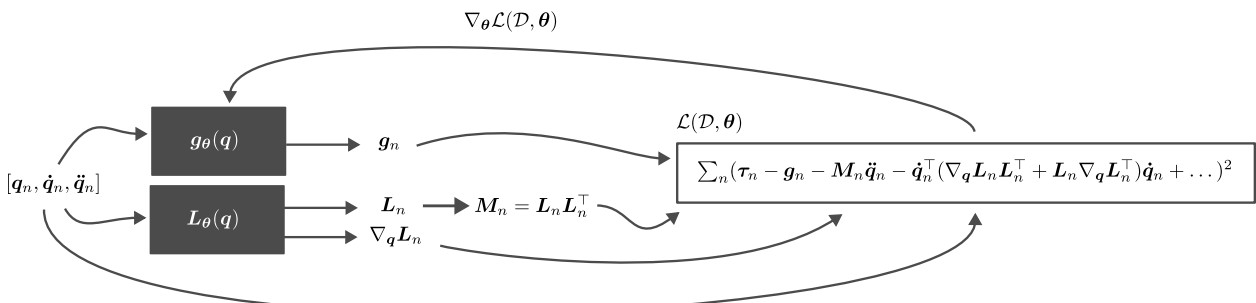

Figure 2: A schematic of deep Lagrangian networks for rigid body physics. Through careful use of automatic differentiation, the Lagrangian loss can be constructed from the state variables and rigid body terms, avoiding the need to differentiate the model with respect to time. The predicted inertial matrix $\boldsymbol{M}(\boldsymbol{q})$ is also guaranteed to be positive semi-definite through careful parameterization.

et al., 2019) harnesses the symplectic integrator (e.g., leapfrog integrator) to rollout multiple trajectories from initial states, then performs backpropagation-through-time to train the RNN $\mathcal{H}_{\boldsymbol{\theta}}$ w.r.t. the Hamiltonian loss (equation (15)). They found SRNN robustly learns the Hamiltonian dynamics under noisy state observations.

Jin et al. (2020) then generalize previous works for identifying both separable and non-separable Hamiltonian systems from data, approximating arbitrary symplectic maps based on appropriate activation functions. Notably, all mentioned works assume energy-conservative autonomous systems; however, many real-world dynamical systems depend explicitly on time and dissipate internal energy. Port-Hamiltonian networks (Desai et al., 2021; Neary and Topcu, 2023) was proposed to parameterize the phase-space ODE

$$\begin{bmatrix} \dot{\boldsymbol{q}} \\ \dot{\boldsymbol{p}} \end{bmatrix} = \left( \begin{bmatrix} \boldsymbol{0} & \boldsymbol{I} \\ -\boldsymbol{I} & \boldsymbol{0} \end{bmatrix} + \boldsymbol{D}(\boldsymbol{q}) \right) \begin{bmatrix} \frac{\partial \mathcal{H}}{\partial \boldsymbol{q}} \\ \frac{\partial \mathcal{H}}{\partial \boldsymbol{p}} \end{bmatrix} + \begin{bmatrix} \boldsymbol{0} \\ \boldsymbol{G}(\boldsymbol{q}) \end{bmatrix} \boldsymbol{u}, \tag{16}$$

with neural networks on the damping $\boldsymbol{D}$ matrix, the Hamiltonian $\mathcal{H}$, the generalized force $\boldsymbol{G}(\boldsymbol{q})\boldsymbol{u}$, effectively recovering the underlying stationary Hamiltonian, time-dependent force, and dissipative coefficient. Recently, Roth et al. (2025) introduced the concepts of Lyapunov and global stability of dynamic systems into Port-Hamiltonian NNs by enforcing the Hamiltonian to be convex and possess a strict minimum. This additional physical bias and mathematical structure greatly improve robustness of training and predictions as instabilities are avoided, enhance extrapolation capabilities, and reduce model variance.

**Parameterizing differential algebraic equations.** The aforementioned methods explicitly evoke the equation of motion derived from Lagrangian/Hamiltonian frameworks in the forward. However, discovering general symbolic rules from data is challenging. At the time of writing, an emergent line of work under the name of algebraically-informed neural network (AINN) (Hajij et al., 2020) parameterizes algebraic structures, modeling a finite number of algebraic objects with a given set of neural networks $\{f_{\boldsymbol{\theta}_i} : \mathbb{R}^{n_i} \to \mathbb{R}^{m_i}\}_{i=1}^N$.

In essence, the algebraic operators of neural networks can be defined straightforwardly given certain conditions. Denoting $\mathcal{N}(\mathbb{R}^n)$ as the set of networks $f_{\boldsymbol{\theta}} : \mathbb{R}^n \to \mathbb{R}^n$, the algebraic operators are defined for

$$
\begin{aligned}
\text{concatenation :} \quad & (f_{\boldsymbol{\theta}_1} \times f_{\boldsymbol{\theta}_2})(\boldsymbol{x}, \boldsymbol{y}) = (f_{\boldsymbol{\theta}_1}(\boldsymbol{x}), f_{\boldsymbol{\theta}_2}(\boldsymbol{y})), \quad f_{\boldsymbol{\theta}_1} \in \mathcal{N}(\mathbb{R}^{n_1}), \quad f_{\boldsymbol{\theta}_2} \in \mathcal{N}(\mathbb{R}^{n_2}), \\
\text{addition :} \quad & (f_{\boldsymbol{\theta}_1} + f_{\boldsymbol{\theta}_2})(\boldsymbol{x}) = (f_{\boldsymbol{\theta}_1}(\boldsymbol{x}) + f_{\boldsymbol{\theta}_2}(\boldsymbol{x})), \quad \text{if} \quad f_{\boldsymbol{\theta}_1}, f_{\boldsymbol{\theta}_2} \in \mathcal{N}(\mathbb{R}^n), \\
\text{scalar multiplication :} \quad & (a * f_{\boldsymbol{\theta}_1})(\boldsymbol{x}) = a * f_{\boldsymbol{\theta}_1}(\boldsymbol{x}), \\
\text{Lie brackets :} \quad & [f_{\boldsymbol{\theta}_1}, f_{\boldsymbol{\theta}_2}] = f_{\boldsymbol{\theta}_1} \circ f_{\boldsymbol{\theta}_2} - f_{\boldsymbol{\theta}_2} \circ f_{\boldsymbol{\theta}_1},
\end{aligned}
$$

where $\circ$ is the composition operator, assuming $\mathcal{N}(\mathbb{R}^n)$ is closed under the compositions. Typically, given a set of generators $\{s_i\}_{i=1}^N$ and algebraic relations $\{r_i\}_{i=1}^K$, a neural network is defined for each generator, and the loss can be minimized with standard stochastic gradient descent (Bottou, 2012), satisfying all algebraic relations. For example, one can learn the Yang-Baxter equation (Hajij et al., 2020), given $R : A \times A \to A \times A$

and some set $A$,

$$(R \times \mathrm{id}_A) \circ (\mathrm{id}_A \times R) \circ (R \times \mathrm{id}_A) = (\mathrm{id}_A \times R) \circ (R \times \mathrm{id}_A) \circ (\mathrm{id}_A \times R) \tag{17}$$

by defining a neural network $f_{\boldsymbol{\theta}} := R$ and minimizing the algebraic relation as the violation of equation (17). Recently, Moya and Lin (2023) parametrize a class of differential-algebraic equations

$$\begin{aligned}\dot{\boldsymbol{q}} &= f(\boldsymbol{q}, \boldsymbol{z}), & \boldsymbol{q}\,(t_0) &= \boldsymbol{q}_0, \\ 0 &= g(\boldsymbol{q}, \boldsymbol{z}), & \boldsymbol{z}\,(t_0) &= \boldsymbol{z}_0,\end{aligned} \tag{18}$$

where $\boldsymbol{q}(t) \in \mathbb{R}^d$ are the dynamic states and $\boldsymbol{z}(t) \in \mathbb{R}^n$ are the algebraic variables. This class of differential-algebraic equations has been used to model various engineering applications, such as power network systems with frequency-dependent dynamic loads or nonlinear oscillations. Under the AINN framework, Moya and Lin (2023) model $f, g$ as neural networks and treat both statements in equation (18) as algebraic relations. Given training data on $\boldsymbol{q}$, they perform a model rollout with an implicit Runge-Kutta method (Iserles, 2009) from the initial conditions, then learn $f, g$ by matching the rollout trajectories against the training data and minimizing the violation of the constraint $g(\boldsymbol{q}, \boldsymbol{z})$, satisfying the defined algebraic relations.

AINN is still in its infancy state with very few works. However, we foresee vast applications of parameterizing algebraic rules with neural networks in many hybrid systems (i.e., discrete-continuous) in robotics or engineering applications.

### 3.3 Learning Simulation Solutions with Physics-Informed Losses

Physics-informed neural networks (Raissi et al., 2017a;b) are a mesh-free approach for solving PDEs that uses a NN as a function approximator for the solution $\boldsymbol{u} \approx \boldsymbol{u}_{\boldsymbol{\theta}}$. Therefore, the PDE (equation (3)) is transformed into an unconstrained optimization objective by penalizing violations of the differential equation and boundary conditions through and squared penalties:

$$\mathcal{L}_{\mathrm{PINN}}(\boldsymbol{\theta}) = c_f \, \mathcal{L}_f(\boldsymbol{\theta}) + c_b \, \mathcal{L}_b(\boldsymbol{\theta}), \; \mathcal{L}_f(\boldsymbol{\theta}) = \int_{\Omega} |\mathsf{L}_{\boldsymbol{a}} \boldsymbol{u}_{\boldsymbol{\theta}}(\boldsymbol{x}) - \boldsymbol{f}(\boldsymbol{x})|^2 \, \mathrm{d}\boldsymbol{x}, \; \mathcal{L}_b(\boldsymbol{\theta}) = \int_{\partial\mathcal{X}} |\boldsymbol{u}_{\boldsymbol{\theta}}(\boldsymbol{x}) - \boldsymbol{b}(\boldsymbol{x})|^2 \, \mathrm{d}\boldsymbol{x}, \quad (19)$$

where $c_f > 0$ and $c_b > 0$ are weights in the loss function. $\mathcal{L}_f$ is referred to as the physics- or residual loss and $\mathcal{L}_b$ as the boundary loss. Note that these two objectives consider different quantities, so careful tuning of the weights is required to balance the multi-objective nature of the problem. Using these loss functions, learning PINNs can be considered unsupervised, as no data is required. However, physical knowledge is embedded in the loss definition through the specification of the PDE operator $\mathsf{L}_{\boldsymbol{a}}$ in the residual term and provides a supervision signal during training. In section 3.6, we also discuss an extended PINN objective that includes a data loss term in addition, similar to the boundary condition loss. If $\mathcal{L}_{\mathrm{PINN}}(\boldsymbol{\theta}_*) = 0$, then $\boldsymbol{u}_{\boldsymbol{\theta}_*}$ satisfies the weak form of the PDE boundary value problem . This approach of 'residual minimization' with function approximators can be traced back to Dissanayake and Phan-Thien (1994) and Lagaris et al. (1998), but has grown in popularity recently due to the maturity of deep learning methods.

In practice, the losses in equation (19) must be computed using finite sums rather than exact integrals, this means that collocation or numerical integration must be applied. Moreover, this approach takes advantage of automatic differentiation (AD) (Rall, 1981), which is already used in training neural networks (Goodfellow et al., 2016). During training, *reverse*-mode AD is used to efficiently compute parameter gradients from the evaluated one-dimensional objective to the high-dimensional parameter space, using a combination of dynamic programming and chain rule. Reverse- or forward-mode AD can be used to compute $\mathsf{L}_{\boldsymbol{a}} \boldsymbol{u}_{\boldsymbol{\theta}}(\boldsymbol{x})$. The choice depends on whether the output or input is higher dimensional, respectively, as the complexity of the AD is dictated by this[1]. Forward mode is particularly attractive as it enables derivatives to be computed during the forward pass. Note that second-order derivatives require a sequence of forward- and reverse-mode AD.

---

[1] Reverse-mode is preferred for deep learning as objectives always have 1D outputs.

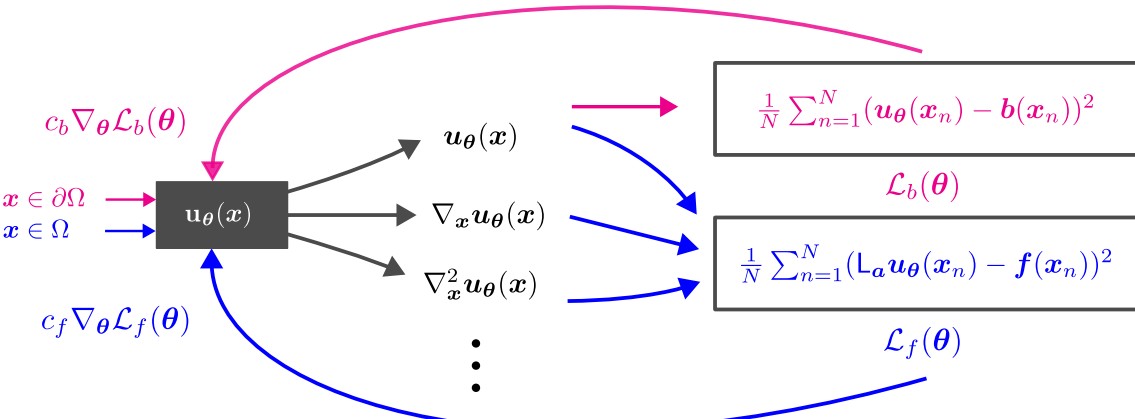

Figure 3: A schematic of physics-informed neural networks, detailing the relationship between the approximated model of the solution $\boldsymbol{u_\theta}(\boldsymbol{x})$ and the physics-informed loss terms, where the integral over the domain has been replaced with a Monte Carlo approximation. Using forward-mode automatic differentiation, the required gradients of the solution are used to minimize the residual PDE error term (blue), while the boundary condition here only uses the solution values (magenta). However, model gradients may be required for the boundary loss, depending on how the boundary condition is defined.

Variational PINNs (Kharazmi et al., 2019) optimize the weak form rather than the strong form in the Petrov-Galerkin setting. The loss function becomes

$$\mathcal{L}_{\boldsymbol{f}}(\boldsymbol{\theta}) = \sum_{m=1}^{M} \left\| \left| \int_{\Omega} \mathsf{L}_{\boldsymbol{a}} \boldsymbol{u_\theta}(\boldsymbol{x})\, \boldsymbol{v}_m(\boldsymbol{x}) \mathrm{d}\boldsymbol{x} - \int_{\Omega} \boldsymbol{f}(\boldsymbol{x})\, \boldsymbol{v}_m(\boldsymbol{x})\, \mathrm{d}\boldsymbol{x} \right\| \right\|^2, \tag{20}$$

where $\boldsymbol{v}_m$ are test functions and $M$ is the number of sampling points. This approach is attractive because it can lower the order of differential operators that the neural network approximates, making the computation of loss functions cheaper. However, the analytic tractability puts severe restrictions on the models and settings that can be considered, and the authors only demonstrate the approach on a one- and two-dimensional case of Poisson's equation.

Separable PINNs (Cho et al., 2023a) propose a different architecture in order for forward-mode AD to be more efficient. Each network takes an element of $\boldsymbol{x}$ as input and has $Q$ outputs. The final prediction is a product over input elements and a summation over network outputs, i.e.,

$$u_{\boldsymbol{\theta}}(\boldsymbol{x}) = \sum_{q=1}^{Q} \prod_{i=1}^{d} f_q(x_i). \tag{21}$$

Since each network has a one-dimensional input, forward-mode AD is efficient. The structure also means that evaluating a $d$-dimensional mesh of $N$ points requires $Nd$ network evaluations rather than $N^d$. Since the sum-product mixing function is relatively cheap in comparison to network evaluation, this architecture scales very gracefully as $d$ gets large in comparison to prior PINN approaches.

The practical challenges of applying PINNs in dynamical systems lie in approximating and optimizing the loss function. Due to the nature of differential equations, the residual loss is often not 'fixed' between time steps, as it requires the model to be self-consistent with a set of its own derivatives (Wang et al., 2021a). This objective means the regression target is not fixed, and the model is regressed against itself during optimization until self-consistency is achieved. This 'bootstrapping' of a function approximator can lead to unstable optimization in practice[2]. As a result, sampling and optimization must be designed carefully and jointly to ensure successful optimization.

---

[2]Bootstrapping and function approximation comprises of two parts of the 'deadly triad' in reinforcement learning (Sutton and Barto, 2018) where this model bootstrapping is also performed when computing value functions

**Sampling.** Approximating an integral across a domain of interest requires extensive care or computation in order to achieve good accuracy. Classical interpolation-based numerical quadrature methods (e.g., Gauss quadrature) typically prescribe weighted lattice-like evaluation points, which would make PINNs similar in implementation and memory complexity to mesh-based solvers. As an alternative, Monte Carlo approximations are attractive as they provide the flexibility of evaluating points in a non-lattice structure. The reduction in memory requirements due to minibatching can be used in conjunction with stochastic optimization methods such as stochastic gradient descent, widely utilized in deep learning. However, Monte Carlo's flexibility results in greater variance in the objective estimation, which impedes the convergence of the stochastic optimization. To mitigate this issue, quasi-Monte Carlo (Morokoff and Caflisch, 1995) methods introduce low-discrepancy pseudo-random sequences that significantly reduce variance. Examples include Sobol and Halton sequences, which have shown promise in PINN optimization (Pang et al., 2019).

Nabian et al. (2021) propose an optimization-orientated 'importance sampling' scheme. For a given set of points, they sample a minibatch according to the categorical distribution proportional to the loss at each point. This prioritization is shown to accelerate convergence. However, this is not a valid importance sampling scheme w.r.t. Monte Carlo methods but is rather an 'active' sampling scheme that incorporates decision-making into the learning process, as done in active learning (Cohn et al., 1996).

Daw et al. (2023) proposes 'retain-resample-release' sampling, which carefully maintains a population of collocation points, where points with high residual loss are maintained, and the low loss points are resampled each iteration. They also present a biased sampling procedure that encourages points that match the predicted temporal evolution of the system. They also anneal the domain's horizon during optimization to mitigate the propagation of large model errors over time.

Subramanian et al. (2023) propose an adaptive self-supervision algorithm to enhance the performance and 'trainability' of PINNs. By reallocating collocation points to regions with higher errors and periodically resampling them, the proposed methods effectively reduce prediction errors and address optimization challenges across various problems, without increasing computational costs.

**Optimization.** Wang et al. (2021b) investigate the training dynamics of PINNs and observed that the magnitude of the parameter gradient increased during training, in part due to the multi-objective loss function. To achieve stable convergence, the authors advocated for a learning rate schedule.

Fonseca et al. (2023) studied the optimization landscape of PINNs, and found that their solutions lie in sharp minima, i.e., regions of the loss landscape where the loss varies significantly under small perturbations of the solution. This is in contrast to folklore in deep learning, where solutions in flat minima are preferred as they are interpreted as solutions that are not overfitting and, therefore, generalizing. Since PINNs models are trained in their domain, the issue of out-of-distribution generalization is less of a concern. This difference results in typical deep learning optimization techniques being less desirable. The authors found that quasi-Newton solvers such as the limited-memory Broyden–Fletcher–Goldfarb–Shanno algorithm (LBFGS) were superior optimizers due to their strength in reaching sharp minima by incorporate loss curvature.

Krishnapriyan et al. (2021) highlight the bootstrapping issue by isolating the residual loss term as the most difficult w.r.t. the combined PINN loss landscape and show that optimization can be improved by gradually annealing this term to be more significant during optimization. They also advocate modeling the solution autoregressively, i.e., $\boldsymbol{u}_{t+\Delta t} = \boldsymbol{f_\theta}(\boldsymbol{u}_t, \Delta t)$, so the prediction can be bootstrapped from $\boldsymbol{u}_0$, and the function approximate does not need to fit the solution for the entire time horizon.

Another approach to mitigating the bootstrapping problem is ensembling models. Haitsiukevich and Ilin (2023) trained an ensemble of PINN models and averaged their predictions in the residual loss term. By averaging over predictions, the errors from bootstrapping were reduced.

The temporal aspect of the bootstrapping issue has also been mitigated by introducing a time-based weighting scheme to the residual loss (Wang et al., 2022a), which attenuates the residual loss at a point if it demonstrates a relatively high loss when integrated from the boundary condition, i.e.,

$$\mathcal{L}_{f,\epsilon,N}(\boldsymbol{\theta}) = \sum_{i=1}^{N} \exp\left(-\epsilon \sum_{k=1}^{i-1} \mathcal{L}_f(t_k, \boldsymbol{\theta})\right) \mathcal{L}_f(t_i, \boldsymbol{\theta}), \tag{22}$$

where $\epsilon > 0$ controls the attenuation. Note that this loss is relevant to temporal PDE cases with known initial conditions.

Choosing the weights of each loss term is another practical challenge since different loss terms converge at different rates. Therefore, it is difficult to identify the optimal relative weights for each term, which has a significant impact on the training stability of the PINNs and predictions as well (Faroughi et al. (2023)). Bischof and Kraus (2021) proposed a self-adaptive loss balancing scheme named ReLoBRaLo (relative loss balancing with random lookback). This approach calculates the progress of each loss term by comparing it with the previous iteration values. Using 'random lookback,' the method can decide whether to consider the progress from the last iteration or the progress from the start of the training process. This approach enables multiple loss terms to be optimized simultaneously in an efficient way. Xiang et al. (2022) proposed a probabilistic interpretation to define the self-adaptive loss function through the adaptive weights for each loss term. The weights are updated in each epoch automatically based on maximum likelihood estimation.

PINNs have difficulty in learning search systems due to spectral bias, which makes them learn low-frequency patterns in the data (Mustajab et al., 2024). Furthermore, even when trained on several initial and boundary conditions, they still suffer when new extrapolated initial and boundary conditions are there (Nakamura et al., 2021).

If these implementation issues discussed above can be overcome, researchers can fully unlock the practical value of PINNs as an alternative to mesh-based solvers. They provide a less accurate but computationally cheaper solution that may be desirable in certain applications. The generality of the approach has led to extensive interest from the scientific community (Cuomo et al., 2022) due to the numerous application domains, such as fluid mechanics (Cai et al., 2021a), heat transfer (Cai et al., 2021b), power systems (Huang and Wang, 2022). There is an open question regarding how and when PINNs are a viable replacement to standard PDE solvers. Grossmann et al. (2023) benchmarked PINNs and FEM methods for several PDE test cases and found that, on the whole, the FEM method was faster (around x10) and more accurate (around 100x). The benefit of PINNs is that they are sometimes faster to evaluate and may potentially scale more gracefully to higher-dimensional problems. However, given their poorer accuracy on lower-dimensional problems, they are not guaranteed to provide high accuracy in this setting.

### 3.4 Operator Learning

So far, this survey discussed learning the solution $u_{\theta}(\boldsymbol{x})$ to a differential equation for a specific configuration $\boldsymbol{a}$, (possible) boundary conditions (equation (4)), and source terms $\boldsymbol{f}$. However, this approach requires solving the system anew for each new configuration. A more efficient alternative is to learn an operator — a model that maps an infinite-dimensional function-space input to the corresponding solution. This enables generalization across different configurations without solving the differential equation from scratch. Specifically, we introduce learning a solution operator $\mathsf{L}_{\boldsymbol{a}}^{-1}\boldsymbol{f}(\boldsymbol{x}) = \boldsymbol{u_a}(\boldsymbol{x})$ that maps from state-varying functions $\boldsymbol{a}(\boldsymbol{x}) : \Omega \to \mathbb{R}^{d_a}$ to the corresponding solution (Figure 4). Operating on functions introduces the same flexible discretization invariance discussed in section 3.1, but now extended beyond time to any continuous state space. Moreover, the training data for such models now extends across several instances of the physical system, so the model is extended in a physics-informed fashion to be conditioned on the system parameters. We will revisit multi-instance learning in section 4 in the context of learning data-driven priors.

While the parameterized solution operator approximates a mapping between two function spaces $\mathsf{S}_{\boldsymbol{\theta}} : \mathcal{A} \to \mathcal{U}$, its implementation is a point-to-point mapping evaluating the solution $\boldsymbol{u}$ for a given input function $\boldsymbol{a}$ at point $\boldsymbol{x}$. The training data is commonly available from a discretization of the domain $\mathcal{D}_i = \{\boldsymbol{x}_i \in \Omega\}_{i=1}^{N}$ for a specific input $\boldsymbol{a}_i \sim \mu(\boldsymbol{a})$. Information about the functions $\boldsymbol{a}$ and $\boldsymbol{u}$ are typically only available at point-wise evaluations. Thus, the loss is the relative $L^2$ loss for $K$ different inputs $\boldsymbol{a}_i$ at the discretization points $\boldsymbol{x}_j \in \mathcal{D}_i$,

$$\mathcal{L}_{\text{NO}}(\boldsymbol{\theta}) = \sum_{i=1}^{K} \sum_{\boldsymbol{x}_j \in \mathcal{D}_i} \frac{||\mathsf{S}\boldsymbol{a}_i(\boldsymbol{x}_j) - \mathsf{S}_{\boldsymbol{\theta}}\boldsymbol{a}_i(\boldsymbol{x}_j)||^2}{||\mathsf{S}\boldsymbol{a}_i(\boldsymbol{x}_j)||^2}. \tag{23}$$

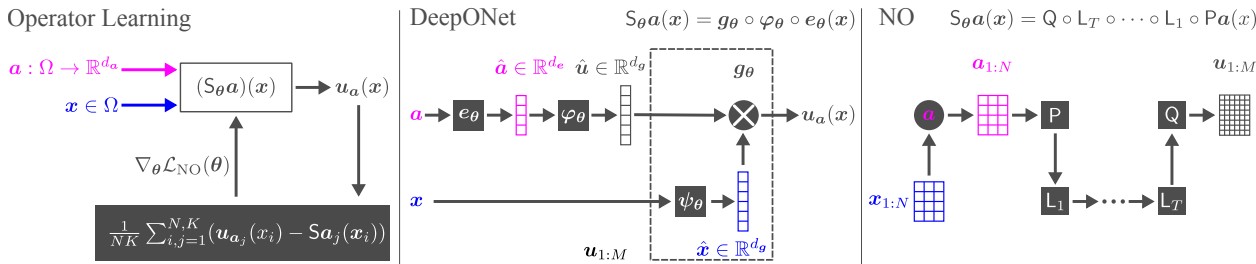

Figure 4: (Left) A schematic of operator networks that learn a functional mapping from the inputs $a$ to the PDE solution $\boldsymbol{u}$ at a designated point $\boldsymbol{x}$. (Middle) DeepONets can approximate any nonlinear operator by learning a finite-dimensional mapping $\hat{\boldsymbol{u}} = \boldsymbol{\varphi_\theta}(\hat{\boldsymbol{a}})$ in a latent space which is spanned by linear encodings $\hat{\boldsymbol{a}} = \boldsymbol{e_\theta}(\boldsymbol{a})$ and linear decoding $\boldsymbol{u} = \boldsymbol{g_\theta}(\hat{\boldsymbol{u}}, \boldsymbol{x})$. (Right) Neural operators directly map a discretized representation of the input $\boldsymbol{a}_{1:N}$ to the solution at an arbitrary discretization $\boldsymbol{u}_{1:M}$.

Kovachki et al. (2023) report that training with the relative $L^2$ loss equation (23) is less prone to overfitting due to normalization compared to the alternative mean squared error (MSE) loss. The specific parameterization of the operator networks has been the primary focus of recent approaches to guarantee discretization invariance, which has a bounded generalization error while being computationally efficient. Following this, we will discuss the two prevailing operator learning architectures, DeepONets (Lu et al., 2019) and neural operators (Li et al., 2020a) highlighted in Figure 4. Although both architectures follow universal approximation theorems to learn arbitrary operators, their conceptual design is fundamentally different.

**Operator learning with finite-dimensional mappings.** In order to deal with infinite-dimensional input and output spaces, deep operator networks (DeepONets) (Lu et al., 2019) are based on the idea of learning the solution operator in a finite-dimensional space $\boldsymbol{\varphi} : \mathbb{R}^{d_e} \to \mathbb{R}^{d_g}$. This requires a functional mapping of the input functions $\boldsymbol{a}$ into a latent representation $\boldsymbol{e} : \mathcal{A} \to \mathbb{R}^{d_e}$ and a respective functional mapping $\boldsymbol{g} : \mathbb{R}^{d_g} \to \mathcal{U}$ that decodes. Thus, the operator learning architecture based on the finite-dimensional mappings can be formulated as $\mathsf{S_\theta a} = \boldsymbol{g_\theta} \circ \boldsymbol{\varphi_\theta} \circ \boldsymbol{e_\theta}$. For this architecture, Chen and Chen (1995) first showed that for linear maps $\boldsymbol{e_\theta}$ and $\boldsymbol{g_\theta}$, and a single-layer network $\boldsymbol{\varphi_\theta}$, there exist finite-dimensional latent spaces $\mathbb{R}^{d_e}$ and $\mathbb{R}^{d_g}$ for which any nonlinear operator can be approximated. This universal approximation theorem for operator learning has recently been expanded to encompass arbitrary deep-learning architectures for mapping in finite-dimensional spaces (Lu et al., 2019). The different embodiments of the three modules lead to distinct approaches, all underlying the same concept of finite-dimensional mappings.

DeepONets (Lu et al., 2019) encode the input function by evaluating $\boldsymbol{a}$ on so-called sensory points $\{\boldsymbol{x}_s^{(i)}\}_{i=1}^{d_e}$ as $\boldsymbol{e} : \boldsymbol{a} \mapsto [\boldsymbol{a}(\boldsymbol{x}_s^{(0)}), \dots, \boldsymbol{a}(\boldsymbol{x}_s^{(d_e)})]^\intercal$. The encoded input representation is processed by any deep neural architecture $\boldsymbol{\varphi_\theta}$. Finally, the decoder is defined as the scalar product between the output of the finite-dimensional mapping and a latent encoding of the domain $\boldsymbol{g_\theta}(\boldsymbol{x}) = \sum_{i=1}^{d_g} \varphi_\theta^{(i)} \psi_\theta^{(i)}(\boldsymbol{x})$. Thus, the decoder approximates the solution operator $\mathsf{S}\boldsymbol{a}(\boldsymbol{x}) \approx \boldsymbol{g_\theta}(\boldsymbol{x})$, where $\boldsymbol{\varphi_\theta}$ and $\boldsymbol{\psi_\theta}$ are parameterized by neural networks. In the literature, the finite-dimensional mapping $\boldsymbol{\varphi_\theta}$ is commonly referred to as the *branch net* while $\boldsymbol{\psi_\theta}$ denotes the *trunk net*. The scalar product between the branch and trunk network yields a scalar output, yet, the DeepONets architecture can be easily extended to vector-valued outputs by adding channels to the branch and trunk nets (Lu et al., 2022). Wang et al. (2022b) propose an improved DeepONet architecture with empirical advantages over the vanilla DeepONet architecture. In particular, the authors argue that the fusion of the encodings of the input parameters $\boldsymbol{a}$ and the domain representation $\boldsymbol{x}$ only happens in the final scalar product step. They proposed intertwining the two input sources earlier. While empirically, the proposed architecture improved the approximation results, the architecture lacks the common universal approximation theorem that related models have shown.

Bhattacharya et al. (2021) propose PCA-based encoding and decoding schemes for the input and output spaces. The encoder projects the Hilbert space $\mathcal{U}$ onto the finite-dimensional sub-space $\boldsymbol{e} : \boldsymbol{a} \mapsto [\langle \boldsymbol{a}, \boldsymbol{v}_e^{(1)} \rangle, \dots, \langle \boldsymbol{a}, \boldsymbol{v}_e^{(d_e)} \rangle]^\intercal$ spanned by $d_e$ basis vectors $\boldsymbol{v}_e^{(i)}$. Given evaluations of $\boldsymbol{a}$ at $N$ observed

data points $\boldsymbol{x}$, $\boldsymbol{v}_i$ correspond to the $d_e$ eigenvectors of the empirical covariance matrix $\boldsymbol{C} = N^{-1} \sum_{i=1}^{N} \boldsymbol{a}_i \otimes \boldsymbol{a}_i$ with the largest eigenvalues. Approximate reconstruction can be obtained through $\boldsymbol{a} \approx \sum_{i=1}^{d_a} \langle \boldsymbol{u}, \boldsymbol{v}_{\boldsymbol{a}}^{(i)} \rangle \boldsymbol{v}_{\boldsymbol{a}}^{(i)}$. By also projecting the solution space $\mathcal{U}$ into a finite-dimensional space using PCA, the decoder can be formulated as $\boldsymbol{g}(\boldsymbol{x}) = \sum_{j=1}^{d_g} \boldsymbol{\varphi}_{\boldsymbol{\theta}}(\boldsymbol{e}(\boldsymbol{a})) \boldsymbol{v}_{\boldsymbol{g}}^{(j)}$ with $\boldsymbol{v}_{\boldsymbol{g}}^{(j)}$ being the eigenvectors of the covariance matrix of $\boldsymbol{u}$.

Similar ideas within the framework of reduced-order methods have been presented by Peherstorfer and Willcox (2016). This method, known as operator inference, involves projecting the state-space onto a lower-dimensional representation using proper orthogonal decomposition (POD) and subsequently learning the linear operators of systems of nonlinear ordinary differential equations (ODEs). However, these models often lack the generalization capabilities seen in more recent methodologies, such as neural operators.

**Neural operators.** As noted by Kovachki et al. (2024), the previous methods rely on a finite-dimensional representation of $\mathcal{A}$ and $\mathcal{U}$, which might be restrictive as they describe mappings in the linear subset of $\mathcal{A}$ and $\mathcal{U}$. The neural operator (NO) architectures presented in this section do not rely on finite-dimensional mapping and, thus, are more principled. Neural operators are motivated by the theory of inhomogeneous linear differential equations. If $\mathsf{L}_{\boldsymbol{a}}$ in equation (3) constitutes a linear operator, the solution of $\boldsymbol{u}$ is characterized by Green's function

$$\boldsymbol{u}(\boldsymbol{x}) = \int_{\Omega} G(\boldsymbol{x}, \boldsymbol{y}) \boldsymbol{f}(\boldsymbol{y}) \, \mathrm{d}\boldsymbol{y} + \boldsymbol{u}_{\text{homo}}(\boldsymbol{x}). \tag{24}$$

Here, $G(\boldsymbol{x}, \boldsymbol{y})$ is the response impulse of the linear operator $\mathsf{L}_{\boldsymbol{a}} G(\boldsymbol{x}, \boldsymbol{y}) = \delta(\boldsymbol{x} - \boldsymbol{y})$ and $\boldsymbol{u}_{\text{homo}}(\boldsymbol{x})$ is the homogeneous solution which only takes values on the domain boundary $\partial\Omega$. Generally, Green's function $G(\boldsymbol{x}, \boldsymbol{y})$ is not trivially available. Therefore, several works propose approximating Green's function with a neural network $G_{\boldsymbol{\theta}}(\boldsymbol{x}, \boldsymbol{y})$ (Boullé et al., 2022; Zhang et al., 2022a).

In numerous relevant applications, the PDE operator exhibits nonlinearity, rendering equation (24) inapplicable. Although approaches utilizing encoding and decoding schemes to transform the problem into sub-spaces amenable to linear operators have been suggested (Gin et al., 2021), they lack the theoretical assurances inherent in operator learning methods. Yet, the idea of using integral kernel operators play a pivotal role in designing neural operators with theoretical guarantees

$$\mathsf{K}\boldsymbol{v}(\boldsymbol{x}) = \int \kappa(\boldsymbol{x}, \boldsymbol{y}) \boldsymbol{v}(\boldsymbol{y}) \, \mathrm{d}\boldsymbol{y}, \tag{25}$$

where $\kappa(\boldsymbol{x}, \boldsymbol{y}) : \mathcal{D} \times \mathcal{D}' \to \mathbb{R}^{d_{v'}} \times \mathbb{R}^{d_v}$ is a kernel function, and $\boldsymbol{v}(x) : \mathcal{D} \to \mathbb{R}^{d_v}$ is a vector valued function. Each integral kernel operator maps $\boldsymbol{v}(x)$ to $\mathsf{K}\boldsymbol{v}(\boldsymbol{x}) : \mathcal{D}' \to \mathbb{R}^{d_{v'}}$. Here, $\mathcal{D}$ and $\mathcal{D}'$ refer to a representation of the domain. The neural operator architecture is split into three sub-components: a pointwise lifting operator $\mathsf{P}$, a sequence of kernel integrations layers $\mathsf{L}_t$; $t = 1, \ldots, T$, and a pointwise projection operator $\mathsf{Q}$. The lifting operator maps the input function to a hidden projection $\boldsymbol{v}_0(\boldsymbol{x}) = \mathsf{P}\boldsymbol{a}(x)$. The series of kernel integration layers processes the lifted representation $\boldsymbol{v}_t(\boldsymbol{x}) = \mathsf{L}_t \boldsymbol{v}_{t-1}(\boldsymbol{x})$, and is finally projected to the solution $\boldsymbol{u}(\boldsymbol{x}) = \mathsf{Q}\boldsymbol{v}_T(x)$. Here, a pointwise operator means that the operator is applied on each action separately, i.e., $\mathsf{P}\boldsymbol{a}(x) = \mathsf{P}(\boldsymbol{a}(x))$. The full architecture is a composition of the previously introduced operators,

$$(\mathsf{S}_{\boldsymbol{\theta}}\boldsymbol{a})(\boldsymbol{x}) = (\mathsf{Q} \circ \mathsf{L}_T \circ \cdots \circ \mathsf{L}_1 \circ \mathsf{P}\boldsymbol{a})(\boldsymbol{x}), \tag{26}$$

$$\mathsf{L}_t \boldsymbol{v}_{t-1}(\boldsymbol{x}) = \sigma \left( \boldsymbol{W}_t \boldsymbol{v}_{t-1}(\boldsymbol{x}) + \mathsf{K}_t \boldsymbol{v}_{t-1}(\boldsymbol{x}) + \boldsymbol{b}_t \right). \tag{27}$$

The kernel integration layers are composed of a linear operator, represented by the matrix $\boldsymbol{W}_t \in \mathbb{R}^{d_{v_t} \times d_{v_{t-1}}}$, a bias term $\boldsymbol{b}_t \in \mathbb{R}^{d_{v_t}}$, and the integral kernel operator (equation (25)). By parameterizing $\mathsf{P}, \mathsf{Q}, \boldsymbol{W}_t, \boldsymbol{b}_t, \kappa_t$, the neural operator can be shown to approximate arbitrary nonlinear operators (Kovachki et al., 2023). Although NOs are not restricted to finite-dimensional mappings, as noted by (Kovachki et al., 2024), in practice, NOs work on discretized representations of the input domain $\boldsymbol{a}_{1:N}$. The main reason is that the integral kernel operator (equation (25)) remains a computationally demanding operation. Therefore, we will examine different architectures to approximate $\mathsf{K}$ efficiently.

**Parameterization of the Kernel Integrations.** Numerical quadrature of the integral in equation (25) requires matrix-vector multiplications on the order of $\mathcal{O}(N^2)$ operations, where $N$ is the number of uniformly sampled discretization points $\boldsymbol{x}_i$, $\boldsymbol{y}_j$ in the domains $\mathcal{D}_t$, $\mathcal{D}_{t-1}$. Thus, recent approaches leverage the theoretical findings of kernel methods and spectral methods to construct efficient neural approximations of the integral kernel operator K.

Li et al. (2020a) propose graph neural operators (GNO), which are based on message-passing graph neural networks (Bronstein et al., 2021) to represent the neighborhood of a data-point $\boldsymbol{x}$. Based on the $J$ nearest neighbors of the data point, the kernel integral can be efficiently computed using Nyström approximation and truncation. With the assumption that $J \ll N$, the computational efficiency can be significantly increased, even though it still requires matrix-vector multiplications in the order of $\mathcal{O}(J^2)$. Multipole GNOs (Li et al., 2020b) extend GNOs with the fast multipole method that enables a hierarchical graph structure that is capable of capturing global phenomena.

Instead of evaluating the kernel operator in the domain space, Fourier neural operators (FNOs) (Li et al., 2021a) leverage Fourier transformations to evaluate the integral kernel operator in its spectral representation. Assuming $\kappa(\boldsymbol{x}, \boldsymbol{y}) = \kappa(\boldsymbol{x} - \boldsymbol{y})$ and applying the convolution theorem, the integral kernel operator K can be represented as

$$\mathsf{F}^{-1}(\mathsf{F}(\kappa_t) \cdot \mathsf{F}(\boldsymbol{v}_{t-1}))(\boldsymbol{x}) = \int \kappa_t(\boldsymbol{x} - \boldsymbol{y})\boldsymbol{v}_{t-1}(\boldsymbol{y})\mathrm{d}\boldsymbol{y} = \mathsf{K}_t, \tag{28}$$

where $\mathsf{F}$ and $\mathsf{F}^{-1}$ denote the Fourier transformation and inverse Fourier transformation, respectively Like GNOs, a discretized domain representation facilitates an efficient evaluation of equation (28) by using fast Fourier transformations (FFTs). Given an equidistantly spaced discretization of the domain $\{\boldsymbol{x}_k\}_{k=1}^N$, the Fourier coefficients of the discretized Fourier transform of the function $\boldsymbol{v}_{t-1}$ are $\hat{\boldsymbol{v}}_{t-1}(k) = \sum_{j=0}^{N-1} \boldsymbol{v}_{t-1}(\boldsymbol{x}_{j+1})e^{-2\pi \mathrm{i} jk/(N)} \in \mathbb{R}^{d_{\boldsymbol{v}_{t-1}}}$. Likewise, we denote the Fourier coefficients of the kernel by $\hat{\boldsymbol{\kappa}}_t(k) \in \mathbb{R}^{d_{\boldsymbol{v}_t} \times d_{\boldsymbol{v}_{t-1}}}$. Finally, the convolution can be expressed as

$$\mathsf{F}^{-1}(\mathsf{F}(\kappa_t) \cdot \mathsf{F}(\boldsymbol{v}_{t-1}))(\boldsymbol{x}) \approx \sum_{k=0}^{N-1} \hat{\boldsymbol{\kappa}}_t(k)\hat{\boldsymbol{v}}_{t-1}(k) \, \exp(2\pi \mathrm{i} kn/N). \tag{29}$$

Instead of projecting $\kappa_t$ to the Fourier space, the kernel is directly parameterized in Fourier space. Specifically, the Fourier coefficients of the kernel are parameterized $\hat{\boldsymbol{\kappa}}_{\boldsymbol{\theta}}$. Empirically, it has been found that directly parameterizing the matrix $\hat{\boldsymbol{\kappa}}_{\boldsymbol{\theta}} \in \mathbb{R}^{N \times d_{\boldsymbol{v}_t} \times d_{\boldsymbol{v}_{t-1}}}$ yields the best performance. In addition to their natural frequency domain processing, FNOs demonstrate computational efficiency with a sub-quadratic complexity of $\mathcal{O}(N \log N)$.

However, several studies (Fanaskov and Oseledets, 2023; Bartolucci et al., 2023) emphasize that choosing the training data from a fixed grid representation leads to a systematic bias. In addition, aliasing can occur for FNOs, as the maximum frequency mode that can be distinguished from lower frequency modes is bounded by the grid resolution. Fanaskov and Oseledets (2023) propose *spectral neural operators* that use Chebyshev or trigonometric polynomials. Additionally, multi-wavelet operators (Gupta et al., 2021) extend FNOs by utilizing multi-resolution wavelets to represent the kernel operator with a piecewise polynomial basis. This results in a sparse kernel representation, improving the operator learning performance.

In recent years, neural operators have gained considerable attention. Different architectures have been proposed based on the recent success in other research disciplines, such as attention-based (Cao, 2021), CNN-based (Raonić et al., 2023), or VAE-based (Seidman et al., 2023) architectures for operator learning. Finally, strides have been made to make neural operators universally applicable in various domains compliant to specific pre-conditions. As FNOs operate on uniform grids, domain-agnostic methods have been proposed that either map complex topologies to a latent uniform grid for the NO (Li et al., 2023a;b) or apply the NO on an extended uniform grid and masking out points outside the domain (Liu et al., 2023a). Furthermore, as neural operators are fully data-driven approaches, there are no guarantees that the learned solution operator adheres to the boundary value conditions of the considered PDE. Therefore, Saad et al. (2023) show that specific parameterizations of the kernel operator 25 can ensure compliance with Dirichlet, Neumann, and

periodic boundary conditions while Négiar et al. (2023) propose to employ a differentiable final layer which guarantees that the boundary constraints are not violated. Further geometric biases have been employed with neural operator architectures to be equivariant to rotations, translations, and reflections (Helwig et al., 2023), or explicitly being SE(3)-equivariant (Cheng and Peng, 2023). FNOs have also been adopted for general machine learning. FNOs have been incorporated into vision transformers to perform 'token mixing' in the Fourier domain rather than in pixel space (Guibas et al., 2022). These *adaptive* FNOs are developed to be more sparse, use fewer parameters for greater scalability, and have been deployed on weather forecasting tasks (Pathak et al., 2022).

The little regularization required for training and high flexibility of neural operators come at the cost of demanding plenty of training data to learn the desired operator. Therefore, one of the ongoing tasks is to incorporate physical priors to facilitate faster and more efficient learning. DeepONets and neural operators have been integrated with physics-informed losses, see section 3.3, thereby enhancing the data-driven operator learning loss with PDE constraints based on the PINN loss $\mathcal{L} = \alpha\mathcal{L}_{\text{NO}} + (1 - \alpha)\mathcal{L}_{\text{PINN}}$, $\alpha \in [0, 1]$ (Wang et al., 2021c; Li et al., 2021b). Setting $\alpha = 0$ recovers PINNs, but instead of a point-wise mapping, an operator is learned that maps an input function $\boldsymbol{a}$ to the solution $\boldsymbol{u}$. However, Saad et al. (2023) show that the auxiliary PINN loss does not always benefit the training of neural operators and may hinder its final performance. Du et al. (2024) highlight the shortcomings of the PINN loss as it requires possibly multiple differentiations through the network architecture. Thus, the authors propose to replace the PINN loss term with a new loss based on spectral theory. The proposed loss evaluates the residual term of the PINN loss in its spectral representations, which can be shown to simplify gradient propagation through the network and thus improve the optimization of neural operators with added physics-inspired losses.

The study of linear differential equations, particularly through Green's functions, has significantly shaped the development of operator learning architectures. Here, nonlinear operator approximation is achieved by compositions of linear operators with nonlinear activations. In the next section (section 3.5), we introduce a related but distinct approach, Koopman theory, which also leverages linear operators to approximate nonlinear dynamical systems. Unlike the compositional method, Koopman theory seeks to embed the domain into a latent space where the dynamics are governed by a linear, though potentially infinite-dimensional, operator.

Gin et al. (2021) demonstrated the close ties between operator learning and Koopman theory by proposing to learn latent representations of input and solution spaces, facilitating the approximation of Green's function (equation (24)) for the solution of linear PDEs. For discretizations $\boldsymbol{U}$ and $\boldsymbol{F}$ of $\boldsymbol{u}(\boldsymbol{x})$ and $\boldsymbol{f}(\boldsymbol{x})$ respectively, they propose learning embeddings $\boldsymbol{u}_{\boldsymbol{\phi}} = \boldsymbol{\phi_\theta}(\boldsymbol{U})$ and $\boldsymbol{f}_{\boldsymbol{\psi}} = \boldsymbol{\psi_\theta}(\boldsymbol{F})$ such that $\boldsymbol{L}\boldsymbol{u}_{\boldsymbol{\phi}} = \boldsymbol{f}_{\boldsymbol{\psi}}$. Solving the PDE in this embedding space is simply $\boldsymbol{U} = \boldsymbol{\phi_\theta}^{-1}(\boldsymbol{L}^{-1}\boldsymbol{\psi_\theta}(\boldsymbol{F}))$, providing $\boldsymbol{\phi}$ and $\boldsymbol{\psi}$ can be learned. The subsequent section will discuss the notion of learning effective representations for dynamical systems and the extension of linear dynamical systems theory to nonlinear operators through observable functions.

### 3.5 Latent Variable Models for Dynamical Systems

Latent variable models (LVM) (Bishop and Nasrabadi, 2006) refers to the assumption that an observed set of measurements are a consequence of an unobserved latent process. Much of statistics, signal processing and machine learning reduce to problems of this form. In the context of learning PDEs and dynamical systems, there is a desire to reduce complex systems into simpler ones using the idea of a latent coordinate system. Thus, LVMs motivate a different view on previously discussed methods for operator learning on finite-dimensional embeddings, see section 3.4, and physically-structured latent coordinate systems, see section 3.2. These methods construct latent coordinate representations enhancing the interpretability of system dynamics and enabling more efficient computational analysis. Building on this motivation, the following section surveys Koopman operator theory, which aims to find a latent coordinate system in which the dynamics become linear (Koopman, 1931; Koopman and Neumann, 1932). The linearization simplifies the forecasting and enables the use of well-established techniques from linear algebra for analyzing Another motivation is that when learning from measurement data, there is a degree of flexibility in the actual definition of the dynamical system and its internal state, and this ambiguity can be leveraged to learn a dynamical system that may not reflect the physical laws of the process, but may be easier to learn, capture the observed

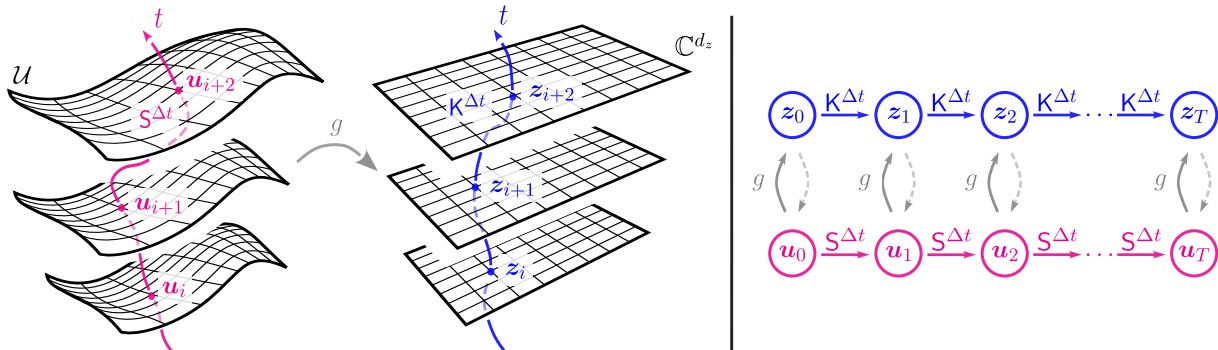

Figure 5: Schematic overview of the Koopman operator theory. **Left:** Other than analyzing the evolution of a dynamical system on $\mathcal{X}$, the theory studies a linear but potentially infinite-dimensional operator on $\mathcal{F} \in \{\mathbb{C}, \mathbb{R}\}$. **Right:** The graphical model provides an overview of how recent methods often address Koopman theory. Given the observations $\{\boldsymbol{u}_i\}_{i=0}^T$, the objective is twofold: (i) finding a feature representation transforming the observation into a latent coordinate system, and (ii) approximating the Koopman operator based on latent variables $\{\boldsymbol{z}_i\}_{i=0}^T$ assuming linear latent dynamics.

data, or use as a predictive model. In this section, we will discuss learning latent dynamics with both the linear and nonlinear assumption.

**Linear latent dynamics.** Koopman operator theory provides an alternative perspective on dynamical systems (Kutz et al., 2016; Brunton et al., 2021). Instead of analyzing the evolution of the system itself, the theory takes an operator-theoretic perspective. Consider $\boldsymbol{u}(\boldsymbol{x}, t) \in \mathcal{U}$ evolving on the Banach space depending on time and spatial coordinates $t \in \mathbb{R}$ and $\boldsymbol{x} \in \Omega$, respectively. Let $\mathsf{S}^t : \mathcal{U} \to \mathcal{U}$ be the corresponding flow $\mathsf{S}^{\Delta t}\boldsymbol{u}(\boldsymbol{x}, t) = \boldsymbol{u}(\boldsymbol{x}, t + \Delta t)$, propagating the current state $\boldsymbol{u}(\boldsymbol{x}, t)$ by a time $\Delta t$ and mapping it to the future state $\boldsymbol{u}(\boldsymbol{x}, t + \Delta t)$. Koopman operator theory revolves around an infinite dimensional linear operator $\mathsf{K}^{\Delta t}$ acting on a Hilbert space $\mathcal{H}$ of all observable functionals $g[\boldsymbol{u}] : \mathcal{U} \to \mathbb{C}$. The continuous-time Koopman operator is defined by

$$\mathsf{K}^{\Delta t}g[\boldsymbol{u}](\boldsymbol{x}, t) = g[\boldsymbol{u}](\boldsymbol{x}, t + \Delta t), \tag{30}$$

with $\mathsf{K}^{\Delta t}g[\boldsymbol{u}]$ being the observable at time $t + \Delta t$ initiated from $g[\boldsymbol{u}](\boldsymbol{x}, t)$ at time $t$ (Kutz et al., 2016). In the context of Hamiltonian systems, Koopman (1931); Koopman and Neumann (1932) introduced the Koopman operator. Later, Mezić (2005; 2013) extended the theoretical framework to address dissipative and non-smooth dynamics arising from the Navier-Stokes equation. While the flow $\mathsf{S}^{\Delta t}$ comes from arbitrary and potentially highly nonlinear dynamics, the observable space represents an infinite-dimensional linear system (see Figure 5). Consequently, we can use spectral analysis to extract coherent spatio-temporal structures of the nonlinear system embedded in the observable space. Given that $g[\boldsymbol{u}]$ is a functional, spectral analysis on the Koopman operator for arbitrary PDEs leads to eigenvalues $\lambda \in \mathbb{C}$ and eigenfunctionals $\psi[\boldsymbol{u}] : \mathcal{B} \to \mathbb{C}$ (Nakao and Mezić, 2020). In practice, however, the observables are often discretized in space and time resulting in a discrete-time dynamical system (Kutz et al., 2018). The system states simplify to finite-dimensional vectors $\boldsymbol{u}_i = \boldsymbol{u}(\boldsymbol{x}_i, t_i) \in \mathcal{U}$ and the observables to scalar functions $g : \Omega \to \mathbb{C}$. A spectral decomposition of the resulting discrete-time Koopman operator $\mathsf{K}^{i+1}$ leads to to $\mathsf{K}^{i+1}\lambda_j = \lambda_j\psi_j(\boldsymbol{u}_i)$, with $\{\psi_j, \lambda_j\}_{j=1}^\infty$ representing the associated eigenfunctions and eigenvalues. Thus, a vector of observables $\boldsymbol{g} = [g_1, \cdots, g_{d_g}]^T \in \mathbb{C}^{d_z}$ can be expressed by

$$\boldsymbol{g}(\boldsymbol{u}_i) = \sum_{j=1}^\infty \psi_j(\boldsymbol{u}_i) \begin{bmatrix} \langle \psi_j, g_1 \rangle \\ \vdots \\ \langle \psi_j, g_{d_g} \rangle \end{bmatrix} = \sum_{j=1}^\infty \psi_j(\boldsymbol{u}_i)\boldsymbol{v}_j, \tag{31}$$

with the $j$-th Koopman mode $\boldsymbol{v}_j \in \mathbb{C}^{d_g}$ projecting the observables $\boldsymbol{g}(\boldsymbol{u}_i)$ onto the $j$-th eigenfunction. Since the observables are linear combinations of the eigenfunctions, i.e. $\boldsymbol{g} \in \mathrm{span}(\{\psi_j\}_{j=1}^\infty)$, the temporal evolution

is given by the eigenvalues $\mathsf{K}^{i+1}\boldsymbol{g}(\boldsymbol{u}_i) = \sum_{j=1}^{\infty} \psi_j(\boldsymbol{u}_i)\lambda_j\boldsymbol{v}_j$. However, $\mathsf{K}^{i+1}$ acts on an infinite-dimensional Hilbert space – with an infinite number of eigenvalues, eigenvectors, and modes. Therefore, applied Koopman theory aims to identify a finite-dimensional latent coordinate system that is invariant to actions of $\mathsf{K}^{i+1}$ (Brunton et al., 2016a). This latent space, spanned by $d_z$ eigenfunctions $\{\psi_j\}_{j=1}^{d_z}$, is termed the Koopman invariant subspace (KIS). Consequently, we obtain a finite-dimensional linear operator $\boldsymbol{K} \in \mathbb{C}^{d_z \times d_z}$. The approximation depends crucially on the correct choice of observables. For multiple dynamical systems, such as the Burger or Schrödinger equations, we can select specific observables and obtain accurate approximations of $\boldsymbol{K}$ (Kutz et al., 2018; Nakao and Mezić, 2020). However, a general scheme for finding the observable functions is an ongoing challenge (Brunton et al., 2021).

Algorithms related to dynamic mode decomposition (DMD) provide entirely data-driven approaches to extract latent coordinate systems from time-series data (Schmid, 2022). They decompose complex systems into spatio-temporal coherent structures employing singular value decomposition (Kutz et al., 2016). Rowley et al. (2009) highlighted the connection between these approaches and Koopman operator theory. Given a temporal sequence $\{\boldsymbol{u}_i\}_{i=1}^{T}$, they operate as supervised learning schemes, addressing least-squares problems of the form

$$\mathcal{J}(\boldsymbol{K},\boldsymbol{\theta}) = \mathcal{L}_{\mathrm{Lin}}(\boldsymbol{K},\boldsymbol{\theta}) + \alpha\mathcal{L}_{\mathrm{Reg}}(\boldsymbol{K},\boldsymbol{\theta}) = \sum_{i=1}^{T-1} ||\boldsymbol{\Psi}'_{\boldsymbol{\theta}}(\boldsymbol{u}_{i+1}) - \boldsymbol{K}\boldsymbol{\Psi}_{\boldsymbol{\theta}}(\boldsymbol{u}_i)||_2^2 + \alpha\mathcal{L}_{\mathrm{Reg}}(\boldsymbol{K},\boldsymbol{\theta}). \qquad (32)$$

Here, $\mathcal{J}(\boldsymbol{K},\boldsymbol{\theta})$ comprises a one-step prediction loss $\mathcal{L}_{\mathrm{Lin}}(\boldsymbol{K},\boldsymbol{\theta})$ ensuring linearity and a regularization loss $\mathcal{L}_{\mathrm{Reg}}(\boldsymbol{K},\boldsymbol{\theta})$ scaled by $\alpha \in \mathbb{R}^+$. The dictionaries $\boldsymbol{\Psi}' : \Omega \to \mathbb{C}^{d_z}$ and $\boldsymbol{\Psi} : \Omega \to \mathbb{C}^{d_z}$ represent linear basis models approximating the invariant subspace based on provided feature functions $\{\phi^j(\boldsymbol{u}) : \Omega \to \mathbb{C}\}_{j=1}^{d_z}$. The original DMD framework, introduced by Schmid (2010), considered a one-to-one mapping, i.e. $\boldsymbol{\Psi}'(\boldsymbol{u}) = \boldsymbol{\Psi}(\boldsymbol{u}) = \boldsymbol{u}$, resulting in linear least-squares problem. Later research, such as extended DMD (EDMD) (Williams et al., 2015) or SINDy (Brunton et al., 2016b), explored more representative features employing nonlinear function approximations. However, choosing $\boldsymbol{\Psi}'$ and $\boldsymbol{\Psi}$ is challenging, commonly known as the bias-variance problem in machine learning (Bishop and Nasrabadi, 2006). It results in conflicting goals between (i) extracting a valid KIS and (ii) not overfitting to the given data (Otto and Rowley, 2019). Therefore, Li et al. (2017) and Yeung et al. (2019) proposed dictionaries with a partly trainable set of features, i.e., $\{\phi^j_{\boldsymbol{\theta}}(\boldsymbol{u}) : \Omega \to \mathbb{C}\}_{j=1}^{k}$ and $k < d_z$. EDMD with dictionary learning (EDMD-DL) considers shallow networks as function approximations and utilizes an $L^2$-regularization for $\boldsymbol{\theta}$ (Li et al., 2017). In contrast, Yeung et al. (2019) proposed deep-DMD employing deep learning architectures and a $L^1$-regularization. Batch methods iterate between a ridge regression to estimate $\boldsymbol{K}$ and a nonlinear gradient descent to optimize $\boldsymbol{\theta}$. Furthermore, Terao et al. (2021) introduced a variant of EDMD-DL, replacing the MLPs with NODEs (section 3.1). These approaches have demonstrated promising results in capturing chaotic behavior, such as that exhibited by the Duffing oscillator (Williams et al., 2015). Also, they found practical applications in fluid mechanics, for instance, in systems described by the Kuramoto-Sivashinsky equation (Li et al., 2017). However, they share the assumption of a full state observable i.e. $\boldsymbol{g}(\boldsymbol{u}) = \boldsymbol{u}$. This assumption is implemented in two ways. Firstly, through a linear reconstruction using the estimated eigenfunctions and Koopman modes. Alternatively, by incorporating a constant feature $\phi(\boldsymbol{u}) = \boldsymbol{u}$ into the dictionary. However, a significant challenge remains in the robust forecasting of out-of-distribution dynamics. Mouli et al. (2024a) addressed this problem employing a meta-learning framework that learns relevant feature functions across tasks. Similarly to SINDy (Brunton et al., 2016b), their approach employs nonlinear function approximators together with task-specific coefficient matrices to learn the dynamics for each task. In addition, they introduce a task-independent indicator matrix that learns the global causal structure across all tasks. Given the causal information, efficient adaptation to new tasks is achieved by learning only the task-specific coefficients. This method demonstrates the potential of meta-learning, discussed in section 4, to address challenges such as out-of-distribution dynamics.

In recent years, researchers explored the application of autoencoder architectures to extract a concise yet informative subspace from time series data (Takeishi et al., 2017; Lusch et al., 2018; Alford-Lago et al., 2022). As a result, they relaxed the assumption of a full state observable. The linear reconstruction becomes obsolete and a nonlinear reconstruction takes place using the decoder network. The objective from equation (32) extends to

$$\mathcal{J}(\boldsymbol{\theta}_e, \boldsymbol{\theta}_d, \boldsymbol{K}) = \alpha_1\,\mathcal{L}_{\mathrm{Recon}}(\boldsymbol{\theta}_e, \boldsymbol{\theta}_d) + \alpha_2\,\mathcal{L}_{\mathrm{Pred}}(\boldsymbol{\theta}_e, \boldsymbol{\theta}_d, \boldsymbol{K}) + \alpha_3\,\mathcal{L}_{\mathrm{Lin}}(\boldsymbol{\theta}_e, \boldsymbol{\theta}_d, \boldsymbol{K}) + \alpha_4\,\mathcal{L}_{\mathrm{Reg}}(\boldsymbol{\theta}_e, \boldsymbol{\theta}_d, \boldsymbol{K}), \quad (33)$$

with $\mathcal{L}_{\mathrm{Recon}}(\boldsymbol{\theta}_e, \boldsymbol{\theta}_d)$ and $\mathcal{L}_{\mathrm{Pred}}(\boldsymbol{\theta}_e, \boldsymbol{\theta}_d, \boldsymbol{K})$ being a reconstruction loss and a prediction loss, respectively. The former penalizes inaccurate reconstructions of the system state, i.e. $\boldsymbol{u}_i \approx (d_{\boldsymbol{\theta}_d} \circ e_{\boldsymbol{\theta}_e})\boldsymbol{u}_i$. The latter aims to correctly predict the next state after propagating the current one through the architecture, i.e. $\boldsymbol{u}_{i+1} \approx (d_{\boldsymbol{\theta}_d} \circ \boldsymbol{K} \circ e_{\boldsymbol{\theta}_e})\boldsymbol{u}_i$. Several works, including Takeishi et al. (2017); Lusch et al. (2018) and Alford-Lago et al. (2022) focused on forward prediction losses $\mathcal{L}_{\mathrm{Pred}}(\boldsymbol{\theta}_e, \boldsymbol{\theta}_d, \boldsymbol{K})$ and $\mathcal{L}_{\mathrm{Lin}}(\boldsymbol{\theta}_e, \boldsymbol{\theta}_d, \boldsymbol{K})$. Azencot et al. (2020) studied the effect of an additional backward prediction loss, i.e. $\boldsymbol{u}_{i-1} \approx (d_{\boldsymbol{\theta}_d} \circ \boldsymbol{D} \circ e_{\boldsymbol{\theta}_e})\boldsymbol{u}_i$. They introduced an additional operator for the backward dynamics $\boldsymbol{D}$ satisfying $\boldsymbol{K} \circ \boldsymbol{D} = \boldsymbol{D} \circ \boldsymbol{K} = \mathbb{I}$. However, these works solely looked at one-step predictions. Otto and Rowley (2019), on the other hand, introduced an autoencoder featuring linear recurrent latent embeddings. The proposed linearly-recurrent autoencoder network (LRAN) enhances the recurrent losses $\mathcal{L}_{\mathrm{Pred}}(\boldsymbol{\theta}_e, \boldsymbol{\theta}_d, \boldsymbol{K})$ and $\mathcal{L}_{\mathrm{Lin}}(\boldsymbol{\theta}_e, \boldsymbol{\theta}_d, \boldsymbol{K})$ in equation (33) by considering a weighted average over $\tau$-step predictions. Similarly, Wang et al. (2022c) and Liu et al. (2024a) both used recurrent structures to learn KIS and operators. However, they employ a composition of latent coordinate systems, each designed to capture different aspects of the time series data. The Koopman neural forecaster (KNF) decomposes the embedding to capture local and global behavior patterns separately (Wang et al., 2022c). In contrast, Liu et al. (2024a) proposed a hierarchy of Koopman forecaster blocks ('Koopa'). In each block, a Fourier filter separates the input signal to capture time-varying and time-invariant components. While these methods were effective for applications like weather forecasting (Liu et al., 2024a) and fluid mechanics, e.g., in modeling wake flow around cylinders (Otto and Rowley, 2019), they showed efficacy in quantifying the impacts of uncertainties in both model construction and forecasting. Therefore, several works, including Morton et al. (2019) and Pan and Duraisamy (2020), proposed using variational autoencoder (VAE) architectures (Kingma and Welling, 2014). These approaches enable the learning of a generative model that closely resembles the distribution over dynamics in the latent space.

Eigenvalues are crucial for understanding the temporal behavior of the dynamics in the latent embedding. Previous works, including Liu et al. (2024a) or Otto and Rowley (2019), have not explicitly addressed stability. As a consequence, the locally-linear latent dynamics were prone to instability. Therefore, recent research proposes physics-informed architectures to promote properties like stability. Erichson et al. (2019) suggested an autoencoder framework targeting stable latent Koopman dynamics, following Lyapunov stability principles. Pan and Duraisamy (2020), on the other hand, employed a skew-symmetric representation of the Koopman operator. This approach inherently forms stable latent dynamics and deviates from Erichson et al. (2019), where stability is integrated as a soft constraint within the objective function equation (33). For quasi-periodic systems, works like Lange et al. (2021) express the objective in the frequency domain leveraging the fast Fourier transformation (FFT). This dual formulation achieves globally linear latent embeddings by construction. To incorporate further physical principles such as conservation or measure preservation, Baddoo et al. (2023) or Colbrook (2023) reframed the regression problem from equation (32) as Procrustes problem. This formulation ensures that the solution lies on a manifold spanned by the physically imposed constraints. While demonstrating convergence guarantees and promising results for systems such as the Volterra integro-differential equation or linearized Navier–Stokes equations, these approaches rely on the choice of feature functions and assume linear reconstruction.

In our discussion so far, we have explored methods that use parameterized function approximations, e.g., polynomials or neural networks. However, these methods can be computationally intensive, especially when dealing with high-dimensional data. Autoencoders, as proposed by Takeishi et al. (2017); Lusch et al. (2018) and Alford-Lago et al. (2022), offer a solution by extracting a latent coordinate system with linear dynamics. However, using autoencoders adds complexity and potentially increases the non-convexity of the optimization problem. Additionally, they usually perform a dimensionality reduction, resulting in a lower-dimensional latent embedding. Yet, in Koopman theory, the observable space may represent an infinite-dimensional linear system. An alternative solution to this problem is to utilize non-parametric models. Williams et al. (2014) introduced a version of DMD called kernel DMD by reframing the equation (32) in its dual form. The resulting dual operator $\hat{\boldsymbol{K}}$ is obtained by inner products in feature space, facilitated by applying the kernel trick. Instead of explicitly defining the features for $\boldsymbol{\Psi}$, e.g., using parameterized function approximations, a kernel function $k : \Omega \times \Omega : \to \mathbb{R}$ is employed to implicitly compute these inner products Seeger (2004). Several works, including Kawahara (2016); Klus et al. (2020) and Das and Giannakis (2020), studied Koopman operator theory in reproducing kernel Hilbert spaces (RKHS). Recently, Kostic et al. (2022) and Bevanda et al. (2024) highlighted the connection between these kernel-based methods for

Koopman theory and statistical learning. This connection establishes a natural concept of risk into the estimation of the Koopman invariant subspace. However, a comprehensive discussion on non-parametric methods is beyond the scope of our paper. We encourage interested readers to explore the relevant literature for further insights.

Despite its theoretical concepts, the practical implementation of Koopman theory is ongoing (Brunton et al., 2021). While promising, autoencoder methods typically serve as dimensionality reduction techniques, conflicting with the concept of a potentially infinite-dimensional latent space. Nevertheless, they have shown remarkable performance in finding low-dimensional embedding for tasks involving high-dimensional solution spaces (Brunton et al., 2021). Furthermore, several works , including Lusch et al. (2018), have rendered the linear latent dynamics as state-dependent resulting in locally-linear dynamics. Although these architectures perform well on high-dimensional nonlinear fluid flow, they loosen the Koopman invariance assumption due to the operator's dependence on the latent state. These results motivate the discussion of architectures that employ dimensionality reduction while relaxing the linearity assumption for the dynamics in latent space , i.e., nonlinear latent dynamics .

**Nonlinear latent dynamics.** Modeling a dynamical system without assuming latent linear dynamics removes the inductive bias but yields a more expressive parametric model. In the context of this survey, these models are less relevant as their only inductive bias is the Markovian structure of the data, but these models perform well on a range of complex tasks, such as modeling natural language (Sutskever et al., 2014), music (Roberts et al., 2018) and video data (Babaeizadeh et al., 2018). One relevant line of work investigated how to implement these models with physically-structured latent space, like the models discussed in section 3.2. These include position-velocity encoders (Jonschkowski et al., 2017), Hamiltonian generative networks (Toth et al., 2020) and Newtonian variational autoencoder (Jaques et al., 2021). Another noteworthy line of research, discussed in section 3.4, concentrates on operator learning. DeepONets employ an autoencoder architecture to model nonlinear operators within a finite-dimensional latent space (Lu et al., 2019).

Encoder-decoder models can also be combined with neural ODEs (section 3.1). Dynamics-aware implicit neural representations (DINo, Yin et al. (2023)) uses an implicit neural representation (INR) as a decoder. INRs describe parametric functions trained to describe a domain, such as those already seen in PINNs (section 3.3) and also neural radiance fields used to render 3D scenes. DINo learns an additional latent variable $\alpha_t$ and corresponding neural ODE $\dot{\alpha} = f_\psi(\alpha)$, such that the true system is decoded using $g_\theta(\alpha_t, x)$, where $g_\theta$ is the INR. Encoding is performed using 'auto-decoding', where $\alpha$ is optimized with gradient-based methods to invert $g_\theta$. The benefit of this approach is that, like PINNs, we do not need to discretize the spatial domain. However, unlike PINNs, the LVM formulation is more expressive and $\alpha$ many different systems to be learned, for example different initial boundary conditions. A similar architecture is the continuous reduced-order modeling approach (CROM, Chen et al. (2023a)), which also combines neural ODEs and INRs, but unlike DINo uses a learned discretization-dependent encoder during learning, rather than auto-decoding.

More recently, significant progress has been made in machine learning for weather prediction using deep latent variable models. While these models are primarily data-driven and do not have physically-interpretable latent spaces, their performance has been achieved using several small but crucial physics-informed design decisions that we will discuss here.

**Nonlinear latent dynamics for weather prediction.** Medium-range numerical weather prediction (NWP) requires forecasting the global weather system over a period of several days using computation models of the Earth's climate system. There are several challenges in achieving accurate NWP. Weather forecasting is inherently multi-modal, multi-scale, and multivariate, predicting temperature, pressure, wind speeds, and precipitation from a diverse range of measurement sources. Forecasting must also accurately anticipate rare extreme weather events, whose effects may be the most consequential. Despite these challenges, NWP has advanced significantly due to progress in weather models, numerical methods, and availability of computational resources (Bauer et al., 2015).

Weather forecasting traditionally relies on computationally intensive physically-derived models, often combined in ensembles, to mitigate uncertainties arising from chaotic atmospheric dynamics and measurement

limitations. Despite advancements, the accuracy of these methods for medium-range, regional, and global climate modeling is dictated by the computational resources and model fidelity. Additionally, they cannot leverage extensive historical weather data beyond optimizing model parameters with system identification. Moreover, the complexity of weather dynamics requires combining models for each component (e.g. clouds, pollution, etc), and jointly tuning these models to improve net prediction accuracy is challenging/ Machine learning approaches offer promise by directly predicting measurements from historical datasets, circumventing some computational challenges. Recent years have witnessed a significant advancement in machine learning methods for weather prediction, enabled by large capacity models, hardware accelerators, and clean historical datasets (Voosen, 2023). We discuss the design of three prominent nonlinear latent variable models in this context. An empirical analysis of deep learning-based weather prediction can be found in Karlbauer et al. (2024).

FourCastNet (Kurth et al., 2023), a neural network for global-scale weather forecasting, combines adaptive Fourier neural operators (AFNO) with a vision transformer (ViT). AFNO serves as a resolution-invariant encoder/decoder, adept at capturing physical phenomena. ViT tokenizes AFNO representation and employs self-attention to model complex spatial and temporal interactions in weather dynamics. Initially trained for single-time step prediction, it's fine-tuned for consecutive predictions, significantly speeding up forecasting compared to physics-based NWP. However, its accuracy falls short of current NWP systems, prompting ongoing improvements.

Bi et al. (2023) proposed Pangu-Weather. Like FourCastNet, this model builds off vision transformers, but instead of using AFNOs, proposes a novel three-dimensional (3D) Earth-specific transformer architecture (3DEST) to better model weather correlation with latitude, longitude, and altitude. The key design decision is in the positional encoding used by the 3DEST, which encodes absolute position coordinates rather than relative ones and incorporates translation invariance. This encoding uses 527 times more embedding parameters, but the authors did not observe issues with prediction or optimization. The model also employs 'hierarchical temporal aggregation' to address error accumulation in long-horizon forecasts. To mitigate this source of error, the authors learn models that are trained on 1-, 3-, 6- and 24-hour intervals so that long-horizon forecasts can be computed using an aggregation of models, using 3-4 predictions at most. However, with each 3D ViT having 64M parameters, this aggregation strategy results in a 256M parameter model, which presents engineering issues at deployment. Similarly, Schmude et al. (2024) proposed Prithvi-WxC, a ViT-based approach with multi-axis attention to facilitate long-range interaction with limited attention window sizes and, to mitigate block boundary artifacts, an attention window shifting method was introduced. The model uses 160 input variables in order to address a more diverse set of downstream tasks. To reason over this many inputs, it uses both local and global masking and attention mechanisms during training to learn to model local and global phenomena. However, the resulting model has 2.3B parameters, posing significant challenges to deployment.

Lam et al. (2023) introduced GraphCast, a graph neural network (GNN) approach for medium-range forecasting. GraphCast builds on the work of Pfaff et al. (2021) learning mesh-based simulations using graph neural networks. GraphCast utilizes a hierarchical graph structure over the globe, resembling 3DEST, with six levels of resolution. The model's encoder projects Earth-oriented grid measurements onto this multi-mesh representation while the decoder aggregates nearby mesh values back to the grid. Operating on a 40,962-node multi-mesh, the GNN conducts simulation via simultaneous message passing at various spatial resolutions. Despite making 6-hour predictions, during training, it is evaluated autoregressively for up to 3 days. Remarkably, GraphCast can predict a 10-day forecast in under a minute at $0.25°$ global resolution with approximately 40 million parameters. Comparative evaluations suggest GraphCast outperforms Pangu-Weather across most measurement modalities, possibly due to the GNN's inductive bias, which resembles finite-element PDE solvers and propagates computations coherently in space. This behavior contrasts with ViT architecture, which needs latent self-attention mechanisms to learn such computations. However, the graph architecture does complicate some aspects of forecasting, such as downscaling and regional prediction.

Further extensions of this work use diffusion models (Sohl-Dickstein et al., 2015; Ho et al., 2020) for decoding. GenCast (Price et al., 2025) extends GraphCast by using a graph transformer for the latent dynamics and a conditional diffusion model for decoding. Compared to GraphCast, which has a deterministic decoder, sampling GenCast produces diverse high-fidelity forecasts. GraphCast is less capable of such high-fidelity

predictions since the deterministic decoder requires averaging over any present stochasticity. Precipitation diffusion (PreDiff, Gao et al. (2023)) also uses a conditional diffusion model for decoding weather predictions. Using a VAE to encode several steps of observations, the conditional diffusion model is trained to forecast a short lookahead window. An interesting aspect of this model is incorporating physical plausibility into predictions. For image generation, *classifer-based guidance* (Dhariwal and Nichol, 2021) uses the gradient of a pretrained classifier to improve the quality of the image generation. Similarly, Gao et al. (2023) replaces the classifier with *knowledge-guidance*, for example, a function that scores samples that reflect energy conservation, and uses this function to guide samples towards these physically plausible characteristics. The authors use energy conservation guidance in an *n*-body simulation experiment and anticipated precipitation intensity for a precipitation forecast experiment.

Despite progress, several challenges persist in current weather forecasting models. A significant hurdle is their reliance on 'reanalysis' data derived from complex Bayesian state estimation processes, which combine meteorological measurements with simulation models. This reliance on processed data prevents these ML models from forecasting in real time using raw data as desired. Additionally, challenges include scaling models to go beyond the $0.25°$ training data and match the high spatial resolution of $0.1°$ of top NWP systems, extending forecasting horizons, and increasing model capacity. There is also the question of the extent to which physics-based elements from NWP can be combined with ML models to enhance their performance, given that NWP forecasts are inherently physically plausible, while current ML-based forecasts lack this feature. This question is an active topic of study, e.g., (Kochkov et al., 2024). The next section explores examples of incorporating physics-informed elements into model learning.

### 3.6 Bridging Domain Knowledge and Function Approximation for System Identification

System identification approaches lie on a spectrum depending on modeling assumptions. If a significant amount of domain knowledge is assumed, the dynamics model will be highly structured. If correct, it will generalize well but will most likely be inflexible to unanticipated observed phenomena. On the other hand, one could model with no domain knowledge and use a black-box function approximator[3]. These models have no generalization guarantees and, therefore, require expansive datasets and regularization to ensure good modeling accuracy across the domain of interest. To achieve an effective combination of these two approaches, one would need to balance structural biases that facilitate generalization with adaptivity to the data that prevents the biases from leading to underfitting (Ljung, 2010; Lutter et al., 2021). This section reviews these approaches, which build on models discussed earlier.

Physics-informed neural networks can incorporate an additional dataset as an additional loss term in the objective. In addition to residual and boundary loss terms, the combined objective (equation (19)) has an additional dataset loss similar to the boundary loss term (Raissi et al., 2017a;b). Moreover, since the PINNs objective is quite general to any differentiable function approximator, Wang et al. (2021d) extend DeepONets by adding the PINNs residual loss to the data-driver operator loss as a way of encoding physics knowledge.

Yang et al. (2020) consider a generative approach using PINNs. Generative adversarial networks (GANs) learn a generator that learns to predict samples that match a dataset and a discriminator that learns to differentiate real and generated samples. Learning both models jointly should converge to a realistic data generator. PI-GANs incorporate PINNs into the stochastic generator using a reference PDE, where samples take the form $\boldsymbol{u}$ and $\mathsf{L}\boldsymbol{u}$ and then passed into the discriminator $\boldsymbol{d} : \mathbb{R}^{6d_u} \to \mathbb{R}$ with true measurements of $\boldsymbol{u}$ and $\boldsymbol{f}$, so for a dataset of size $N$ the discriminator is evaluated on all six terms, i.e.,

$$\boldsymbol{d}(\boldsymbol{u_\theta}(\boldsymbol{x}_i, \xi_i), \boldsymbol{u}(\boldsymbol{x}_i), \mathsf{L}\boldsymbol{u_\theta}(\boldsymbol{x}_i, \xi_i), \boldsymbol{f}(\boldsymbol{x}_i), \boldsymbol{u_\theta}(\boldsymbol{x}_j, \xi_j), \boldsymbol{b}(\boldsymbol{x}_j)), \quad \xi \sim \mu(\cdot), \ \boldsymbol{x}_i \in \mathcal{X}, \ \boldsymbol{x}_j \in \partial\mathcal{X}, \ i, j \in [0, N].$$

The benefit of this approach is that the discriminator learns a data-driven metric between the model and the dataset, in contrast to the squared error used by vanilla PINNs for all terms. However, the paper does not compare the performance between objectives, and PI-GANS are more expensive to train than PINNs.

---

[3]One could argue that using many popular function approximators, such as Gaussian processes, come with a smoothness assumption

Chen et al. (2021) propose a physics-informed approach for using learning PDEs from data. They posit that the solution $\boldsymbol{u}$ is linear in a broad set of pre-specified physics-informed features, e.g.

$$\boldsymbol{u}(\boldsymbol{x}, t) = \boldsymbol{W}\boldsymbol{\phi}(t, \boldsymbol{x}), \tag{34}$$

$$\boldsymbol{\phi}(t, \boldsymbol{x}) = [1, \boldsymbol{u}(\boldsymbol{x}, t), \boldsymbol{u}(\boldsymbol{x}, t)^2, \ldots, \operatorname{vec}(\nabla_{\boldsymbol{x}}\boldsymbol{u}(\boldsymbol{x}, t)), \nabla_t \boldsymbol{u}(\boldsymbol{x}, t), \ldots, \sin(\boldsymbol{u}(\boldsymbol{x}, t)), \ldots]^\top. \tag{35}$$

The training objective combines measurements of $\boldsymbol{u}$ and its temporal and spatial derivatives similar to PINNs. Moreover, to encourage sparsity in the learned solution with respect to the features, an $L^1$ penalty on $\boldsymbol{W}$ is also incorporated into the training objective.

Haußmann et al. (2021) consider neural stochastic differential equations with partial knowledge of the dynamics, captured by $\boldsymbol{r}(\boldsymbol{x}, t)$, which captures the mean function of the process with weights $\gamma \in [0, 1]^d$

$$\mathrm{d}\boldsymbol{x}_t = \boldsymbol{f_\theta}(\boldsymbol{x}_t, t)\,\mathrm{d}t + \gamma \circ \boldsymbol{r}(\boldsymbol{x}_t, t)\,\mathrm{d}t + \boldsymbol{g}(\boldsymbol{x}_t, t)\,\mathrm{d}w_t, \quad \boldsymbol{y}_t \sim p(\cdot \mid \boldsymbol{x}_t). \tag{36}$$

To optimize $\boldsymbol{\theta}$, a PAC Bayesian approach is adopted to regularize the parameters against a prior $p(\boldsymbol{\theta})$,

$$\mathbb{E}_{\boldsymbol{\theta} \sim q_\phi(\cdot)}\Big[\sum_t \log(\boldsymbol{y}_t \mid \boldsymbol{x}_t, \boldsymbol{\theta})\Big] + \sqrt{\mathbb{D}_{\mathrm{KL}}[q_\phi(\boldsymbol{\theta}) \mid\mid p(\boldsymbol{\theta})] + \log(4\sqrt{N}/\delta))/2N}. \tag{37}$$

where $N$ is the dataset size and $\delta \in [0, 1]$ is the probability of the PAC Bayes generalization bound. This objective is optimized using reparameterized gradients and the Euler-Maruyama integration of the stochastic process.

Long et al. (2022) combine differential equations and Gaussian processes by leveraging that the derivatives of a Gaussian process are also Gaussian processes as long as the kernel is sufficiently differentiable. Their method is a pseudo-Bayesian method that combines the PINNs residual objective with the typical Gaussian process prior and likelihood, minimizing the following expression w.r.t. pseudo-posterior $q(\boldsymbol{u})$ over solutions $\boldsymbol{u}$, where $p(\boldsymbol{u} \mid \boldsymbol{X}) = \mathcal{GP}(\boldsymbol{0}, \mathcal{C}(\boldsymbol{X}))$ and

$$\mathbb{E}_{\boldsymbol{u} \sim q(\cdot)}[-\log p(\boldsymbol{Y} \mid \boldsymbol{u}, \boldsymbol{X})] + \alpha^{-1}(\mathsf{L}\boldsymbol{u}(\boldsymbol{X}) - \boldsymbol{f}(\boldsymbol{X}))^2 + \mathbb{D}_{\mathrm{KL}}[q(\boldsymbol{u}) \mid p(\boldsymbol{u})]. \tag{38}$$

As a result, the pseudo-posterior solution combines the typical GP prior and data likelihood with the PDE residual pseudo-likelihood, controlled by hyperparameter $\alpha > 0$:

$$q(\boldsymbol{u} \mid \mathcal{D}) = \mathcal{N}\big(\boldsymbol{0}; \mathsf{L}\boldsymbol{u}(\boldsymbol{X}) - \boldsymbol{f}(\boldsymbol{X}), \alpha \boldsymbol{I}\big)\, p(\boldsymbol{Y} \mid \boldsymbol{u}, \boldsymbol{X})\, p(\boldsymbol{u}). \tag{39}$$

This posterior cannot be obtained in closed form, so approximate inference is performed. This is done by parameterizing the nonparametric GP with parametric inducing points as a synthetic dataset and optimizing these points to minimize equation (38). These inducing points play the role of collocation points in PINNs. The authors also demonstrate that an additional GP can be incorporated into the residual term to capture any unknown terms in the PDE, which means two GPs are jointly inferred. In their experiments, this model was shown to perform better than PINNs with fewer collocation points. PINNs could outperform their method with more collocation points, and the authors did not demonstrate that their method scaled to a high number of inducing points. This is because non-parametric GPs suffer from a $\mathcal{O}(N^3)$ complexity for $N$ training points when arbitrary kernels are used. However, specialized kernels exist that enable more graceful scaling.

### 3.7 Physics-inspired Model Invariances

Physical models in the form of ODEs, PDEs, or algebraic equations are often invariant to specific types of operations or transformations. For instance, many dynamical processes such as molecular, rigid body, elasto or fluid dynamics, chemical reaction kinetics, etc. are modeled under the assumption of energy conservation and the laws of thermodynamics (Cueto and Chinesta, 2023). This also translates to continuum material models, which should be invariant to the orientation of the coordinate system and consider the orientation of the material's microstructure. Likewise, in computer vision, object detection should be invariant to the position, orientation, and scale of the object. Graph-based representations, e.g., in the context of molecular

interactions or particle systems (Sanchez-Gonzalez et al., 2020), should often also be invariant to rotations, translations, reflections, and permutations. To discover invariants and conservation quantities from datasets using machine learning, Wetzel et al. (2020) proposed a "Siamese twin" neural network architecture, which aims to classify data points as related to the same invariant value. Similarly, Ha and Jeong (2021) introduced an intuitive noise-variance loss function for training a NN to classify data points related and non-related to the same invariant value.

Mathematically defining and incorporating relevant invariance and equivariances into machine learning models has led to several advancements in applying machine learning to complex physical domains, e.g., (Cohen and Welling, 2016; Fuchs et al., 2020; Satorras et al., 2021; Hoogeboom et al., 2022; Weiler et al., 2023; Thiemann et al., 2025). Denoting a set of transforms as $\mathsf{T}$, the $\mathsf{T}$-invariance of a function or operator can be expressed as:

$$f(T(\boldsymbol{x})) = f(\boldsymbol{x}) \quad \forall T \in \mathsf{T}, \tag{40}$$

and the $\mathsf{T}$-equivariance as:

$$f(T(\boldsymbol{x})) = T(f(\boldsymbol{x})) \quad \forall T \in \mathsf{T}, \tag{41}$$

For instance, a "rotation invariant" function $f : \mathbb{R}^3 \to \mathbb{R}$ is invariant under transformations in the 3D rotational group $\mathsf{T} = SO(3)$, where $T : \mathbb{R}^3 \to \mathbb{R}^3$, $T(\boldsymbol{x}) = \boldsymbol{R}\boldsymbol{x}$ with $\boldsymbol{R} \in \mathbb{R}^{3 \times 3}$, $\boldsymbol{R}^T \boldsymbol{R} = \boldsymbol{I}$. According to Noether's theorem, such rotational symmetries are also closely linked to conservation laws, which imply that some quantity $f$ of a system, such as energy, momentum, mass, charge, etc., is conserved under any admissible trajectory $T$, i.e., invariant.

When applying machine learning in these contexts, embedding such known invariances and conservation properties into the models can greatly increase their accuracy, robustness, and generalization abilities and at the same time reduce training data requirements. Prior knowledge about invariances can be considered at all stages of the design of a physics-informed ML pipeline, i.e., in the problem conception or feature engineering (inputs and outputs of the model), the training and test data collection and curation, the ML architecture design, and the loss function formulation. Thus, we propose the following taxonomy for incorporating a $\mathsf{T}$-invariance, as defined above in equation (40), and likewise also a $\mathsf{T}$-equivariance, into parametric a model $f_{\boldsymbol{\theta}}$:

$$\text{(model-based)} \qquad f_{\boldsymbol{\theta}}(\boldsymbol{x}) = f_{\boldsymbol{\theta}}(T(\boldsymbol{x})) \quad \forall T \in \mathsf{T}, \tag{42}$$

$$\text{(feature-based)} \qquad f_{\boldsymbol{\theta}}(\phi(\boldsymbol{x})) = f_{\boldsymbol{\theta}}(\phi(T(\boldsymbol{x}))) \quad \forall T \in \mathsf{T}, \tag{43}$$

$$\text{(data-based)} \qquad \mathcal{D}_{\text{aug}} = \{\{y_n, \boldsymbol{x}_n^{(m)}\}_{n=1}^N\}_{m=1}^M, \; \boldsymbol{x}_n^{(m)} = T^{(m)}(\boldsymbol{x}_n), \; T^{(m)} \sim p(T), \tag{44}$$

$$\text{(objective-based)} \qquad \mathcal{L}_{\text{invar}}(\boldsymbol{\theta}) = \mathcal{L}(\boldsymbol{\theta}) + \mathbb{E}_{T \sim p(\cdot)}[\mathcal{D}(f_{\boldsymbol{\theta}}(\boldsymbol{x}_n), f_{\boldsymbol{\theta}}(T(\boldsymbol{x}_n)))]. \tag{45}$$

In the following, we now explore how these different implementations of invariances have been used in practice in various fields of application.

**Thermodynamic consistency of material models.** In mechanical, thermal, electromagnetic, and similar (including coupled, multi-physical) continuum theories, the governing PDEs, see equation (3) include constitutive models that describe the material behavior and must adhere to the laws of thermodynamics (Šilhavý, 1997). Generally, energy conservation can be ensured by formulating material models in terms of potentials such as the internal, Gibbs, or Helmholtz free energy densities. For instance, in hyperelastic material theory, the mechanical stress tensor $\boldsymbol{P} \in \mathbb{R}^{3 \times 3}$ is derived as the gradient of the internal (strain) energy density $\Psi(\boldsymbol{F}) : \mathbb{R}^{3 \times 3} \to \mathbb{R}$ with respect to the deformation gradient tensor $\boldsymbol{F} \in \mathbb{R}^{3 \times 3}$:

$$\boldsymbol{P}(\boldsymbol{F}) = \frac{\mathrm{d}\Psi}{\mathrm{d}\boldsymbol{F}} \qquad \Rightarrow \qquad \frac{\mathrm{d}\Psi}{\mathrm{d}t} - \boldsymbol{P} : \frac{\mathrm{d}\boldsymbol{F}}{\mathrm{d}t} = 0. \tag{46}$$

Thus, the high degree of flexibility offered by ML methods can be exploited to formulate constitutive models that strictly ensure energy conservation in a *model-based* fashion by parameterizing $\Psi$. So far, such models have been developed for elastic, electro-elastic, and magneto-elastic materials in terms of internal energy potentials $\Psi_{\boldsymbol{\theta}}$ using, e.g., neural networks (Le et al., 2015; Fernández et al., 2020; Linka et al., 2021; Klein

et al., 2022a; Kalina et al., 2024), Gaussian process regression (Fuhg et al., 2022; Ellmer et al., 2024; Pérez-Escolar et al., 2024), or sparse regression (Flaschel et al., 2021). In the training process of these models, since the main quantity of interest is in fact not the potential $\Psi_{\boldsymbol{\theta}}$ itself, but its input-derivative $\boldsymbol{P} = \mathrm{d}\Psi_{\boldsymbol{\theta}}/\mathrm{d}\boldsymbol{F}$, it is highly beneficial to include the stress MSE with a loss term, which is denoted as Sobolev training, see Czarnecki et al. (2017); Vlassis and Sun (2022):

$$\mathcal{L}_{\text{Sobolev}}(\boldsymbol{\theta}) = \frac{C_{\Psi}}{m} \sum_{i=1}^{m} \left( \Psi_i - \Psi_{\boldsymbol{\theta}}(\boldsymbol{F}_i) \right)^2 + \frac{C_{\boldsymbol{P}}}{m} \sum_{i=1}^{m} \left( \boldsymbol{P}_i - \frac{\mathrm{d}\Psi_{\boldsymbol{\theta}}}{\mathrm{d}\boldsymbol{F}}(\boldsymbol{F}_i) \right)^2 . \qquad (47)$$

Furthermore, it should be mentioned that the energy conservation and favorable properties of the material models can only be preserved throughout a simulation when dedicated spatial discretization and time integration methods are used. For example, the mixed finite element methods and energy-momentum scheme developed for hyperelastic, physics-augmented NNs in Franke et al. (2023).

Strictly speaking, dissipative systems do not possess an invariance property, but they must fulfill the second law of thermodynamics, which is mathematically formulated as an inequality. For dissipative material behaviors such as elastoplasticity, viscoelasticity, or damage in mechanics, magnetic hysteresis, viscous flows, etc., adherence to the second law of thermodynamics can be guaranteed by formulating the models according to the generalized standard materials (GSM) framework (González et al., 2019). Here, additional internal (latent, non-observable) variables $\boldsymbol{\gamma}$ are introduced, which the internal energy potential $\Psi$ as well as a dissipation potential $\Phi$ depend on. The evolution of the internal variables $\dot{\boldsymbol{\gamma}}$ is then defined in terms of a compatibility condition between both potentials:

$$\Psi = \Psi(\boldsymbol{F}, \boldsymbol{\gamma}), \quad \Phi = \Phi(\boldsymbol{F}, \boldsymbol{\gamma}, \dot{\boldsymbol{\gamma}}) \qquad \Rightarrow \qquad \boldsymbol{P}(\boldsymbol{F}, \boldsymbol{\gamma}) = \frac{\mathrm{d}\Psi}{\mathrm{d}\boldsymbol{F}}, \quad \frac{\mathrm{d}\Psi}{\mathrm{d}\boldsymbol{\gamma}} + \frac{\mathrm{d}\Phi}{\mathrm{d}\dot{\boldsymbol{\gamma}}} = 0, \quad -\frac{\mathrm{d}\Psi}{\mathrm{d}\boldsymbol{\gamma}} \cdot \dot{\boldsymbol{\gamma}} \geq 0. \qquad (48)$$

Flaschel et al. (2024) demonstrated sparse regression concepts for obtaining GSM models for viscoelastic and elastoplastic material behaviors in the small strain regime, while Rosenkranz et al. (2023) compared various NN-based architectures, including RNNs with not guaranteed thermodynamics consistency and GSMs with FFNN potentials $\Psi$ and $\Phi$, and Holthusen et al. (2023) extended and applied the concepts in the finite deformation regime using constitutive ANNs. However, thermodynamic consistency and the fulfillment of the dissipation inequality can also be ensured by transferring classical rheological constitutive models to the ML context, see e.g. (Masi et al., 2021; Fuhg et al., 2023; Meyer and Ekre, 2023; Abdolazizi et al., 2024), thus applying a very strong, potentially over-restrictive bias on the model architectures.

**Rotational invariances of material models.** In material modeling, also the consideration of rotational invariances is crucial. Hyperelastic material models, see equation (46), have to be *objective*, i.e., invariant to changes of the observer in terms of arbitrary rotations $\boldsymbol{Q} \in SO(3)$, and material frame indifferent, i.e., invariant to changes of the reference configuration in terms of rotations $\boldsymbol{R} \in \mathcal{G}$ that are within their material symmetry group $\mathcal{G} \subseteq O(3)$ (Šilhavý, 1997). Mathematically, this can be formalized as:

$$\Psi(\boldsymbol{Q} \cdot \boldsymbol{F} \cdot \boldsymbol{R}^T) = \Psi(\boldsymbol{F}) \quad \forall \boldsymbol{F} \in GL^+(3), \boldsymbol{Q} \in SO(3), \boldsymbol{R} \in \mathcal{G}, \qquad (49)$$

which directly leads to the equivariances of the stress tensor as:

$$\boldsymbol{P}(\boldsymbol{Q} \cdot \boldsymbol{F} \cdot \boldsymbol{R}^T) = \boldsymbol{Q} \cdot \boldsymbol{P}(\boldsymbol{F}) \cdot \boldsymbol{R}^T \quad \forall \boldsymbol{F} \in GL^+(3), \boldsymbol{Q} \in SO(3), \boldsymbol{R} \in \mathcal{G}. \qquad (50)$$

In the classical, analytical way of formulating material models, adherence to both invariance (and equivariance) conditions is usually achieved by not expressing $\Psi$ directly in terms of the second-order tensor $\boldsymbol{F}$, but using ***scalar invariants***. For the simplest case of isotropic materials, which exhibit completely direction-independent behavior as $\mathcal{G} = O(3)$, the three scalar invariants $I_1, I_2, I_3 \in \mathbb{R}$ of the tensor $\boldsymbol{C} = \boldsymbol{F}^T \boldsymbol{F}$:

$$I_1 = \mathrm{tr}(\boldsymbol{C}), \quad I_2 = \tfrac{1}{2} \left( (\mathrm{tr}(\boldsymbol{C}))^2 - \mathrm{tr}\left(\boldsymbol{C}^2\right) \right), \quad I_3 = \det(\boldsymbol{C}), \qquad (51)$$

are both objective and material frame indifferent.

Thus, also machine learning-based hyperelastic material models can be physics-augmented and by construction fulfill these rotational invariances in a *feature-based* way by representing the strain energy in terms of

these invariants, i.e., $\Psi(\boldsymbol{F}) = \Psi_{\boldsymbol{\theta}}(I_1, I_2, I_3)$, using neural networks, Gaussian processes, etc. (Klein et al., 2022b; Linka et al., 2021; Fuhg et al., 2022). Similarly, for anisotropic materials with other symmetry groups $\mathcal{G}$ and also for electro- or magneto-elastic couplings, invariants can be derived and employed for machine learning of material models (Klein et al., 2024; Kalina et al., 2024; Pérez-Escolar et al., 2024). Furthermore, invariants can also be used to develop alternative representations of the material models, e.g., circumventing the learning of the internal energy density $\Psi$ and instead directly expressing stress coefficients through NNs (Fuhg et al., 2024a), which avoids the computational effort of differentiation of the NN through back-propagation in order to compute the stresses.

Another *model-based* route for ensuring these rotational invariances can also be to express $\Psi$ in terms of the tensors $\boldsymbol{F}$ or $\boldsymbol{C}$ (where the latter already ensures objectivity as $(\boldsymbol{QF})^T(\boldsymbol{QF}) = \boldsymbol{F}^T\boldsymbol{F}$), but augmenting the ML model architecture with algebraic group symmetrization approaches, see Itskov (2001) and its adaption for NNs in Fernández et al. (2020); Klein et al. (2022b):

$$\Psi(\boldsymbol{F}) = \frac{1}{|\mathcal{G}|} \sum_{\boldsymbol{R} \in \mathcal{G}} \Psi_{\boldsymbol{\theta}}(\boldsymbol{F} \cdot \boldsymbol{R}). \tag{52}$$

Here, the training and evaluation of the model require multiple evaluations of the NN $\Psi_{\boldsymbol{\theta}}$ (or any other type of ML model), which is computationally costly, especially for the stress evaluations. Furthermore, this approach can only exactly incorporate the material symmetry requirement if the symmetry group is finite, i.e., $|\mathcal{G}| < \infty$. Nevertheless, in Klein et al. (2022b), it was also applied to transverse isotropy by sampling a uniformly distributed subset of 60 rotations from the corresponding infinite material symmetry group, yielding highly accurate results.

As the model formulation in equation (52) using $\boldsymbol{F}$ does not fulfill the objectivity condition, Klein et al. (2022b) also uses a *data-based* augmentation. Here, the training dataset, which consisted of deformations $\boldsymbol{F}_i$ with typically only a specific observer direction, was augmented by including additionally also rotated data points $\{\boldsymbol{Q} \cdot \boldsymbol{F}_i; \Psi_i, \boldsymbol{Q} \cdot \boldsymbol{P}_i\}$ with $\boldsymbol{Q} \in SO(3)$. Using 64 uniformly sampled rotations, the models could then learn to be objective up to the desired model accuracy. While this approach increases the training duration due to the extended dataset size and only weakly includes the physical requirement, it preserves the faster evaluation of the trained model compared to the algebraic model augmentation using group symmetrization. Similarly, instead of augmenting the dataset, the same effect could also be achieved using a penalty approach, similar to the PINN objective section 3.3, by adding a loss function term that penalizes violation of equation (49) for $\boldsymbol{Q}$ and $\boldsymbol{R}$ being sampled in (subsets of) the respective groups. Again, this would increase the training time but would not affect the evaluation of the trained models.

**Functional requirements of material models.** Machine learning models that are meant to substitute physical models or numerical simulations, or ingredients thereof, are typically subject to further mathematical requirements such as continuity and continuous differentiability, monotonicity, convexity or concavity, invertibility, integrability, etc. Though these are not directly invariances, these functional requirements are often crucial for ensuring invariances, well-behavior, and robustness of ML models.

For instance, in continuum mechanics, the PDE boundary value problem only has a well-defined, unique solution if the included hyperelastic material model is elliptic (Šilhavý, 1997). This can be ensured by formulating the internal energy potential as polyconvex, i.e., $\Psi(\boldsymbol{F}) \equiv \Psi(\boldsymbol{F}, \det(\boldsymbol{F})\boldsymbol{F}^{-T}, \det(\boldsymbol{F}))$ must be convex in terms of its three separate arguments. While analytically formulating multivariate functions as convex is a difficult task, the so-called input-convex neural networks (ICNN) proposed by Amos et al. (2017) have relatively simple to implement restrictions that ensure convexity: their weights must be non-negative (except for the first layer) and the activation functions monotonously increasing and convex. Thus, using ICNNs, the functional requirement of polyconvexity can be embedded into the architectures of NN-based models $\Psi_{\boldsymbol{\theta}}$ for hyperelastic materials, see Klein et al. (2022b); Linden et al. (2023); Fuhg et al. (2024a). When combined with the other aforementioned requirements, such as thermodynamic consistency, objectivity, and material symmetry, these physics-augmented NN material models can be accurately trained on very sparse training data, show excellent generalization behavior, and capture highly nonlinear (meta-) material behaviors (Klein et al., 2022b; Linden et al., 2023). Furthermore, the interpretability (and generalization) can be increased by applying sparsification strategies such as $L^0$-regularization in the training process (Fuhg

et al., 2024b). Similar convexity requirements also exist in electro- or magneto-elasticity and were successfully implemented into NN-based constitutive models, see Klein et al. (2022a); Kalina et al. (2024). However, it should be noted that ICNNs were also observed to be overly restrictive in some cases, as the conditions on the network architecture are sufficient but not necessary to ensure convexity, see Kalina et al. (2024); Klein et al. (2024).

Furthermore, when material models are to be equipped with additional input parameters, such as material parameters or microstructure descriptors, they should be polyconvex in terms of $\boldsymbol{F}$, $\boldsymbol{H}$, and $J$, but may have no functional requirements regarding the additional features, or be monotonously increasing or decreasing, or concave in those. Also, this prior knowledge can be incorporated into NN architectures, see also Amos et al. (2017) for general formulations of partially input-convex NNs and Klein et al. (2023) for the application to hyperelastic material models. In particular, thermoelastic material models of the form $\Psi = \Psi(\boldsymbol{F}, T)$ require polyconvexity in $\boldsymbol{F}$ but concavity in the temperature $T$, which has also been successfully realized using convex-concave NN architectures in Fuhg et al. (2024b). Furthermore, also GSM models, see equation (48), require polyconvexity of $\Psi(\boldsymbol{F}, \boldsymbol{\gamma})$ in $\boldsymbol{F}$ and convexity of $\Phi(\boldsymbol{F}, \boldsymbol{\gamma}, \dot{\boldsymbol{\gamma}})$ in $\dot{\boldsymbol{\gamma}}$, which can be realized by partially convex NN architectures, see Rosenkranz et al. (2023); Holthusen et al. (2023).

In summary, material modeling is subject to several types of invariance requirements such as thermodynamic consistency, rotational invariances, and functional requirements. It has already been demonstrated through various techniques that this prior knowledge can be successfully incorporated into ML frameworks for constitutive modeling. In particular, ML models can be fully physics-augmented, i.e., the requirements can be embedded into the model architectures by careful choice of features, outputs, and NN structure. This additional structure generally greatly improves the model performance, generalization, robustness, and interpretability and reduces data requirements.

**Invariances in dynamic systems and neural operators.** Symmetries and energy conservation are also important aspects of the modeling of dynamic systems. As already discussed in section 3.2, the equations of motion of a dynamical system can be formulated in an energy-conserving, i.e., thermodynamically consistent, fashion by deriving them through the Lagrangian or Hamiltonian formalisms (Lutter and Peters, 2023; Cranmer et al., 2020b; Greydanus et al., 2019). In these approaches, the invariance is *model-based* and incorporated through the model conception as the (Lagrangian $\mathcal{L}$, Hamiltonian $\mathcal{H}$, ...) potentials are learned/parameterized by a NN – and not the dynamics themselves in the form of an ODE, e.g., through a neural ODE, see section 3.1. This concept has also been applied to Koopman operators (section 3.5), for the data-based discovery of conservation laws in Kaiser et al. (2018) and could be further extended to neural operators. Likewise, enforcing conservation laws can also be achieved in the context of neural solution fields (Richter-Powell et al., 2022), c.f. section 3.3, neural operators (Liu et al., 2023b; 2024b), c.f. section 3.4, or neural processes (Hansen et al., 2023), c.f. section 4.3.

However, energy conservation is usually an idealization of real system behavior, and dissipative effects are present in most real-world systems. Dissipation behavior can be considered in a similar fashion through application of formalisms such as the Port-Hamiltonian (Desai et al., 2021; Neary and Topcu, 2023; Roth et al., 2025), see equation (16), pseudo-Hamiltonian (Eidnes et al., 2023), or GENERIC (General Equation for the Non-Equilibrium Reversible–Irreversible Coupling) frameworks (González et al., 2019; Hernández et al., 2021), which is structurally very similar to the GSM framework mentioned above for material modeling.

Further prior knowledge about the model structure can be incorporated in these frameworks, e.g., by ensuring convexity or a quadratic positive definite form of sub-potentials such as the kinetic energy component of the Lagrangian, see Lutter and Peters (2023), or the skew-symmetry of the cosymplectic matrix and the symmetry and positive semi-definiteness of the friction matrix in the GENERIC framework, see Hernández et al. (2021; 2022). For dissipative systems, this further structure is crucial in ensuring the thermodynamic consistency of the models, and the GENERIC formalism even requires the fulfillment of an additional consistency condition, which is embedded by adding a further penalty-type loss term, c.f. equation (19), in Hernández et al. (2021). Furthermore, it should be mentioned that preserving the thermodynamic consistency of (classical or ML-based) models also requires suitable, tailored time integration schemes for the resulting ODEs, see section 2.1, which as already realized for the GENERIC framework in Hernández et al. (2021; 2022).

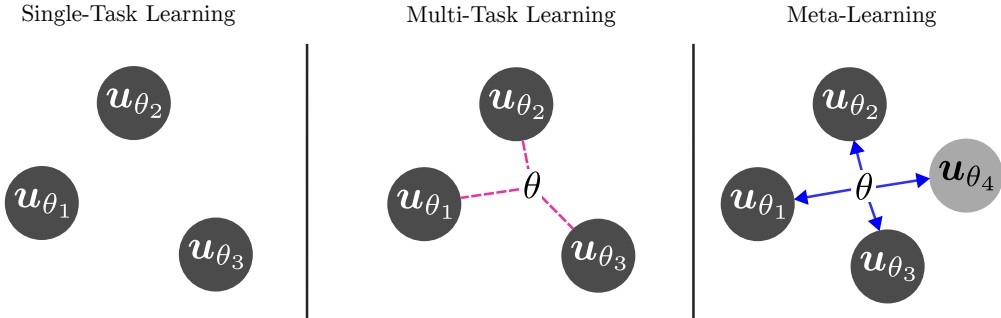

Figure 6: Visual illustration of the single-task learning, multi-task learning (MTL), and meta-learning paradigms for PDEs. In single-task learning, different models are being learned separately for each PDE. On the other hand, MTL learns a set of PDEs concurrently, hence finding a set of parameters $\boldsymbol{\theta}$ that compromises the single-task solutions and generalizes across them. Finally, meta-learning aims for a general solution that can be transferred later for novel PDEs (e.g., $\boldsymbol{u_{\theta_4}}$) where a few gradient steps can be taken to arrive by the single-task models.

As already mentioned in section 3.4, neural operator architectures have also already been extended to be equivariant to rotations, translations, and reflections in Helwig et al. (2023) by applying group convolutions similar to equation (52), or explicitly being $SE(3)$-equivariant in Cheng and Peng (2023), as well as also being momentum conserving in (Liu et al., 2023b). Furthermore, PINNs can be enhanced to approximate the symmetries of PDEs using additional loss terms, see Akhound-Sadegh et al. (2023), and neural processes, which will be discussed in detail in section 4.3, can be designed to be invariant to translations, rotations, and reflections, as demonstrated by Holderrieth et al. (2021) also using a group theoretic approach.

Lastly, it should be mentioned that these concepts for employing invariances as prior knowledge for machine learning can be applied to a wide range of problems, e.g., in material science or chemistry. For instance, Batzner et al. (2022) developed a potential-based and thus energy-conserving, as well as $E(3)$-invariant/equivariant graph-convolution NN for learning interatomic potentials from ab-initio calculations for molecular dynamics simulations. Furthermore, Döppel and Votsmeier (2023) enhanced the chemical reaction neural networks (CRNNs) for autonomous mechanism discovery, which is a neural ODE-type model, with stoichiometric constraints that ensure mass conservation. Moreover, in these cases, informing the NNs with invariances drastically reduced training data requirements, accelerated training, and increased accuracy and robustness compared to previous state-of-the-art ML models.

## 4 Data-Driven Priors from Experiments

While section 3 and 3.7 focus on designing models with some form of prior knowledge from physical, mathematical, and numerical modeling, data and the learning process itself can also be leveraged as a form of prior knowledge. Meta- and multi-task learning are two learning paradigms that exploit knowledge gained from previous or concurrent experiments (Figure 6). Multi-task learning (Caruana, 1997) aims to use the knowledge gained from each model (or 'task') to improve the model through generalization, while meta-learning (Vilalta and Drissi, 2002) seeks to improve the learning algorithm instead, either by optimizing the model initialization or how the model updates during learning. Neural processes (Garnelo et al., 2018) combine ideas from meta- and multi-task learning by conditioning the model on a small 'contextual' dataset, which facilitates fast adaption to different settings. Since forecasting typically considers a range of domains in practice, such as boundary conditions and variations in physical parameters, there is ample opportunity to leverage knowledge from multiple experiments across these different domains to improve performance.

### 4.1 Multi-task Learning

System identification typically considers learning a model for a single system under study. However, in practice, there may be a fleet of systems of interest. In reality, these systems will differ in behavior and may have different operating regimes, which warrants training separate models. Multi-task learning (MTL) attempts to learn these models jointly to reduce the measurement burden. By balancing the need to model each system while acknowledging the similarity between systems, the aim is that the chore of learning multiple models can, in fact, be leveraged to improve sample efficiency and performance by using the shared properties across the physical systems.

Following section 2, system identification can be extended to the MTL setting when the objective is to learn more dynamical systems at once. A dataset of $N$ input-output sequences $\mathcal{D} = \{(\boldsymbol{w}_{1:T_i}^{(i)}, \boldsymbol{y}_{1:T_i}^{(i)})\}_{i=1}^N$, where $T_i$ is the length of sequence $i$, and each sequence $i$ is defined by a distinct dynamical system with state variables $\boldsymbol{x}_t^{(i)} \in \mathcal{X}$, $t = 1, ..., T_i$ for each $i = 1, ..., N$ with $\boldsymbol{x}_0^{(i)}$ being an initial state variable. Each dynamical system can be assumed as a different but related task.

A naive MTL approach for learning a common dynamical system is to exploit task information (e.g., task ID, task description) in the learning process. There are two ways to integrate such information: initial state customization or bias customization. For initial state customization, task information $\tau$ is used to customize the initial state $\boldsymbol{x}_0$. On the other hand, the task information can be deeply integrated by being appended to the inputs $\tilde{\boldsymbol{w}}_t \leftarrow [\boldsymbol{w}_t^\intercal, \tau^\intercal]^\intercal$ for each time-step $t$.

Spieckermann et al. (2015) use RNNs for multi-task system identification by structuring some weights per task, referred to as tensor factorization. By considering similar system identification tasks with similar task information, Spieckermann et al. (2015) use an indicator variable $\mathcal{I} = \{i_0, \dots, \}, i_k \in \mathbb{Z}^+, |\mathcal{I}| = N$ for $N$ tasks. These indicator variables are added to the dataset and used in the RNN architecture to index the parameter weights, such as linear weights or bias terms. This work explores different approaches to factorizing the parameters to trade off the number of parameters learned per task and across tasks.

On the other hand, Bird et al. (2022) proposes a probabilistic approach, called multi-task dynamical system (MTDS), for extending MTL to time series models. MTDS learns a set of hierarchical latent variables for modeling dynamical systems of different sequences. This model allows for adapting dynamical systems to the inter-sequence variations, hence enabling personalization or customization of the models. MTDS assumes a different parameterization $\boldsymbol{\theta}^{(i)} \in \Theta$, which depends on the hierarchical latent variable $\boldsymbol{z}^{(i)} \in \mathcal{Z}$, for each sequence $i$ generated from a specific dynamical system. MTDS is defined mathematically by the following equations,

$$\boldsymbol{\theta}^{(i)} = \boldsymbol{h}_{\boldsymbol{\phi}}(\boldsymbol{z}^{(i)}), \quad \boldsymbol{z}^{(i)} \sim p(\boldsymbol{z}), \tag{53}$$

$$\boldsymbol{x}_t^{(i)} \sim p(\boldsymbol{x} \mid \boldsymbol{x}_{t-1}^{(i)}, \boldsymbol{w}_t^{(i)}, \boldsymbol{\theta}^{(i)}), \tag{54}$$

$$\boldsymbol{y}_t^{(i)} \sim p(\boldsymbol{y} \mid \boldsymbol{x}_t^{(i)}, \boldsymbol{w}_t^{(i)}, \boldsymbol{\theta}^{(i)}), \tag{55}$$

where $\boldsymbol{h}_{\boldsymbol{\phi}} : \mathcal{Z} \to \mathbb{R}^d$ is vector-valued function that maps a latent variable $\boldsymbol{z}^{(i)}$ to a model parameter $\boldsymbol{\theta}^{(i)} \in \mathbb{R}^d$. MTDS explicitly provides visibility of the task specialization, which can be controlled directly. MTDS is evaluated on two applications: motion-capture data of people walking in various styles using a multi-task recurrent neural network and patient drug-response data using a multi-task pharmacodynamic system.

PINNs suffer from issues related to varying initial conditions, boundary conditions, and domains due to the trade-off between generalization vs. specificity. Pellegrin et al. (2022) proposed the use of transfer learning for increasing the generalization, where a base neural network with different "heads" for various initial and boundary conditions is initially trained. After this initial training, they kept the base network unchanged and fine-tuned the heads for new conditions. This method allows the base network to retain a general understanding of the physical systems, thus reducing training time for new scenarios because the base does not need retraining.

Flamant et al. (2020) tried to enhance PINNs across different scenarios by taking PINN models as solution bundles, which is a collection of solutions for the same PDE under various conditions. They adjust the model's loss function by adding more weight to errors at the initial state. This is beneficial because it

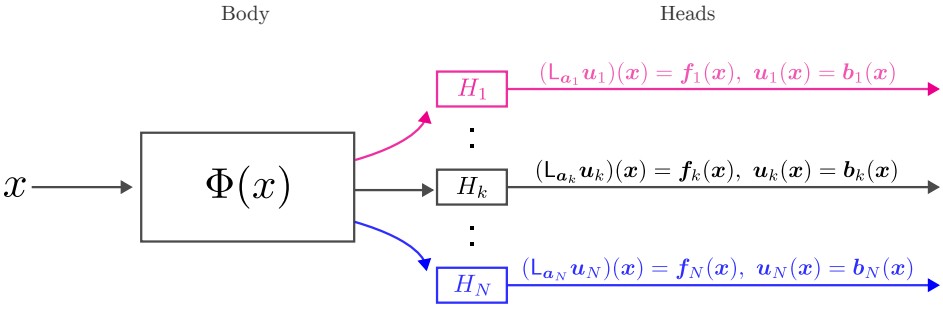

Figure 7: A schematic of the multi-head physics-informed neural networks (MH-PINNs) from Zou and Karniadakis (2023), where a body $\Phi$ is shared by $N$ Tasks while having task-specific heads $H_k$, $k = 1, ..., N$.

focuses more on initial error reduction that prevents them from propagating and amplifying. Therefore, the network learns multiple solutions simultaneously for variable conditions and shows good convergence properties.

Thanasutives et al. (2021) propose an MTL approach of solving the original PDE along with an auxiliary related PDE with different coefficients. In this work, they integrate MTL at the architecture and optimization levels. For the architecture, *cross-stitch* (Misra et al., 2016) is employed to share information between two parallel baseline networks for original and auxiliary tasks. At each layer, a cross-stitch unit is used to linearly combine the input activations from the two networks to allow flexible sharing of representations between the two branches. The authors evaluate two MTL techniques for learning, an *uncertainty-weighting* loss (Kendall et al., 2017) and *gradient surgery* (Yu et al., 2020). The uncertainty-weighted physics-informed loss function (UNCERT) is defined as a combination of task-specific PINNs loss functions weighted by the uncertainty of each PDE solution. The loss function is defined as

$$\mathcal{L}_{\text{UNCERT}}(\boldsymbol{\theta}, \boldsymbol{\sigma}) = \sum_{i=1}^{N} \left( \frac{1}{2\sigma_i^2} \mathcal{L}_{\text{PINN}}(\boldsymbol{\theta}_i) + \log \sigma_i \right), \tag{56}$$

where $N$ refers to the number of tasks, $\sigma_i$ is a gradient-based trainable parameter that indicates the uncertainty of each PDE solution, and $\mathcal{L}_{\text{PINN}}$ is a similar loss function as in equation (19). "Gradient surgery" is implemented using PCGrad (Yu et al., 2020) to modify the average of the unweighted PINNs loss functions for mitigating conflicting gradients between tasks. PCGrad projects the loss gradient of task $i$ onto the normal vector of task $j$'s loss gradient and vice versa. The model parameters are updated using $\delta_{\text{PC}}\boldsymbol{\theta}$ shown below,

$$\delta_{\text{PC}}\boldsymbol{\theta} = \sum_{i=1}^{N} \boldsymbol{g}_{\text{PC}}^{i}, \quad \{\boldsymbol{g}_{\text{PC}}^{i}\} = \text{PCGrad}(\{\nabla_{\boldsymbol{\theta}} \mathcal{L}_{\text{PINN}}(\boldsymbol{\theta}_i)\}),$$

where $\boldsymbol{g}_{\text{PC}}^{i}$ is the modified gradient of task $i$ after applying PCGrad. The proposed algorithm provides a better generalization on highly nonlinear domains in terms of lower error rates on unseen data points in comparison to related baselines.

Zou and Karniadakis (2023) propose a new PINN architecture that suites MTL. In this work, a multi-head architecture is presented, named multi-head physics-informed neural network (MH-PINN), where a body $\Phi$ is shared by a collection of tasks (e.g., ODEs/ PDEs) while task-specific heads $H_k$, $k = 1, ..., N$ are learned separately for each model. The body consists of nonlinear hidden layers, while linear layers are used for the different heads. Consequently, the body provides a set of basis functions for each head to generate task-specific solutions. The body $\Phi : \mathbb{R}^{d_x} \to \mathbb{R}^d$ is a function parameterized with parameter $\boldsymbol{\theta}$, and each head $H_k$ is a linear function. The task-specific solution is defined as $\hat{\boldsymbol{u}}_k(\boldsymbol{x}) = H_k \circ \Phi(x), \forall x \in \Omega$. Given an MTL dataset $\{\mathcal{D}_k\}_{k=1}^N$, the MTL loss function is defined as follows:

$$\mathcal{L}_{\text{MH-PINN}}(\{\mathcal{D}_k\}_{k=1}^N; \boldsymbol{\theta}, \{H_k\}_{k=1}^N) = \frac{1}{N} \sum_{k=1}^{N} \mathcal{L}_k(\mathcal{D}_k; \boldsymbol{\theta}, H_k), \tag{57}$$

where $\mathcal{L}_k$ is the PINN loss function as in equation (19). Moreover, Zou and Karniadakis (2023) makes use of the learned model to construct prior knowledge for downstream tasks. For instance, the heads are considered as samples used for estimating the PDF and a generator for $H$ using normalizing flows (NFs) (Rezende and Mohamed, 2015). The acquired prior knowledge is useful for downstream tasks where limited data points are available.

Similarly, Wang et al. (2023) ease the training of PINNs by employing a multi-task optimization paradigm where multiple auxiliary tasks are solved together with a main task. Knowledge transfer from one task facilitates solving others. The problem of automatic traffic modeling is considered, which has multiple modalities such as traffic speed and density. Given several datasets measuring different traffic scenarios and MTL over datasets and modalities, a similar PINN architecture, as in Zou and Karniadakis (2023), is trained with a shared body network and only a final layer per task. This architecture design significantly increases the data available to train the initial PINN layers, which build an internal feature representation. The MTL ensures that this feature representation is useful across traffic conditions and modalities. As a drawback, the model requires an adaptive training scheme in order for the multi-task aspect to be effective, which significantly increases the implementation complexity.

### 4.2 Meta Learning

Whereas multi-task learning aims to exploit common knowledge while learning a given set of tasks in parallel, meta learning aims to optimize a learning procedure on the given tasks that can later be applied to novel tasks. Both paradigms can leverage a set of source tasks to learn better solutions on target tasks and are therefore closely related to transfer learning (Zhuang et al., 2021b). However, whereas multi-task learning optimizes source and target tasks jointly, meta learning considers the source tasks only during pre-training allowing for computationally efficient learning on the target tasks. In general, the result of meta-learning could be a complete algorithm. However, the methods discussed in the following are content with learning suitable initial parameters or other hyperparameters for given algorithms.

Meta learning methods for PDEs are often specific to a certain type of PDE but can adapt to different instances, such as different boundary conditions or different coefficients in the governing equations. We can distinguish between meta learning methods in which the network has an additional input to inform it about the particular instance and those in which adaptation happens solely via fine-tuning of the weights.

The latter class of methods is typically based on 'model agnostic' meta learning (MAML) (Finn et al., 2017), its first-order approximation FOMAML, or closely related meta learning techniques, such as REP-TILE (Nichol et al., 2018). MAML optimizes initial model parameters to minimize the expected test loss on the given set of tasks, which is achieved by optimizing the tasks on a training set, e.g., using a small number of gradient updates. FOMAML and Reptile are computationally more efficient than MAML by discarding higher-order information, and thereby avoiding backpropagating through the inner optimization. Despite using less accurate meta-gradients, both methods have been shown to perform similarly well compared to MAML (Finn et al., 2017; Nichol et al., 2018). A comparison between MAML and REPTILE is shown in Algorithm 1.

Several works have applied these techniques for learning dynamics models (Lin et al., 2020; Qin et al., 2022). Meta-L (Lin et al., 2020) uses MAML for linear time-variant (LTV) systems. The model is optimized to achieve a small prediction error after updating the linear parameters of the dynamics with a single gradient step. During deployment, the fast adaptation makes it possible to adapt to changes of the dynamics online. MetaPDE (Qin et al., 2022) employs a similar approach to nonlinear dynamics. They compare MAML with a slightly different meta learning technique, LEAP (Flennerhag et al., 2019), for meta learning a PINN that can be quickly adapted to different PDE parameters. While LEAP was faster during meta-learning, MAML performed faster in deployment. NRPINN (Liu et al., 2022) is a similar approach that uses REPTILE (Nichol et al., 2018) on a modified PINN loss, which contains an additional term that can make use of labeled data, if available.

Another MAML-based approach is used by iMODE (Li et al., 2023c), which adapts a latent input to a more general PINN that is shared among different PDE instances. A bi-level optimization problem is solved to optimize both, the common PINN network and the initialization of its latent input. Furthermore, PCA is

---

**Algorithm 1** Meta learning with MAML and Reptile

---

**Require:** initial parameters $\boldsymbol{\theta}^{(0)}$, task distribution $p_{\text{task}}(\boldsymbol{\tau})$, task loss function $\mathcal{L}(\boldsymbol{\theta}, \boldsymbol{\tau})$, task batch size $N_{\boldsymbol{\tau}}$, number of iterations $N_{\text{iter}}$ number of gradient steps $K$, task learning rate $\alpha$, meta learning rate $\beta$.

 1: **for** $i = 1, 2, \ldots, N_{\text{iter}}$ **do**
 2:     sample batch of tasks $\boldsymbol{\tau}_1, \ldots, \boldsymbol{\tau}_{N_{\boldsymbol{\tau}}}$ from $p_{\text{task}}(\boldsymbol{\tau})$
 3:     **for** $j = 1, 2, \ldots, N_{\boldsymbol{\tau}}$ **do**
 4:         $\boldsymbol{\theta}_j^{(0)} \leftarrow \boldsymbol{\theta}^{(i-1)}$
 5:         **for** $k = 1, 2, \ldots, K$ **do**
 6:             $\boldsymbol{\theta}_j^{(k)} \leftarrow \boldsymbol{\theta}_j^{(k-1)} - \alpha \frac{\partial}{\partial \boldsymbol{\theta}} \mathcal{L}(\boldsymbol{\theta}_j^{(k-1)}, \boldsymbol{\tau}_j)$
 7:         **end for**
 8:     **end for**
 9:     $\boldsymbol{\theta}^{(i)} \leftarrow \boldsymbol{\theta}^{(i-1)} - \beta \nabla_{\boldsymbol{\theta}^{(i-1)}} \frac{1}{N_{\boldsymbol{\tau}}} \sum_{j=1}^{N_{\boldsymbol{\tau}}} \mathcal{L}(\boldsymbol{\theta}_j^{(K)}, \boldsymbol{\tau}_j)$ `// MAML`
10:     $\boldsymbol{\theta}^{(i)} \leftarrow \boldsymbol{\theta}^{(i-1)} - \beta \frac{1}{N_{\boldsymbol{\tau}}} \sum_{j=1}^{N_{\boldsymbol{\tau}}} \left[ \boldsymbol{\theta}_j^{(K)} - \boldsymbol{\theta}^{(i-1)} \right]$ `// REPTILE`
11: **end for**

---

applied to learn a compressed form of the latent input. When examples with labeled PDE parameters are available during training, iMode can learn a diffeomorphism between the labeled parameters and the compressed latent inputs, which can be used for system identification. DyAd (Wang et al., 2022d) directly feeds the parameters of the PDE as input to the model. The architecture is geared towards 2D images/velocity fields. It uses 3D convolutions and AdaPad, which predicts boundary conditions using a nonlinear transformation of the latent used for padding the images. As the PDE parameters are often unknown, DyAd also trains an encoder network that predicts the parameters from a history of states. The encoder is trained using weak supervision, e.g., by exploiting when a subset of the parameters are known.

Instead of predicting latent inputs, hypernetworks can be used to predict the weights of the common PINN. For example, HyperPINN (de Avila Belbute-Peres et al., 2021) trains a hypernetwork that takes the parameters of a given class of PDEs (e.g., Burger's equation) as input and outputs the parameters of a PINN. To reduce the number of the parameters of the PINN, Hyper-LR-PINNs (Cho et al., 2023b) restrict the weight matrices of the neural network to be low-rank, parameterized by the matrices of an SVD decomposition, that is, the weight matrix of layer $i$ is given by $\mathbf{W}_i = \mathbf{U}_i \boldsymbol{\Sigma}_i \mathbf{V}_i^\top$, with rectangular matrices $\mathbf{U}_i$ and $\mathbf{V}_i$, and a diagonal matrix $\boldsymbol{\Sigma}_i$. During offline pre-training, $\mathbf{U}$ and $\mathbf{V}$ are optimized directly along with the hypernetwork that predicts the elements of $\boldsymbol{\Sigma}$. During deployment, only the diagonal elements are optimized, using the hypernetwork for predicting their initial values. An alternative way to reduce the number of environment-specific parameters was proposed by Kirchmeyer et al. (2022). Their method, CoDA, learns shared nominal parameters of the dynamics model along with a shared linear projection matrix that maps environment-specific low-dimensional parameters to an offset for the nominal parameters. By penalizing the $\ell_1$ or $\ell_2$ norms of the projection matrix and the environment-specific parameters, CoDA aims to concentrate the resulting offset parameters around the nominal parameters for faster optimization during deployment.

While the aforementioned methods were targeted towards a specific type of PDEs, some approaches aim to learn even more general networks. LEADS (Yin et al., 2021) learns a dynamics model, where the dynamics may also change over time. The model predicts the state derivatives and decomposes them into shared and environment-specific components. It is trained by minimizing the squared error between predicted derivatives and targets, with an additional convex regularizer penalizing the complexity of the environment-specific terms. Different regularizers are proposed for the linear and nonlinear cases, based on sample complexity analysis

Iwata et al. (2023) consider PDEs with polynomial governing equations. A single PINN is trained, which takes the query point and the coefficients of the PDE as input. Furthermore, Dirichlet boundary conditions are represented by a set of boundary points and their evaluations. The PINN is optimized with respect to the PINN-error, using self-generated data from randomly sampled PDEs. However, to keep the number of coefficients small, the experiments are limited to two-dimensional PDEs of order 2 and use polynomials of degree 2.

Figure 8:    Neural processes compress a context dataset $\mathcal{C} = \{\boldsymbol{x}_c, \boldsymbol{y}_c\}_{c=1}^{C}$ in an invariant fashion into a compact representation by an encoder. This context is then used during prediction to adapt the model appropriately given the context dataset. For conditional models, this representation is deterministic, but for latent neural processes, this context representation is a distribution that is sampled from at prediction time.

The aforementioned methods can be used to obtain initial model parameters that can serve as a starting point for fine-tuning. This fine-tuning is crucial for methods that learn a single initialization for all tasks, such as the MAML-based approaches. However, we also discussed methods that learn hyper-networks to predict initialization based on a given task specification  (de Avila Belbute-Peres et al., 2021; Cho et al., 2023b; Kirchmeyer et al., 2022) . By using these methods without fine-tuning, we can construct zero-shot methods, that is, we can avoid training a new model when given a new PDE. In the next section, we consider neural processes, which are data-driven models that are particularly suitable for such fast adaptations.

### 4.3   Neural Processes

Neural processes (Garnelo et al., 2018) are probabilistic models over functions that are conditioned on a context represented by a set of $C$ measurements, enabling the model to adapt to relevant data,

$$\boldsymbol{f} \sim q_{\boldsymbol{\theta}}(\cdot \mid \boldsymbol{x}, \mathcal{C}), \quad \mathcal{C} = \{\boldsymbol{x}_i, \boldsymbol{y}_i\}_{i=1}^{C}, \quad \boldsymbol{f} : \mathcal{X} \to \mathcal{Y}. \tag{58}$$

Constructing a model this way has many applications, and covers several domains discussed earlier. Deep operator networks also have a context as input, corresponding to the parameters of the PDE being modeled. Multi-task learning can also be designed to identify tasks through an auxillary context input, and meta learning considers rapid adaptation to a new task, which is captured here by the context representing a small dataset. The idea of inference over functions was also covered as part of PINNs and neural operators, as mesh-free modeling techniques using function approximators provide flexible predictions and can work with more unstructured datasets. In this section, we will discuss how neural networks can be used to design these models and how they have been applied to modeling physical systems from data. For a more comprehensive review of these models, see Jha et al. (2022).

**Architectures.**   As a latent variable model, with latent variable $\boldsymbol{z}$ inferring from the contextual data, we discuss the architectural design decision of neural processes along three axes: the encoder, the decoder, and whether $\boldsymbol{z}$ is modeled deterministically or stochastically. The last point is required to differentiate conditional- and latent NPs respectively (Garnelo et al., 2018). The trade-off here is that conditional NPs use simpler, deterministic context representations, whereas a latent NP uses a random variable for the context and, therefore, represents a richer probabilistic model but is burdened with requiring approximate inference during learning. The objective is now an amortized evidence lower bound (ELBO), which requires sampling from the encoded latent distribution and stochastic optimization using reparameterized gradients (Kingma and Welling, 2014).

$$\max_{\boldsymbol{\theta}} \mathbb{E}_{\boldsymbol{z} \sim q_{\boldsymbol{\theta}}(\cdot \mid \mathcal{C}, \boldsymbol{\theta})}[\log p(\mathcal{D} \mid \boldsymbol{z})] - \mathbb{D}_{\mathrm{KL}}[q_{\boldsymbol{\theta}}(\boldsymbol{z} \mid \mathcal{C}) \mid\mid p(\boldsymbol{z})]. \tag{59}$$

This approach benefits the model by allowing it to capture ambiguity in the approximate process induced by the finite context set of measurements. As a result, the decoder can focus on modeling statistical ('aleatoric') uncertainty while the encoder's latent variable captures model-based ('epistemic') uncertainty. This separation is important for uncertainty quantification when there is limited training data and over-parameterized models, as well as for statistical decision-making such as active learning, where the epistemic uncertainty is used actively to reduce model uncertainty.

Beyond the latent variable assumption discussed above, a major component of the encoder is implementing a set-based model that is invariant to the ordering of the context points. The implementation details of this aspect go beyond the remit of this survey, so we refer readers to sections 2 and 3 of Jha et al. (2022) for a comprehensive discussion, but on a high-level, many NP architectures learn a parametric feature space $\phi$ to transform the measurement pair $\{\boldsymbol{x}_c, \boldsymbol{y}_c\}$ into a combined representation $\boldsymbol{r}_c$, and then these representations are combined with a query point $\boldsymbol{x}_*$ in a set-invariant fashion with respect to $\mathcal{C}$. Examples of set-invariant operations include averaging the representations into a single vector (Garnelo et al., 2018), or averaging over function evaluations (Murphy et al., 2018), such as convolution- (Gordon et al., 2019) and attention-based (Nguyen and Grover, 2022) operations. These representations are used directly by the conditional NP, while for the latent NP these representations are used for the amortized variational posterior shown in equation (59). These architectures are shown in Figure 8. The decoder model has more flexibility in incorporating useful inductive biases from deep learning, such as deconvolutions (Garnelo et al., 2018), recurrent architectures (Willi et al., 2019), and transformers (Nguyen and Grover, 2022).

**Applications.**   While a relatively new architecture, NPs have been used for several prediction applications, especially for settings that have a spatial consideration that should be incorporated to improve performance.

Holderrieth et al. (2021) introduce 'steerable' conditional neural processes (CNP), which are motivated to model scalar- and vector-fields, such as temperature readings and wind maps. In this setting, a relevant inductive bias is invariant to translation, rotation, and reflection, so that the model is invariant to any innocuous spatial perturbation of the data. NPs are well suited for this setting, and the authors use steerable convolutional layers in the encoder and decoder architecture to capture this invariance, as steerable feature maps have the desired invariances. The authors demonstrate that this model can effectively model measured wind direction vector fields from weather data, and the incorporated invariances improve generalization.

Vaughan et al. (2022) use a ConvCNP for downscaling climate data. Rather than downsample from a high to low resolution grid using an interpolation strategy, NPs are flexible enough to be evaluated at arbitrary locations, and can learn their interpolation behaviour in a data-driven fashion. They show this approach outperforms an interpolation-based baseline and show the NPs continuous nature is well suited for estimating maximum and minimum values.

One application of Bayesian prediction models is active learning (Cohn et al., 1996), where the model selects samples for its dataset to accelerate learning. This is relevant in settings where collecting data is expensive. One example of this is weather monitoring, where environmental sensors must be installed to monitor the weather, and it is desirable to place sensors optimally to improve the quality of the dataset and minimize hardware costs. Andersson et al. (2023) use a Gaussian ConvNP for active sensor placement. Gaussian ConvNPs are effective here because they are scalable models with tractable uncertainty quantification and can learn the non-stationary spatial and temporal phenomena seen in weather systems through the context set.

Scholz et al. (2023) consider the issue of training NP models on highly processed data, such as reanalysis data in weather forecasting, discussed in section 3.5. This data is close to ideal, e.g., spatially uniform, and smoothed with physical models, but there is a 'reality gap' between it and raw measurement data. The authors use ConvCNP to train the model first on reanalysis data, and then fine-tune on raw data. The fine-tuned model outperforms variants trained on only reanalysis or raw data

Hansen et al. (2023) consider the task of combining the data-driven NP with physical plausibility. They use an attentive NP as a solution prior, which provides an initial estimate of the solution through a multivariate Gaussian predictive distribution. To incorporate a soft physics-based  conservation  law constraint, they consider a linear constraint PDE of the form $\mathsf{L}\, u(t, x) = \int_{\mathcal{X}} u(t, x)\mathrm{d}x = b(t)$, which can be approximated using discretization, which essentially translates obeying the soft physics constraint penalty into solving a regression problem subject to a weighting hyperparameter of the penalty. Note that this soft constraint becomes hard at the limit of the penalty weighting hyperparameter going to infinity. By adopting a multivariate Gaussian solution prior, this prior can be conditioned on the constraint points to yield an updated distribution that is more physically plausible in closed form. The authors primarily consider the porous medium equation, which is used to model underground fluid flow, nonlinear heat transfer, and water desalination. This PDE

can incorporate challenging solution constraints and discontinuities, which is why the solution's physical plausibility is essential. The authors verify that the constraint correction step improves the solution towards the ground truth value.

## 5 Discussion on industrial applications

From the scale of this survey, it is clear that the combination of machine learning with physics knowledge is a rapidly expanding field, with many researchers across disciplines hoping to see the scale of progress seen in machine learning domains such as computer vision and language modeling transfer to the physical sciences. Various industrial applications of the methods described in this review paper have been reported in scientific literature. Besides more exotic applications such as utilizing physics-informed losses as accident scenario simulation in nuclear power plants (Antonello et al., 2023), utilizing meta learning to forecast industrial sales (Kück et al., 2016), predictive maintenance of batteries (Wen et al., 2023) or using physics-informed losses for non-intrusive power load monitoring (Huang et al., 2022b). Additionally, we have identified several clusters of industrial applications, as detailed in Table 3, which lists relevant publications.

In sectors like manufacturing, anomaly detection by utilizing meta-learning (Garouani et al., 2021a) or physics-informed losses (Kim et al., 2022) forms a cluster of industrial applications. Another field of industrial application is the prediction and control of additive manufacturing processes with physics-informed losses (e.g. Kapusuzoglu and Mahadevan (2020), Zhu et al. (2021), Nguyen et al. (2022), Würth et al. (2023), Zhang et al. (2022b)). Guo et al. (2022) provide a thorough review of physics-informed ML applications in additive manufacturing from product design over process planning to fabrication quality control.

In the automobile industry, physics knowledge is commonly used for predictive maintenance (Wen et al. (2023)) and system identification (He et al., 2020; Nath et al., 2023). Using machine learning techniques, such as system identification using physics-informed neural networks (Grimm et al., 2021; Zhang et al., 2023) and operator learning for protein structure prediction (Jumper et al., 2021), has made significant advancements in the field of epidemiology and medical industries.

Industry sectors such as power (Zideh and Solanki, 2023; Hu et al., 2021; Antonello et al., 2023) and thermal (Yan et al., 2024; Cai et al., 2021c; Daw et al., 2019) have also widely incorporated the physics knowledge in machine learning for anomaly detection and parameterizing equations. Furthermore, physics-based machine learning has also found its application in robotics (Sanyal and Roy, 2023; Nicodemus et al., 2022; Huang et al., 2022b; Kamranfar et al., 2021) where PINNs have been used to control quadrotors and multi-link manipulators. In optics, PINNs have also been used for system identification (Lu et al., 2021; Wang et al., 2020).

In the process industry (e.g., chemical engineering, oil and gas processing, water treatment) physics-informed lossess have been used to predict process signals (Asrav et al., 2023; Franklin et al., 2022; Ge and Chen, 2017). Failure detection was approached using meta-learning (Gao et al., 2024) and latent variable models (Kong and Ge, 2021)). Finally, physics-informed losses have been used to create soft or virtual sensors in fluid dynamical settings in oil processing (Du et al., 2023; Franklin et al., 2022). The process control industry also utilizes physics-informed losses for modeling oil wells (Kittelsen et al., 2024) and water tanks (Antonelo et al., 2024).

Across industry segments, a number of use cases gravitate around the prediction of wear in mechanical systems using meta-learning approaches (Chen et al., 2023b; Kamranfar et al., 2021; Li et al., 2021c; Hu et al., 2021) as well as physics-informed losses (Hua et al., 2023).

Despite the promising applications, it is still uncertain whether these methods will replace traditional numerical approaches. In data-rich environments like weather forecasting, ML has demonstrated significant progress (Lam et al., 2023; Kurth et al., 2023). However, the real challenges of applying machine learning in industrial settings, such as process industries or manufacturing, lie in their distinct requirements compared to those in the consumer market (Hoffmann et al., 2021a). Industrial settings often deal with sparse data, and operators may be skeptical of black-box systems due to their opacity.

| Industry | Method | Study |
|---|---|---|
| Aerospace | Operator learning | Perrusquía et al. (2024) |
| Mechanical | Parameterizing equations | Flamant et al. (2020) |
| Additive manufacturing | System identification | Würth et al. (2023); Zhang et al. (2022b); Guo et al. (2022); Nguyen et al. (2022) |
|  | Anomaly detection | Chen et al. (2023b); Garouani et al. (2021a); Kim et al. (2022) |
|  | Industrial data science | Garouani et al. (2021b; 2022) |
|  | Physics-informed losses | Hua et al. (2023); Kapusuzoglu and Mahadevan (2020) |
| Automobile | Physics-informed losss | Wen et al. (2023) |
|  | System identification | He et al. (2020); Nath et al. (2023) |
| Epidemiology | System identification | Grimm et al. (2021); Zhang et al. (2023) |
|  | Operator learning | Jumper et al. (2021) |
| Engines | Anomaly detection (w/ PINNs) | Cohen et al. (2023); Zgraggen et al. (2023) |
| Solar energy | Anomaly detection (w/ PINNs) | Zgraggen et al. (2023) |
| Fluid dynamics | Operator Learning | Falas et al. (2020); Wen et al. (2022) |
|  | System identification | Jin et al. (2021); Cai et al. (2021d); Rui et al. (2023) |
| Industrial robotics | Operator Learning | Sanyal and Roy (2023); Nicodemus et al. (2022) |
|  | Physics-informed losses | Huang et al. (2022b) |
|  | Predictive maintenance | Kamranfar et al. (2021) |
| Optics | System identification | Lu et al. (2021) |
|  | Physics-informed losses | Wang et al. (2020) |
| Power industry | Anomaly detection (w/ PINNs) | Zideh and Solanki (2023); Hu et al. (2021) |
|  | Physics-informed losses | Antonello et al. (2023) |
| Thermal processes | Parameterizing equations | Yan et al. (2024); Cai et al. (2021c) |
|  | Anomaly detection (w/ PINNs) | Daw et al. (2019) |
| Process control | Physics-informed losses | Antonelo et al. (2024); Kittelsen et al. (2024) |
| Process monitoring | Physics-informed losses | Du et al. (2023); Franklin et al. (2022); Asrav et al. (2023) |
|  | Anomaly detection | Gao et al. (2024); Kong and Ge (2021) |
|  | Latent variable model | Ge and Chen (2017) |
| Climate | System identification | Beucler et al. (2019) |
| Cross-segment | Anomaly detection | Li et al. (2021c) |
|  | System identification | Kück et al. (2016) |

Table 3: Industrial application cases of machine learning with physics knowledge described in literature.

These challenges may be well addressed by a combination of classical machine learning and understanding of the production process as well as knowledge of the physics of the production assets. Such approaches would also improve the explainability of these models, which is a major challenge in industry (Kotriwala et al., 2021). Furthermore, industrial solutions are usually embedded in cyber-physical systems. Hoffmann et al. (2021b) report that these pose additional challenges for various disciplines to work together. On the one hand, cyber-physical systems offer data of a physical asset as part of its "digital twin", which may ease the implementation of ML-based services. On the other hand, the entanglement of the physical and digital world also requires a better understanding and common terminology between developers from both worlds. Physics-informed machine learning stands poised to bridge this gap in the context of the industrial digital twin, tapping into substantial potential where both data and system knowledge are only partially available. In this survey, we focus on the prediction of the physical quantity of interest under the forward settings. However, many industrial applications, such as sensing, require solving a physical system under the inverse settings, namely estimating certain conditions from indirect information. The physics-informed framework will also play an important role in this regard because of its advantages in linking data and physics efficiently and robustness against data and system uncertainties. This dual capacity enhances both the applicability and reliability of machine learning in complex industrial environments.

# 6 Conclusion

This survey has covered the breadth of methods integrating machine learning with prior physics knowledge for predictive modeling, underscoring the potential to bridge the gap between traditional numerical methods and modern data-driven approaches. These methods aim to enhance reliability, robustness, and generalization out-of-distribution by leveraging scientific knowledge, particularly in environments with limited data. The various techniques surveyed, ranging from the physics-informed losses of PINNs to the purely data-driven neural processes, demonstrate the diverse ways in which prior knowledge can be encoded into machine learning models.

Significant progress has been made in domains such as weather forecasting (section 3.5), facilitated by large quantities of clean, open-access data. Such datasets have enabled significant advancements in predictive accuracy and reliability. In section A of the Appendix, we discuss the open-source ecosystem of software and datasets for physics-informed machine learning. However, in many industrial applications, similar data is unlikely to become open-source due to confidentiality concerns. Consequently, progress in these areas may be predominantly driven by private industry in a closed-source manner, potentially limiting the collaborative and open development of new methods in academia.

Secondly, while many of the machine learning methods discussed in this survey can predict solutions rapidly, it seems unlikely they will fully replace classical solvers (Grossmann et al., 2023). The extensive time required to train ML models often diminishes their attractiveness compared to classical methods. A more effective approach could involve using ML models to 'warm start' classical solvers, combining the speed of ML predictions with the robustness and reliability of traditional methods to achieve optimal performance.

Additionally, there is a need for more research on uncertainty quantification, especially for industrial applications operating in data-scarce environments. Methods like neural processes (section 4.3), which can be trained on extensive prior knowledge and fine-tuned with minimal contextual data, are particularly promising. These models not only adapt quickly but also provide valuable uncertainty estimates, which are crucial for informed decision-making in practical settings. This need is further emphasized in recent works such as Psaros et al. (2023), which provides a comprehensive overview of uncertainty quantification methods, metrics, and comparisons in scientific machine learning, and Mouli et al. (2024b), that explores the role of uncertainty quantification in improving out-of-domain generalization for PDE-based learning tasks. These contributions highlight the importance of probabilistic models like neural processes and Gaussian processes in addressing these challenges and represent a key area for future exploration.

The potential for data scarcity suggests that real-world data might be effectively augmented with simulated data from physical models to enrich data-driven models. However, this decision introduces a reality gap, which suggests that the source of the data should be carefully handled during model learning to carefully balance the sources of model bias (Scholz et al., 2023).

Finally, physics-based machine learning will inevitably follow the trend of foundation models (Bommasani et al., 2021). This trend builds on the multi-task, meta, and contextual learning paradigms discussed in section 4 by using large capacity transformer models that can learn from large datasets, as seen for weather forecasting in section 3.5. The goal of foundation models for physics applications is to learn a model across many settings, which can then be fine-tuned on new instances, potentially enabling data-efficient models through data-intensive pretraining (Subramanian et al., 2024; McCabe et al., 2023; Herde et al., 2024).

This survey demonstrates that combining physics knowledge with machine learning has the power to reshape predictive modeling by building systems that are not only efficient and accurate but also generalizable and trustworthy. By leveraging centuries of scientific knowledge alongside the flexibility of modern machine learning, researchers can address problems that were previously deemed intractable.

As the field advances, the synergy between physics-based insights and data-driven methods will redefine scientific discovery and engineering design, unlocking solutions to challenges in domains as varied as climate modeling, materials science, and industrial process optimization. The future of this interdisciplinary field lies in striking the perfect balance between computational rigor, data efficiency, and practical applicability, ensuring its impact across both academic and industrial landscapes.

**Acknowledgments**

We wish to thank the anonymous reviewers for their feedback on this survey, and providing additional references to related work. This work was partly funded by Hessian.ai through the project 'The Third Wave of Artificial Intelligence – 3AI' by the Ministry for Science and Arts of the state of Hessen, by the grant "Einrichtung eines Labors des Deutschen Forschungszentrum für Künstliche Intelligenz (DFKI) an der Technischen Uni- versität Darmstadt", and by the Hessian Ministry of Higher Education, Research, Science and the Arts (HMWK). O.W. acknowledges the financial support provided by the Deutsche Forschungsgemeinschaft (DFG, German Research Foundation, project number 492770117).

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

# A  Open-source Libraries and Datasets

This section reviews the open-source projects that are related to physics-informed prediction. ★ denotes the number of GitHub stars at the time of writing.

`lululxvi/DeepXDE` [2100★] is a library for PINNs and Deep Operator models that is implements for `PyTorch`, `Jax` and `TensorFlow` backends. The library accommodates intricate domain geometries without mandating mesh generation, utilizing diverse primitive shapes like intervals, triangles, rectangles, polygons, disks, ellipses, cuboids, spheres, hypercubes, and hyperspheres, with the ability to construct other shapes using constructive solid geometry operations. It also handles geometries represented by point clouds. DeepXDE offers five boundary condition types, including Dirichlet, Neumann, Robin, periodic, and a general BC adaptable to arbitrary domains or point sets, alongside approximate distance functions for hard constraints. The library incorporates diverse sampling methods such as uniform, pseudorandom, Latin hypercube sampling, Halton sequence, Hammersley sequence, and Sobol sequence. DeepXDE's loosely coupled components contribute to its well-structured nature, enabling high configurability and ease of customization to cater to evolving requirements.

`pnnl/neuromancer` [601★] (Neural Modules with Adaptive Nonlinear Constraints and Efficient Regularizations) is a library for solving parametric constrained optimization problems, physics-informed system identification, and parametric model-based optimal control using automatic differentiation. It contains implementation of PINNs, neural ODEs, Koopman operator-based timeseries models. It is written in `PyTorch` and integrates optimization libraries such as cvx.

`NVIDIA/modulus` [456★] is a library for physics-based ML written in `PyTorch`. While focusing on Neural Operators and PINNs, it also supports projects involving diffusion models and graph neural networks. It also has a front-end for processing symbolic equations, and a back-end for multi-node, multi-GPU large-scale training. It also supports complex domains, contraints and boundary conditions.

`SciML` is a collection of libraries in `julia` for physics-based ML. To date, it contains 156 repositories, including `SciML/DifferentialEquations.jl` [2700★] for differential equation solvers and `SciML/DiffEqFlux.jl` [806★] for neural ODEs, `SciML/NeuralPDE.jl`, [852★] for PINNs and `SciML/NeuralOperators.jl` [200★] for neural operators. The SciML suite in Julia leverages the language's design, emphasizing performance and ease of use, allowing users to seamlessly combine symbolic mathematics, automatic differentiation, and efficient numerical solvers. This ecosystem offers a variety of solvers for differential equations and a flexible interface for building models, enabling a streamlined process for scientific simulations and model development. Compared to Python, Julia's SciML packages often demonstrate superior performance due to Julia's focus on high-performance computing and its just-in-time (JIT) compilation, resulting in faster execution times for numerical simulations and complex scientific computations. However, Python libraries such as JAX also provide JIT optimization as well.

For neural operators, `neuraloperator/neuraloperator` [1600★] , in PyTorch, is the accompanying implementation from Li et al. (2021a). The github profile also has reference implementations of physics-informed neural operators (Li et al., 2021b), graph kernel network (Li et al., 2020b), (Li et al., 2022a) and geometry-aware Fourier neural operator (Li et al., 2022b).

Many more libraries exist. For neural differential equations, there is `NeuroDiffGym/neurodiffeq` [554★] by (Chen et al., 2020) in PyTorch. However, somewhat confusingly, the 'neural differential equations' implemented in this library are essentially PINNs, but the authors cite the older literature. For PINNs, there is `analysiscenter/pydens` [266★] and `Photon-AI-Research/NeuralSolvers` [118★] in PyTorch, which implements a few extensions to PINN models. `idrl-lab/idrlnet`, also tackling PINNs in PyTorch, but can handle complex domain geometries. `tensordiffeq/TensorDiffEq` [102★] also implements PINNs, but using Tensorflow 2. In general, while these libraries appear impactful, they are typically older and less maintained as those discussed above.

For evaluation, `pdebench/PDEBench` [545★] by Takamoto et al. (2022) is a benchmark suite for PDE solvers to evaluate ML-based methods across a range of PDEs, parameters, and boundary conditions. Additionally, `PDEArena` [241★] by Gupta and Brandstetter (2022) provides an extensible platform for benchmarking PDE

solvers, while `APEBench` [59★] focuses on adaptive physics-based machine learning techniques (Koehler et al., 2024). `nn-benchmark` [25★] is designed for evaluating neural networks on scientific computing tasks (Otness et al., 2021), and `The Well` [746★] serves by Ohana et al. (2024) as a benchmark for geophysical modeling and machine learning in subsurface environments.

