# OpenReview forum: "Machine Learning with Physics Knowledge for Prediction: A Survey"
_TMLR — Accepted by TMLR_

### Review · Reviewer_4egC · 2024-10-12

**Summary Of Contributions:**

This work covers an important survey about the use of machine learning in the physical sciences and computational fluid dynamics community. In particular, it provides a comprehensive review of works that incorporate physics into the learning process via various mechanisms including through the architecture, loss function and in the data itself, i.e., meta-learning and data augmentation.

**Audience:**

Yes

**Broader Impact Concerns:**

I do not have ethical or broader impact concerns with this work.

**Claims And Evidence:**

Yes

**Requested Changes:**

I would recommend acceptance of this paper upon revision and review. In particular, the main critical component is to add the below references and discussions as well as the references mentioned in the weakness section so that the survey is thorough to cover all the state-of-the-art SciML methods and their advantages and limitations.

- Need to add clear contribution of this work in comparison to the similar survey by Wang et. al. and add reference with appropriate discussion.
- Also I think MeshGraphNet GNN-based models need to be added and thoroughly discussed especially since GraphCast which is based on these methods is discussed.
- I think breaking up Section 3 into different sections and reorganizing it would also be helpful for the flow (see the notes in weaknesses) and possibly have separate sections for the ODEs and dynamical systems works from the PDEs works.

**Strengths And Weaknesses:**

## Strengths
- Nice comparison of methods that add physics into the architecture or loss function vs. just having physics satisfied in the data.
- Nice overview of an underexplored and important topic of using machine learning in scientific fields through PDEs and so the survey is very relevant and needed in the field.
- Nice overview in the introduction and the challenges for adopting ML for real-world scientific applications especially with limited data.
- Good survey across different state-of-the-art methods for SciML, e.g., PINNs and Neural Operators rather than just focusing on one of them.
- Nice perspective from the practical engineering community
- Good that boundary conditions are introduced and the three common ones are addressed. See Saad et. al "Guiding continuous operator learning through Physics-based boundary constraints", ICLR, 2023 for a general representation of linear boundary conditions using G operator,
- Nice connection at the beginning of section 3 that there are other areas of machine learning where domain knowledge is incorporated through the architecture, i.e., CNNs in computer vision.
- Good motivation on solving PDEs on very fine spatio-temporal resolutions and the challenges, need for ML methods, ways to improve them.
- Nice discussion and overview of the limitation of PINNs in terms of the optimization.
- Good discussion about the comparison to PINNs and classical methods, e.g., FEM and the current limitations of PINNs on practical problems.
- Nice physical application discussion in Section 3.7 and the importance of invariances and conservation.
- Table 3 provides a useful and practical summary
- Good motivation in identifying the limitations in applying these methods in industry and with real sparse data
- Good survey to bridge the gap between classical numerical methods and ML models and emphasizing that the accuracy of the ML models still does not match that.

## Weaknesses
-  Missing citation of another overview of ML for physics methods: Wang, R., Yu, R., "Physics-Guided Deep Learning for Dynamical
Systems: A Survey", https://arxiv.org/pdf/2107.01272, 2023.
- It is true that physics-informed methods are useful for weather forecasting and FourCastNet is cited but GraphCast which is state-of-the-art is purely data-driven GNN method and Pangu-Weather is data-driven and Transformer based. Both works should be cited.
- The Solve column in Table 1 is a bit unclear with the description of the methods.
- While PINNs and Neural Operators are discussed, references to another state-of-the-art GNN-based MeshGraphNet (Pfaff et. al, "Learning Mesh-Based Simulation with Graph Networks", ICLR 2021) model is not directly discussed - only through GraphCast which is based on this architecture.
- Missing reference Saad et. al, "Guiding continuous operator learning through Physics-based boundary constraints", ICLR, 2023 on hard-constraining Neural Operators to satisfy boundary conditions and Negiar et al. "Learning differentiable solvers for systems with hard constraints", ICLR, 2023 on enforcing PDEs as a hard-constraint rather than a soft-constraint in PINNs, which can also be added to the Operator Learning section.
- Additional references include Chalapathi, N. et. al, "Scaling physics-informed hard constraints with mixture-of-experts", ICLR 2024 and Du et. al, "Neural Spectral Methods: Self-supervised learning in the spectral domain", ICLR 2024 especially the latter would be relevant to cite since spectral methods are also discussed.
- For the physics-informed loss on the medical and ODEs, I think Wang et. al, "Bridging physics-based and data-driven modeling for learning dynamical systems", L4DC, 2021 would be good to cite.
- Missing reference to LeVeque, "Finite Difference Methods for Ordinary and Partial Differential Equations", SIAM, 2007 and LeVeque, "Finite-Volume Methods for Hyperbolic Problems", Cambridge University Press, 2002 and Hughes, "The Finite Element Method: Linear Static and Dynamic Finite Element Analysis" for the numerical methods
- A smoother transition between sections 2.1-2.2 and 2.3 would be nice. It currently jumps from ODEs and PDEs to probabilities and loss functions. It is also unclear where $\mathcal{L}$ is defined in Eqn. 5.
- I think the NN background in Section 2.4 should be familiar to the reader and can be moved to an appendix.
- On page 5, add reference for a RNN
- While the background on Neural ODEs is thorough, I'm not sure if it is needed. Additional references here include Ott et. al, "ResNet After All? Neural ODEs and their Numerical Solution", 2020 and Krishnapriyan et. al, "Learning continuous models for continuous physics", Communications Physics 6 (1), 319, 2023.
- There is also a nice potential connection to highlight between the adjoint method in Neural ODEs and the adjoint method using by Per-Olof Persson and Zhar in constrained optimization numerical methods for PDEs.
- On page 8 it would be good to add reference to the Multi-wavelet Neural Operator: Gupta et. al, "Multiwavelet-based Operator Learning for Differential Equations", NeurIPS, 2021.
- For clarity in Eqn. 13 should refer back to Eqn. 3 for the PDE definition and variables
- Typo in Poisson's Eqn at the end of page 8.
- Another method to address the limitations with PINNs to be cited is Subramanian, S., et. al, "Adaptive self-supervision algorithms
for physics-informed neural networks," arXiv:2207.04084, 2022, which proposes an adaptive update of collocation points.
- I've also seen in practice that BFGS works better in PINNs optimization. It would be nice if the authors explained a bit more into why.
- The authors may consider reordering the subsections 3.2 and 3.3. There is quite a jump from learning the solution with PINNs in 3.2 to learning the equations in 3.3 and going back to ODEs.
- The Moore-Penrose inverse is briefly discussed at the end of page 11 and I think the numerical challenges of computing it should also be addressed.
- Since POD and DMD is mentioned on pg. 14, I think adding references to works on ROM for background would also be useful.
- There are recent works on general geometries Li, Z. "Geometry-Informed Neural Operator for Large-Scale 3D PDEs", NeurIPS 2023 and Li, Z. "Fourier Neural Operator with Learned Deformations for PDEs on General Geometries", Journal of Machine Learning Research 24 1-26, 2023, which should be added to address the limitations of a fixed uniform mesh
- Multi-wavelet Neural Operator should be included with the spectral methods on page 16
- Chebyshev is spelled incorrectly
- Du et. al, "Neural Spectral Methods: Self-supervised learning in the spectral domain", ICLR 2024 is another recent spectral-based method to add.
- There is a discussion about adding PINNs to Neural Operators with PINO. PINO paper and Saad et. al "Guiding continuous operator learning through Physics-based boundary constraints", ICLR, 2023 show that sometimes adding this physics-informed loss does not improve performance.
- There is a jump from Neural Operators in 3.4 to Koopman Operator Theory in 3.5 and the flow is not that clear.
- Mouli, S.C., et.al, "Metaphysica: Ood robustness in physics-informed machine learning", ICLR 2024 improves upon SINDy for OOD learning and should be cited.
- For details on weather benchmarking of DL models, it would be good to cite Karlbauer, M. "Comparing and Contrasting Deep Learning Weather Prediction Backbones on Navier-Stokes and Atmospheric Dynamics", https://arxiv.org/abs/2407.14129, 2024.
- I think the weather forecasting part should be it's own section. It doesn't really fit in Section 3.5
- For diffusion based models for weather forecasting see PreDiff (Zao et. al, NeurIPS 2023) and GenCast (Price et. al, 2024)
- The PINNs discussion in 3.6 could be moved up to the PINNs section where it is first discussed. In general, PINNs seem to be randomly discussed throughout the paper (also see Section 4.1) and the organization/outline could be improved.
- Minor: Capital S typo in PINNs at the top of page 23
- The motivation of Hansen et. al, "Learning Physical Models that Can Respect Conservation Laws" is satisfying conservation and can be discussed in Section 3.7. In addition see Richter-Powell, J., "Neural Conservation Laws: A Divergence-Free Perspective", NeurIPS, 2022, Liu et. al, "Harnessing the Power of Neural Operators with Automatically Encoded Conservation Laws", ICML 2024, Liu et. al, "INO: Invariant Neural Operators for Learning Complex Physical Systems with Momentum Conservation", AISTATS, 2023 and Beucler, T., et. al, "Enforcing Analytic Constraints in Neural-Networks Emulating Physical Systems". Physical Review Letters, 126(9):098302, 2021 and adding the corresponding Neural Operator papers to page 27.
- More recent methods using diffusion methods for PDEs, e.g., DiffusionPDE (Huang et. al, "DiffusionPDE: Generative PDE-Solving Under Partial Observation", https://arxiv.org/abs/2406.17763, 2024) could also be discussed.
- The authors may also want to consider discussing uncertainty quantification (see Psaros et. al, "Uncertainty Quantification in Scientific Machine Learning: Methods, Metrics, and Comparisons. Journal of Computational Physics, 477:111902, 2023) and out-of-domain generalization  (see Mouli et. al, "Using Uncertainty Quantification to Characterize and Improve Out-of-Domain Learning for PDEs", ICML 2024) especially since probabilistic models, e.g., Neural Processes) and Gaussian Processes are discussed. I see this is discussed as future work.
- I'm not sure why Neural Processes explicitly at put in Section 4 with meta-learning and multi-task learning subsections. I think it would fit better in the sections with the discussions of the other models.
- Additional recent works to add include DINo (Yiu et. al, Continuous PDE Dynamics Forecasting with Implicit Neural Representations, ICLR 2023) and CROM (Chen et. al, "CROM: Continuous Reduced-Order Modeling of PDEs Using Implicit Neural Representations", ICLR 2023).
- Typo on page 34, "conversation" should be "conservation"
- The constraint method in Hansen et. al does not need to be a soft constraint and with the $sigma_G$ parameter set to 0 it can be a hard-constraint projection method and satisfy the constraint exactly unlike soft-constrained PINNs type methods.
- For additional references of ML in the automotive domain, see Ashton, N. et. al,  "WindsorML--High-Fidelity Computational Fluid Dynamics Dataset For Automotive Aerodynamics", NeurIPS Datasets and Benchmarking Track, 2024, Ananthan, V et. al, Machine Learning for Road Vehicle Aerodynamics, SAE Technical Paper and Li et. al, Geometry-Informed Neural Operator for Large-Scale 3D PDEs", NeurIPS 2023.
- A stronger concluding sentence would be good.
- Subramanian et al., 2024 is cited twice on page 36.
- For more PDE foundation models to add see Herde et. al, Poseidon: Efficient Foundation Models for PDEs, 2024.

---

> ### Author Response · Authors · 2025-03-04
> **Response to Reviewer 4egC**
>
> We would like to thank the reviewer for their extensive feedback, which we found immensely helpful. Please let us know if you have additional issues (outside of (c))
>
> Regarding the requested changes:
>
> a) We now cite and discuss the Wang et al. survey in the introduction
>
> b) We briefly discuss the MeshGraphNet work in the context of GraphCast
>
> c) We have not yet changed the structure of Section 3, as we wish to discuss this (large) change with the other reviewers first.
>
> Regarding the wider weaknesses
> 1) We have updated Section 1 to discuss Wang et al. 2023
> 2) We discuss FourCastNet, GraphCast and PanguWeather equally in Section 3.5. Which passage are you referring to with this weakness comment?
> 3) We have expanded the column headings in Table 1 to be informative
> 4) We now briefly discuss the MeshGraphNet work in the context of GraphCast in Section 3.5. Would you like a longer discussion of the model before the weather application?
> 5) We now discuss Saad et al. 2023 and Negiar et al. 2023 in the NO section
> 6) We now discuss Du et al. 2024. We have opted not to cite Chalapathi et al. 2024 as it looks at a scalable implementation rather than incorporating domain knowledge.
> 7) We have discussed Wang et al. 2021 in Section 3.3
> 8) We now cite LeVeque and Hughes in Section 2
> 9)  One additional paragraph has been added at the beginning of Section 2.3 for a better flow.
> 10) RNN reference added
> 11) Ott et. al. and Krishnapriyan et. al. added to section 3.1
> 12) We now discuss  Per-Olof Persson and Zhar methods in section 3.1. Very good point!
> 13) We reference Gupta et al. 2021 in the NO section 3.4
> 14) For clarity in Eqn. 13 should refer back to Eqn. 3 -- done!
> 15) Typo in Poisson's Eqn -- fixed!
> 16) We have added Subramanian, S., et. al. to the PINNs section
> 17) We expanded the quasi-newton PINN solver discussion
> 18) We reordered 3.2 and 3.3 for flow
> 19) We flag the MP inverse challenges in section 3.2
> 20) We think a ROM background is a bit out of scope. We have ensured Section 3.4 and 3.5 are well-cited with background material.
> 21) 'There are recent works on general geometries...' -- It is added in Section 3.4 wrt. domain agnostic neural operators.
> 22) Multi-wavelet Neural Operator should be included -- added to 3.4
> 23) Chebyshev is spelled incorrectly -- fixed!
> 24) Du et. al is now added to 3.4
> 25) We discuss Saad et. al in Section 3.4 now
> 26) Thanks for the suggestion. We did small changes at the end of Section 3.4 to point to LVM. Also, in the motivation of Section 3.5, we added a reference back to section 3.4 and 3.3 that they can be seen as LVMs and that LVM gave us an alternative perspective on these methods. Lastly, we added a short reference back to DeepOnets to the linear latent dynamics subsection as they can be seen as nonlinear LVMs.
> 27) Mouli, S.C., et al. is added in Section 3.5 at the end of DMD paragraph.
> 28) Karlbauer, M. has been added to 3.5
> 29) The weather forecasting models use variations of latent dynamics models, so we believe it is a relevant application domain of Section 3.5
> 30) We now discuss the newer diffusion-based weather models in 3.5
> 31) We wanted to separate the discussion of domain knowledge in the 'forward' problem (simulation), the inverse problem (model learning) and the challenges of generalization in the survey (i.e. Table 1). This is why PINNs are discussed throughout the survey.
> 32) Minor: Capital S typo in PINNs at the top of page 23 -- fixed!
> 33) Thanks for the additional reference regarding the enforcement of conservation properties in neural fields, operators, and processes. We have added these to the discussion in Section 3.7
> 34) We believe Neurips 2024 papers are a bit out of scope for the survey since we wrote in in early 2024
> 35) We have added the UQ papers to the further work section
> 36) Neural Processes are data-driven models are they are given additional contextual data (e.g. Figure 1). While NP can also use architectural inductive biases too, we wanted to frame them as a means of data-driven generalization via the context.
> 37) We discuss DINo and CROM in the LVM section 3.5
> 38) Typo on page 34, "conversation" should be "conservation" -- fixed!
> 39) Re: soft constraints, you could argue that any soft constraint can be hard 'hard' by increasing the weighting of the penalty term to the limit. We have added a note to this in the discussion though.
> 40) Thanks for the recommendations for the automotive domains! However, the suggested references are highly
> domain-specific, which falls a bit outside the general scope of our discussion. Moreover, the techniques deployed in both references have been well-addressed in this survey. Therefore, we would not consider them.
> 41) A stronger concluding sentence would be good. -- The conclusion is updated to provide stronger statements.
> 42) Subramanian et al., 2024 is cited twice on page 36. -- fixed!
> 43) For more PDE foundation models to add see Herde et. al, Poseidon: Efficient Foundation Models for PDEs, 2024. -- added!

---

> > ### Comment · Reviewer_4egC · 2025-03-04
> > **Response to Rebuttal**
> >
> > I would like to thank the authors for the detailed update and response and adding the requested references. For a minor clarifying point, Hansen et al., 2023 does not use a penalty parameter and can be a hard-constraint method. As shown in the paper, the conservation error is 0 when the noise in the constraint sigma_G = 0 and then we have a constrained least squares problem.
> > With the specified revisions, I am voting for acceptance of the paper.

---

### Review · Reviewer_MFpK · 2024-11-03

**Summary Of Contributions:**

This paper covers the recent development of ML + Physics, with a focus on PDE approximation. The survey is comprehensive.

**Audience:**

Yes

**Broader Impact Concerns:**

Not applicable.

**Claims And Evidence:**

Yes

**Requested Changes:**

A schematic graph at the beginning to describe the relationship between different concepts/approaches would be great.

**Strengths And Weaknesses:**

Strength: comprehensive survey.

Weakness: I don't foresee any. This is a much needed paper in this evolving domain.

---

> ### Author Response · Authors · 2025-03-04
> **Response to review MFpK**
>
> We wish to thank the reviewer for their feedback.
>
> We were hoping the reviewer could expand on their request 'A schematic graph at the beginning to describe the relationship between different concepts/approaches would be great.', which the draft currently has Table 1, which describes the similarity and differences of all the subsections for four variables, and Figure 1, which visualise a subset of these concepts in a shared visual language. Could you let us know what sort of additional graph you think would explain the connections further?
>
> Also, please let us know if you have any additional feedback!
>
> Also, do you agree with Reviewer 4egC’s proposed change “I think breaking up Section 3 into different sections and reorganizing it would also be helpful for the flow (see the notes in weaknesses) and possibly have separate sections for the ODEs and dynamical systems works from the PDEs works.”?

---

### Review · Reviewer_Zh69 · 2025-02-16

**Summary Of Contributions:**

The authors itemize and summarize various lineages of methods that incorporate both physics and machine learning. The taxonomy explores a few axes of variation, in terms how physics is imposed, the structure of the interaction that may be learned, and how general the model learned is.

**Audience:**

Yes

**Claims And Evidence:**

Yes

**Requested Changes:**

It seems to me that a few concepts are confounded in the authors' taxonomy. This might be a matter of definitions being unclear, or of deeper confusion. I recommend that it resolved other by crisper and more consistent definition of important terms, or by adjusting the actual taxonomy. I will mention what seem to me to be the biggest examples of the difficulties.

## Figure 1

I applaud the authors for diagramming the taxonomy they are understanding. great idea. Figure 1 seems plausible, but there are some questions that I feel I must request answers for. Let us examine the categories

* Physics-informed models
* Physics-informed losses
* Data-driven generalization
* Physics-informed data augmentation
* Learning with contexts

Since all these topics relate to the methods taxonomised in this paper, would it not make sense to link to which sections of the paper deal with these, e.g. should "physics informed losses" link to section 3.2

Why is there a color bar on this graph from "informed" to "uninformed"? There are no shade of gray on the graph that I can see.

## Table 1

The central distinctions made between "solve/infer/generalize" seem very important but are only briefly explained:

> learning can refer to the solution (or simulation) of a physical system, inferring the
> physical system from data, and learning a model that generalizes across several instantiations of the physical system, e.g., different boundary condition"

I think this need to be mathematically formalised, since we use it a lot. "Solve" sounds like "produce values on the solution surface $u_\theta$, infer means something like learn $\theta$ which is a parameter generating  $u_\theta$, and generalize means something like learning $u_\theta| x$ for some covariate $x$? If so, may I request the authors clarify?

The entries in the column headed _inductive bias_ suggest a non-standard use of the term, or at least I don't understand the use here fully because it seems to mean something more specific than I am used to. What does "Structure objective" mean, for example? It sounds like the authors wish to describe when and how the known laws of physics are applied to the learning algorithm. If so, this is an excellent idea, but the table should be clarified so that we know what they mean. Is the "Structure objective" the part of the inductive bias which includes known laws of physics? Or is the the structure of the generating process which produces the data, such as a hypothesised structural equation model underlying the data? Or is it both?

Concretely, when I say Structural Equation Model, I mean this kind of thing:
```text
     +--------+       +--------+       +--------+
     | θ      |--->   | x_t-1  |--->   |  x_t   |---> ...
     +--------+       +--------+       +--------+
                        |               |
                        v               v
                +--------------+   +--------------+
                |  y_t-1       |   |   y_t        |
                +--------------+   +--------------+
```
The above is the graphical model encoding of a system identification problem. See, e.g. Koller and Friedman 2009. I suspect that conceptually separating the graphical structure, and the physics structure of the problems might make the paper much clearer.


## 3.6 Bridging System Identification and Function Approximation

Section 3.6 is a good example of a non-standard terminology use and confusing taxonomy used together. _Function approximation_ (introduced in section 2.4) is simply learning to approximate a function. System identification (section 2.3) is about learning systems with serial dependence in time conditional upon known inputs $w_t$ and latent parameter $\theta$.

>System identification involves defining the dynamics of a system through a sequence of inputs $\boldsymbol{w}_t$, measurements $\boldsymbol{y}_t$, and an initial state $\boldsymbol{x}_1$ over a finite horizon $\mathcal{T}$ as a trajectory, i.e. $\boldsymbol{Y}=\left[\boldsymbol{y}_1, \ldots, \boldsymbol{y}_T\right]$ . [where ] the state $\boldsymbol{x}_{t+1}=\boldsymbol{f}_\theta\left(\boldsymbol{x}_t, \boldsymbol{w}_t\right)$ and the observation $\boldsymbol{y}_t=\boldsymbol{g}_{\boldsymbol{\theta}}\left(\boldsymbol{x}_t\right)$ are both parameterized by $\boldsymbol{\theta}$, […] A probabilistic approach considers the joint distribution $p(\boldsymbol{Y}, \boldsymbol{X} \mid \boldsymbol{W}, \boldsymbol{\theta})$ which, continuing the Markov assumption, factorizes into $p\left(\boldsymbol{x}_1\right) \prod_{t=1}^T p\left(\boldsymbol{y}_t \mid \boldsymbol{x}_t, \boldsymbol{\theta}\right) p\left(\boldsymbol{x}_{t+1} \mid \boldsymbol{x}_t, \boldsymbol{w}_t, \boldsymbol{\theta}\right)$.

So far I think this is not to controversial. I have seen the problem of identifying $\theta$ as system identification in the literature, but also sometimes the identification of the unobserved $X$ is included. Then the uasge of this term gets confusing:

>System identification approaches lie on a spectrum depending on modeling assumptions. If a lot of domain knowledge is assumed, the dynamics model will be highly structured. If correct, it will generalize well but will most likely be inflexible to un-modelled phenomena. On the other hand, one could model with no modeling assumptions and use a black-box function approximator

I am not sure what is being said here. As far as I understand it, almost _everything_ in this paper is function approximation according to the definitions introduced by the authors. So there is nothing to "bridge". If we read further into section 3.6 I cannot work out what the common theme is: there are GANs, there are PINNs, there are GPs with a physics-informed kernel.  What is the common theme in 3.6? Possibly it is simply _system identification_. If so great! This could be clarified in the first paragraph of the section.

## What are the dimensions of the paper's taxonomy?

_System identification_ seems to me to be an assumption about the underlying graphical model generating the data, and that does not need to involve physics, necessarily. But we can imagine that it *might*, if for example the system to be identified was generated by the laws of physics. If I can propose my idea about what the authors are trying to get at here I think there might be two separate axes of variation, the probabilistic graphical model underlying the system of interest, and the means by which physics is integrated. For example, my way of writing that would be as follows:

| Section / Technique              | Generating Structural Equation Model      | Means of Physics Integration                     |
|-----------------------------------|----------------------------------------------------|---------------------------------------------------|
| **3.1 Neural ODEs/PDEs**         | Continuous-time state-space models                 | Architectural physics (ODE/PDE structure)        |
| **3.2 Physics-Informed Losses**   | Deterministic regression with residual constraints | Strong physics (PDE residuals as objectives)      |
| **3.3 Hamiltonian/Lagrangian NNs**| Structured dynamical systems                       | Full physics (energy conservation built-in)       |
| **3.4 Neural Operators**         | Operator-valued regression                        | Physics-informed architectures (Green's functions)|
| **3.5 Koopman Operators**         | Linear latent space models                         | Physics-inspired (spectral decomposition)       |
| **3.6 Hybrid System ID**          | Hierarchical Bayesian models                       | Partial physics (blackbox + known terms)         |
| **3.7 Invariance Methods**        | Factor graphs with symmetry constraints            | Strong physics (Noether's theorem applications)   |
| **4.1 Multi-task Learning**     | Hierarchical Bayesian models                    | Data-driven               |
| **4.2 Meta Learning**           | Hierarchical Bayesian models           | Data-driven                  |
| **4.3 Neural Processes**       | Hierarchical Bayesian models              | Data-driven                  |

Here is an alternative table that categories the methods differently, generated by an LLM:

| Section & Method              | Integration Mechanism          | Learning Objective              | Generating Structural Equation Model     |
|-------------------------------|----------------------------------|----------------------------------|---------------------------------------|
| **3.1 Neural ODEs/PDEs**       | Architectural constraints       | Solve PDEs/Infer parameters      | Continuous-time dynamics              |
| **3.2 PINNs**                  | Loss-based physics              | Solve PDEs                       | Structured objective (PDE residuals) |
| **3.3 Hamiltonian/Lagrangian** | Architectural constraints       | Infer system parameters          | Symplectic structure                  |
| **3.4 Neural Operators**        | Architectural constraints       | Generalize across conditions     | Multi-instance data                   |
| **3.5 Koopman Autoencoders**    | Architectural constraints       | Infer system parameters          | Structured latent models             |
| **3.6 Hybrid System ID**        | Mixed architectural/loss-based   | Solve & Infer parameters         | Structured models/objectives          |
| **3.7 Invariance Priors**       | Architectural constraints       | Solve & Generalize              | Structured models/features/data/obj.  |
| **4.1 Multi-task Learning**     | Data-driven                     | Generalize across tasks          | Multi-instance data                   |
| **4.2 Meta Learning**           | Data-driven                     | Generalize across conditions      | Multi-instance data                   |
| **4.3 Neural Processes**       | Data-driven                     | Solve & Generalize              | Contextual data                       |

Both these seem like reasonable taxonomies to me, but I am open to other ones. They key point is that the paper should consistently use terminology so that we can infer such a taxonomy.

**Strengths And Weaknesses:**

This is a really interesting paper that has done a lot of useful work, but has weaknesses in the execution. I suspect that this mostly terminology difficulties and easy to fix, but I am not 100% certain.

Strengths:

* Comprehensive overview of many methods
* Lineages of research are well connected
* surfaced some excellent lesser-known papers.

Weaknesses

* Non-standard terminology use — I do not think that the distinctions that they draw with  _inductive bias_  and _function approximation_  are used in the standard way
* taxonomy seems muddy to me; I think an attempt made to draw out some different axes of variation amongst the ML models, but the axes are either confounded or not clearly communicated
* many small typos and unclear statements (For the sake of time I will skip these in my initial review and add a comment with them later)

---

> ### Comment · Reviewer_Zh69 · 2025-02-17
> **Typos/minor notes**
>
> * title "3.1 Learning Differential Models from Data" Should this be "Learning Differentiable Models from Data"?
> * “Perhaps the earliest endeavor of learning differential equations is the ResNet architecture (i.e.,residual flow block) proposed by He et al. (2016),"  “Perhaps the earliest _predecessor_ to _neural_ learning of differential equations is the ResNet architecture (i.e.,residual flow block) proposed by He et al. (2016)" (learning ODEs without NNs is much older, depending on your definition; adjoint models were certainly common in meterology in the 1990s)
> * "Training neural ODEs. The common optimization paradigm in learning neural function approximations is backpropagation through learning objectives. When learning ODEs, the objective is generally to fit the parameters θ of f...This paradigm typically requires a sufficiently large dataset to avoid overfitting for learning highly non-linear ODEs; otherwise, one should resort to classical (Voss et al., 2004) or _neura_l (Chen et al., 2021; Forgione and Piga, 2021) system identification methods." What is the distinction we are drawing here? Are we not already talking about neural learning of ODEs? It sounds like the authors are saying "if the data set is not big enough we should switch from neural ODEs to neural ODEs", but I think this is not what they mean
> * Augmented neural odes: "This can prevent the trajectories of the system from intersecting each other and thus allows the representation of more general system behaviors". I think the wording here could be clearer with something like "augmented neural ODEs in _permit_ the (observed) trajectories to intersect and that allow learning more general dynamics over the (observed) state space".
> * “3.4 Operator Learning: So far in this survey, we have considered learning a single solution uθ(x) to a differential equation, which requires solving the dynamical system over the domain of interest for a given set of system parameters, such as boundary conditions. A more general model to learn would be the mapping from system parameters to solution, which could enable extrapolating to new system parameters without having to solve the system.”. What does this mean? I am not sure of the definitions of all the terms here, but I cannot work out what the distinction is. Perhaps write out an equation, or introduce a definition?
>
> * 3.4 "The little regularization and flexibility of neural operators come at the cost of demanding plenty of training data to learn the desired operator." -> "The limited attention required for  regularization and high flexibility flexibility of neural operators come at a high cost in volume of training data" ?
> * "The development of operator learning architectures is strongly influenced by the analysis of linear differential equations. However, the theoretical foundations solely shaped the design of the neural architecture, which is geared towards approximating non-linear operators. In contrast, Koopman operator theory (section 3.5) exclusively addresses dynamical systems represented by linear operators." The authors need to say in which sense they mean "linear" here, because the connection  Koopman operators operating linearly in "feature" space to the dynamics of the learned operators is not trivial (and explained pretty well in section 3.5) ; also I'm not sure whcih theoretical foundations we are talking about here. Could you clarify?
> * 3.7 title "Invariance-Driven Priors". I don't think this is a standard use of "driven", Perhaps "Invarance-motivated priors" or "Invarant Priors" or "Priors imposing symmetry" or something like that?
> * Appendix: “Neural differential equations’ are essentially PINNs,” <- A bold statement; link to supporting section in the main body if you wish to make this claim
> * Appendix: The list of PDE benchmarks an evaluations is not complete. I would recommend
>
>   * [PDEArena](https://pdearena.github.io/pdearena/)
>   * [APEBench](https://tum-pbs.github.io/apebench/)
>   * [karlotness/nn-benchmark](https://github.com/karlotness/nn-benchmark)
>   * [The Well](https://polymathic-ai.org/the_well/)

---

> > ### Author Response · Authors · 2025-03-04
> > **Response to Reviewer Zh69**
> >
> > Thank you for the thoughts and extensive review, we really appreciate it!
> >
> > **Figure 1.** We have now linked to the relevant subsection in the figure caption and removed the gradient in the legend.
> >
> > **Table 1.** We have expanded the column headings and inductive bias cells to be more informative. The column headings refer to the forward problem (simulation), inverse problem (model learning) and generalization problem. The forward and inverse formulation is quite well documented in the ODE / PDE literature and we have added a discussion in Section 2.3.
> >
> > **System identification.** Thanks for pointing this out. We have expanded the introduction to section 2.3 and the heading of section 3.6 to be clearer in what we mean and clarify some of the definitions. You are right in that section 3.6 combines function approximation and specific physics knowledge for system identification in an interesting hybrid "grey-box" fashion.
> >
> > **Taxonomy.** We thank the reviewer for their suggestions and think they are interesting ideas. We like our Table 1 since the forward (solve) and infer (inverse) taxonomy is well established in the PDE literature. The generalization column is our contribution thanks to the advent of ML methods in this space.
> >
> > **Differential vs Differentable.** We specifically refer to differential models (ie NODEs) since most function approximators are differentable.
> >
> > **Wording issues.** Thanks for these concerns. We have amended and edited all these passages for clarity.
> >
> > **Neural differential equations’ are essentially PINNs.** We were referring to what the linked library called 'Neural differential equations’, it was not a general statement. We have amended the text for clarity. Sorry for the confusion.
> >
> > **Benchmarks.** Thanks for the list, we have added them to the appendix.
> >
> > Please let us know if you have additional concerns with the new draft.
> > Also, do you agree with Reviewer 4egC’s proposed change “I think breaking up Section 3 into different sections and reorganizing it would also be helpful for the flow (see the notes in weaknesses) and possibly have separate sections for the ODEs and dynamical systems works from the PDEs works.”?

---

### Author Response · Authors · 2025-03-04
**Rebuttal overview and extension request**

We wish to thank the reviewers for their feedback. The reviews went beyond our expectations and we believe the additional changes have improved the draft greatly. In this comment we will make a high-level summary of the changes. In reviewer reply comments, we will go into more depth regarding the specific reviewer feedback.

We would also like to follow up on the two-week rebuttal extension request we made privately to the Action Editor last week via email that we have not heard a verdict on. Since the three reviews came in significantly later than the four weeks TMLR indicates, we were unable to plan for the rebuttal period and were caught off guard during the past two weeks as several authors have been on business trips and/or vacation, so we were not able to invest enough time in incorporating the last review and some of the larger changes the reviewers alluded to.

Since the rebuttal extension has not been acknowledged yet, we will present the changes made so far to adhere with the original rebuttal period. As a summary, we believe the feedback was largely positive, with reviewers saying our submission was a 'comprehensive' 'overview of many methods' (MFpK, Zh69), 'Lineages of research are well connected' (Zh69), 'good survey to bridge the gap between classical numerical methods and ML models' (4egC), 'surfaced some excellent lesser-known papers' (Zh69), 'Good motivation in identifying the limitations in applying these methods in industry and with real sparse data' (4egC). The criticism mainly concerned some missing citations, terminology, and typos, which we believe we have been able to address. There are also some larger open questions regarding restructuring the survey for flow, which we would like to discuss with all reviewers during this period. As a result, the changes in the new version of the draft largely concern additional references, clarifying sentences, fixing typos, and improving the flow. These changes were largely in Section 2 and 3. Cyan represents the new text, while lavender represents the old text that has been replaced.

---

> ### Comment · Action_Editor_6kS2 · 2025-03-14
>
> Dear authors,
>
> Thanks for the response and preparing a revision.
>
> Reviewer Zh69: Can you discuss with the authors if your concerns have already been addressed, or if further revisions are needed?
>
> Since the paper is longer than usual, we can spend some further time revising the paper if reviewers and authors agree. However, let's hope for wrapping this up in a reasonable time.
>
> best, AE

---

### Decision · Action_Editor_6kS2 · 2025-04-27

**Recommendation:** Accept as is

**Comment:**

The paper is a long review of physics-connected machine learning. Such works are inherently useful and interesting for the community. The reviewers agree on acceptance on both criteria. The reviewers gave exceptionally thorough commentary and suggestions on the paper, which the authors addressed sufficiently. This is a well-written review paper that helps categorise a fast-moving domain. I recommend the survey certification.

**Audience:**

Yes

**Claims And Evidence:**

Yes